# Fast Projection-Free Approach (without Optimization Oracle) for Optimization over Compact Convex Set

**Chenghao Liu[1],    Enming Liang[1,*],    Minghua Chen[1,2,*]**
[1]Department of Data Science, City University of Hong Kong
[2]School of Data Science, The Chinese University of Hong Kong, Shenzhen

## Abstract

Projection-free first-order methods, e.g., the celebrated Frank-Wolfe (FW) algorithms, have emerged as powerful tools for optimization over simple convex sets such as polyhedra, because of their scalability, fast convergence, and iteration-wise feasibility without costly projections. However, extending these methods effectively to general compact convex sets remains challenging and largely open, as FW methods rely on expensive linear optimization oracles (LOO), while penalty-based methods often struggle with poor feasibility. We tackle this open challenge by presenting **Hom-PGD**, a novel projection-free method without expensive (optimization) oracles. Our method constructs a homeomorphism between the convex constraint set and a unit ball, transforming the original problem into an equivalent ball-constrained formulation, thus enabling efficient gradient-based optimization while preserving the original problem structure. We prove that Hom-PGD attains *optimal* convergence rates matching gradient descent with constant step-size to find an $\epsilon$-approximate (stationary) solution: $\mathcal{O}(\log(1/\epsilon))$ for strongly convex objectives, $\mathcal{O}(\epsilon^{-1})$ for convex objectives, and $\mathcal{O}(\epsilon^{-2})$ for non-convex objectives. Meanwhile, Hom-PGD enjoys a low per-iteration complexity of $\mathcal{O}(n^2)$, without expensive oracles like LOO or projection, where $n$ is the input size. Our framework further extends to certain non-convex sets, broadening its applicability in practical optimization scenarios with complex constraints. Extensive numerical experiments demonstrate that Hom-PGD achieves comparable convergence rates to state-of-the-art projection-free methods, while significantly reducing per-iteration runtime (up to 5 orders of magnitude faster) and thus the total problem-solving time.

## 1 Introduction

We consider constrained optimization where the objective is smooth, possibly non-convex, and the constrained set is compact convex. Although popular second-order methods, such as interior-point methods [PW00, Wri97] and cutting plane methods [B+15], achieve linear convergence rates, their per-iteration computational complexity scales super-linearly with the problem size, typically on the order of $\mathcal{O}(n^3)$ due to solving a linear system. Consequently, these methods become impractical for large-scale problems. Alternative approaches, such as projection-based gradient descent (PGD) first-order methods (see, e.g., [Bec17, ZPL22, ZL22]), provide a computational benefit for simple convex sets where orthogonal projections can be performed efficiently, such as Euclidean balls and boxes. Despite their slower convergence rate of $\mathcal{O}(1/\epsilon)$ in the convex smooth setting and $\mathcal{O}(1/\epsilon^2)$ in the non-convex smooth setting, these methods are favorable in practice due to their relatively low per-iteration cost. However, the projection operation, i.e., solving a convex problem with a quadratic objective over constraints, is computationally expensive except for simple constraint sets.

---

*Corresponding authors: Enming Liang (enming.cityu@gmail.com) and Minghua Chen (minghua@cuhk.edu.cn)

39th Conference on Neural Information Processing Systems (NeurIPS 2025).

Table 1: Summary of existing projection-free methods for solving optimization problems.

| Reference | Settings: Obj. | Settings: Ctr. | Key Assumption | Algorithm | Step-size[2] | Per-iteration Complexity | Convergence Rate[3] |
|---|---|---|---|---|---|---|---|
| [LMY23] | NC | Simplex | - | Hadamard Parameterization + Pertubed PGD | Constant | $\mathcal{O}(n)$ | $\mathcal{O}(\epsilon^{-2})$ |
| [LG23] | C | C | SC (Obj + Ctr Set) S (Obj + Ctr Set) SC & S (Obj + Ctr Set) | Sub-GD Alg. for the RD[4] | Implicit and Diminishing | QOO | $\mathcal{O}(\epsilon^{-1})$ $\mathcal{O}(\epsilon^{-0.5})$ $\mathcal{O}(\log \epsilon^{-1})$ |
| [Gri24b] | C | C | Upper Radial Obj | Accelerate Sub-GD Alg. for the RD[4] | Implicit or Vanishing | MO | $\mathcal{O}(\epsilon^{-0.5})$ |
| [MHSY25] | DC | C | - | Frank-Wolfe | Diminishing | LOO | $\mathcal{O}(\epsilon^{-2})$ |
| Theorem 1 Theorem 2 Theorem 3 | C SC NC | C C C | ND Minimizer - - | **Hom-PGD** (Sec. 3) | Constant | MO | $\mathcal{O}(\epsilon^{-1})$ $\mathcal{O}(\log \epsilon^{-1})$ $\mathcal{O}(\epsilon^{-2})$ |

[1] **Abbreviations**: C = "convex", NC = "non-convex", DC = "difference of convex", SC = "strongly convex", S = "Smooth", Obj = "objective", Ctr = "constraint", GD = "gradient descent", ND = "non-degenerate", RD= "radial dual", LOO = "linear optimization oracle", QOO = "quadratic optimization oracle", MO = "membership oracle".

[2] **Step-size**: (i) vanishing step-size: depends on $\epsilon$, (ii) diminishing step-size: decreases as $\text{poly}(1/K)$ with the number of iterations $K$, (iii) constant step-size: is independent of both $\epsilon$ and $K$, and (iv) implicit step-size: has implicit parameters such as smoothness and optimal objective.

[3] **Convergence Rate**: number of iterations for finding an $\epsilon$-approximate stationary point for non-convex optimizations or an $\epsilon$-approximate optimum for convex optimizations.

[4] The radial dual (RD) of a convex constrained problem is an unconstrained min-max problem [Gri24a, Gri24b].

To circumvent these issues, projection-free methods based on the Frank-Wolfe (FW) algorithm [FW+56] have been widely studied (e.g., [MZWG16, THZK21, Mha22, MHSY25]). These methods employ a linear optimization oracle (LOO) at each iteration instead of projections, with the former often being performed efficiently [CP21]. However, LOO can still be computationally expensive over complex constrained sets; therefore, FW methods are confined to scenarios where the LOO is efficient. Moreover, they exhibit oscillatory behavior near the solution, resulting in slow convergence [BRZ24, FG16]. Beyond FW methods, penalty-based approaches [Ber76, SCB+97, LMX22] struggle with ill-conditioning as penalty parameters increase and perform poorly, particularly when dealing with complex constraints. Recent advances explore projection-free strategies leveraging techniques such as *reparameterization* [LMY23, TT24, CV25] and *radial dual* reformulation [Gri24a, Gri24b]. These methods highlight recent efforts to address the limitations of classical projection-free methods, but remain restricted to structured constraint sets and may suffer from practical drawbacks, such as impractical step-size choices (see Table 1). Please refer to Appendix A for a detailed discussion of related work to reduce the per-iteration cost and accelerate convergence for solving convex-constrained optimization. Despite the success of existing projection-free methods, the research gap still remains:

*Can we design a projection-free approach for optimization over a general compact convex set with desirable properties, including fast convergence and cheap per-iteration cost?*

In this paper, we propose a novel *projection-free* framework that positively answers this question. Concretely, we make the following contributions:

▷ In Sec. 3, we design the novel projection-free method, termed as **Hom-PGD**, that re-parameterizes the optimization over convex compact sets to equivalent ball-constrained optimization. By solving the equivalent problem with gradient descent with closed-form projection and mapping the converged solution back, we obtain the solution to the original constrained problem.

▷ In Sec. 4, we establish convergence and complexity analysis for **Hom-PGD**: $\mathcal{O}(1/\epsilon)$ in the convex setting, $\mathcal{O}(\log 1/\epsilon)$ in the strongly convex setting, and $\mathcal{O}(1/\epsilon^2)$ in the non-convex setting, where established convergence rates are *optimal* [2] under unaccelerated settings. Moreover, the per-iteration complexity of Hom-PGD is cheap as $\mathcal{O}(n^2)$ without linear/quadratic optimization oracles. We also extend our framework to optimization over certain non-convex sets in Sec. 5.

---

[2]Optimal convergence rate means it matches the lower bound on iteration complexity. (i) The optimal convergence is $\mathcal{O}(\epsilon^{-2})$ in the non-convex smooth setting [CDHS20]; (ii) For constant step-size, the optimal convergence rate for GD is $\mathcal{O}(\log 1/\epsilon)$ in the strongly convex smooth setting and $\mathcal{O}(1/\epsilon)$ in the convex smooth setting; see e.g. [AP23].

▷ In Sec. 6, through extensive numerical experiments over convex and non-convex problems, including applications to max-cut SDP problems, we demonstrate **Hom-PGD** outperforms existing first-order approaches in computational efficiency, which achieve similar convergence rate but significantly lower per-iteration cost (up to 3-5 orders of magnitude).

To the best of our knowledge, the proposed **Hom-PGD** is the *first* projection-free, first-order framework capable of solving optimization over general convex compact sets while achieving the *optimal* convergence rate under unaccelerated settings without expensive optimization oracles.

## 2 Problem Statement

We consider the following continuous convex constrained optimization problem:

$$\min_{\mathbf{x}} \; f(\mathbf{x}), \quad \text{s.t.} \; \mathbf{x} \in \mathcal{K}, \tag{P}$$

where $\mathbf{x} \in \mathbb{R}^n$ is the decision variable, $f(\cdot)$ is the objective function, and the constraint set $\mathcal{K} \subset \mathbb{R}^n$ is compact and convex. For ease of analysis and without loss of generality, we assume the constraint set $\mathcal{K}$ is defined by inequalities[3] as $\mathcal{K} = \{\mathbf{x} \in \mathbb{R}^n \mid \mathbf{g}(\mathbf{x}) \leq \mathbf{0}\}$ with $\mathbf{g} = (g_1, \cdots, g_m)$, where $g_i : \mathbb{R}^n \to \mathbb{R}$ are functions.

**Open Issues:** Although convex optimization has been extensively studied, existing methods face significant limitations when dealing with complex constraints. As discussed in Sec. 1, projection-based approaches incur high computational costs beyond simple sets, while projection-free methods such as FW and primal-dual approaches are largely restricted to structured convex sets. For example, when solving semidefinite programming (SDP), the LOO required by FW involves solving an SDP itself, defeating its purpose as a low-cost alternative. These challenges underscore the need for projection-free algorithms that preserve fast convergence and maintain computational efficiency across broader convex programs.

## 3 Homeomorphic Optimization Approach

Motivated by recent advances in low-complexity schemes and reparameterization techniques for solving constrained optimization problems [LCL23, LMY23, PP23, LCL24, RSW24, LC25], we propose a novel approach that transforms the original constrained problem via a homeomorphic mapping between the convex constraint set $\mathcal{K}$ and the unit ball $\mathcal{B}$. This transformation preserves the essential problem structure while replacing the potentially complex constraint set with the geometrically simple unit ball, thereby enabling efficient gradient-based optimization without expensive projection operations.

**Definition 3.1** (Homeomorphic constrained optimization). Given a homeomorphism $\psi : \mathcal{B} \to \mathcal{K}$, we define the transformed optimization problem with objective function $h(\mathbf{z}) = (f \circ \psi)(\mathbf{z})$ and constraint set $\mathcal{B} = \psi^{-1}(\mathcal{K})$ as:

$$\min_{\mathbf{z}} \; h(\mathbf{z}), \quad \text{s.t.} \; \mathbf{z} \in \mathcal{B} \tag{H}$$

Homeomorphism (or homeomorphic mapping) is a bi-continuous bijection from two topological spaces, guaranteeing the topological equivalence. It is a classic result that any compact convex set is homeomorphic to a unit ball [Ges12, Bre13], i.e., there exists a homeomorphism $\psi$ such that $\mathcal{B} = \psi^{-1}(\mathcal{K})$ and $\mathcal{K} = \psi(\mathcal{B})$. Thus, we can transform any optimization problem **P** over a compact convex set into a ball-constrained program **H**. In practice, this transformation relies on an explicit homeomorphic mapping, which we will discuss how to obtain in Sec. 3.2.

**Remark**: The transformed problem **H** has a *non-convex* objective function $h(\cdot)$ due to the non-linear mapping $\psi$ even though the original objective is convex, but it features a simple constraint set as a unit ball, leading to a closed-form projection. Moreover, under the homeomorphic transformation, the original problem and its homeomorphic counterpart are equivalent, i.e., there exists a bijective correspondence between their optimal solution sets $\mathbf{P}^*$ and $\mathbf{H}^*$, where $\mathbf{P}^* = \{\mathbf{x} \mid \mathbf{x} \in \arg\min\{\mathbf{P}\}\}$ and similarly for $\mathbf{H}^*$. Specifically, for any $\mathbf{x} \in \mathbf{P}^*$, there exists a unique $\mathbf{z} \in \mathbf{H}^*$ such that $\mathbf{x} = \psi(\mathbf{z})$, and vice versa. Thus, we can solve the transformed problem **H** without expensive projection to obtain the corresponding optimal solution of the original problem **P**.

---

[3]Linear equality constraints can be removed without loss of generality, see Appendix B.1 for discussions.

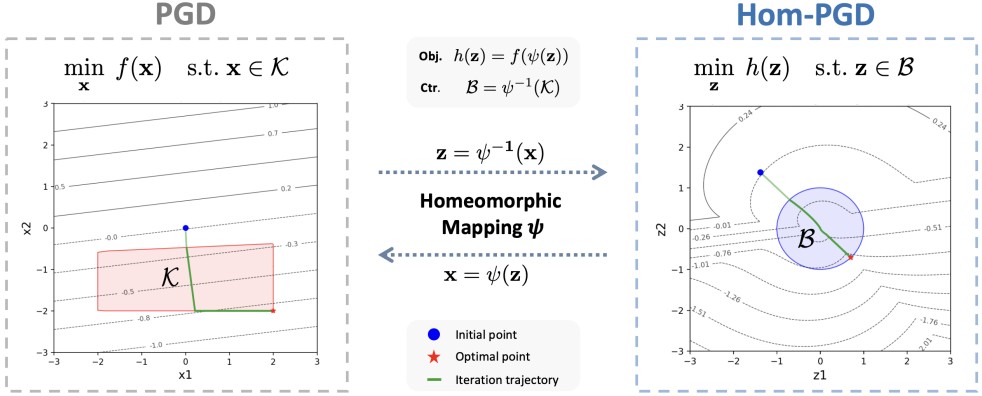

Figure 1: **Hom-PGD Framework**: Hom-PGD conducts the standard PGD algorithm in the transformed space by a homeomorphic mapping $\psi(\cdot)$, where the transformed constraint set $\mathcal{B}$ is a simple unit ball, and the transformed objective function $h(\cdot)$ is non-convex but structured.

## 3.1 Algorithm Overview

As illustrated in Fig. 1, PGD in the original space suffers from expensive projection operations during iteration over the constraint boundary. To solve problem **P** without expensive projection, we transform problem **P** into a ball-constrained optimization **H** by a homeomorphic mapping $\psi$. Then we apply regular PGD to efficiently solve the homeomorphic optimization **H** with a closed-form projection, thereby termed *projection-free*[4] methods. Finally, we map the obtained solution back to the original space to recover the corresponding solution for the original problem.

---

**Algorithm 1** Hom-PGD

**Input:** initial point $\mathbf{z}_0$, problem **H** with $\psi$ and maximum iteration number $K$
**for** $k = 0$ **to** $K$ **do**
    Compute stepsize $\alpha_k$
    **Update:** $\mathbf{z}_{k+1} = \Pi_{\mathcal{B}}\left(\mathbf{z}_k - \alpha_k \nabla h(\mathbf{z}_k)\right)$
**end for**
**Output:** $\mathbf{x}_K = \psi(\mathbf{z}_K)$

---

We call the combination of homeomorphic transformation and regular PGD as **Hom-PGD**, shown in Alg. 1. Next, we discuss how to construct an explicit $\psi$ for a general compact convex set $\mathcal{K}$.

## 3.2 Construction of Homeomorphism

We introduce an explicit-form homeomorphic mapping between convex set $\mathcal{K}$ and unit ball $\mathcal{B}$ as follows, termed Gauge mapping [TZ22, LLC25]:

**Definition 3.2** (Gauge mapping). Let $\gamma_{\mathcal{K}}(\mathbf{x}, \mathbf{x}^\circ) = \inf\{\lambda \geq 0 \mid \mathbf{x} \in \lambda(\mathcal{K} - \mathbf{x}^\circ)\}$ be the Gauge/Minkowski function [BM08] given an interior point $\mathbf{x}^\circ \in \text{int}(\mathcal{K})$. The gauge mapping $\psi : \mathcal{B} \to \mathcal{K}$ is defined between a unit ball and a compact convex set:

$$\psi(\mathbf{z}) = \frac{\|\mathbf{z}\|}{\gamma_{\mathcal{K}}(\mathbf{z}, \mathbf{x}^\circ)}\mathbf{z} + \mathbf{x}^\circ, \ \forall \mathbf{z} \in \mathcal{B}; \qquad \psi^{-1}(\mathbf{x}) = \frac{\gamma_{\mathcal{K}}(\mathbf{x} - \mathbf{x}^\circ, \mathbf{x}^\circ)}{\|\mathbf{x} - \mathbf{x}^\circ\|}(\mathbf{x} - \mathbf{x}^\circ), \ \forall \mathbf{x} \in \mathcal{K}. \quad (1)$$

First, gauge mapping $\psi$ establishes a homeomorphism between any compact convex set and the unit ball, ensuring that $\mathcal{K} = \psi(\mathcal{B})$ and $\mathcal{B} = \psi^{-1}(\mathcal{K})$. Intuitively, this mapping transforms the unit ball by first translating it to align with an interior point of the convex set, then scaling points radially outward from this interior point until the ball's boundary conforms to the convex set's boundary (illustrated in Fig. 2). Moreover, the gauge mapping has *closed-form* expressions for common convex sets (linear, quadratic, second-order cone, and linear matrix inequality

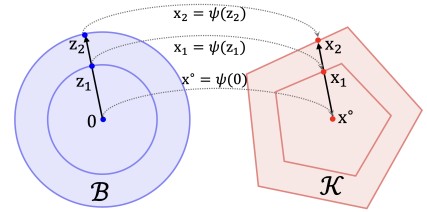

Figure 2: Gauge mapping illustration.

constraints) and can be efficiently computed via *bisection* methods for general convex constraints. For a comprehensive property and computation of gauge mapping, we refer readers to Appendix B.

---

[4]The term "projection-free" specifically refers to avoiding expensive projections onto original complex constraint sets, a usage that aligns with standard conventions in the literature [LMY23, LBGH23].

Leveraging this explicit homeomorphic gauge mapping $\psi$, we transform problem $\mathbf{P}$ to problem $\mathbf{H}$, and apply Hom-PGD (Alg. 1) to solve it without expensive projection. However, the gauge mapping $\psi$ depends on the choice of interior point $\mathbf{x}^\circ$ (shown in Def. 3.2). Different choices of $\mathbf{x}^\circ$ yield distinct gauge mappings that alter the landscape of the transformed problem $\mathbf{H}$ and affect the convergence behavior of Hom-PGD. As we will establish rigorously in Sec. 4, gauge mappings with smaller Lipschitz constants favor faster convergence of Hom-PGD. Thus, we proceed to analyze the Lipschitz properties of the gauge mapping as:

**Proposition 3.3** (Bi-Lipschitz constants of the gauge mapping). *Let $\mathcal{K} \subset \mathbb{R}^n$ be a compact convex set and let $\mathbf{x}^\circ \in \mathrm{int}(\mathcal{K})$ be an interior point. Define the inner and outer radii with respect to $\mathbf{x}^\circ$ as*

$$r_i := \sup\{r \geq 0 : \mathcal{B}(\mathbf{x}^\circ, r) \subseteq \mathcal{K}\}, \quad r_o := \inf\{r \geq 0 : \mathcal{K} \subseteq \mathcal{B}(\mathbf{x}^\circ, r)\},$$

*such that $\mathcal{B}(\mathbf{x}^\circ, r_i) \subseteq \mathcal{K} \subseteq \mathcal{B}(\mathbf{x}^\circ, r_o)$. Then the Lipschitz constant (denoted as $\mathrm{L}(\cdot)$) of gauge mapping $\psi$ associated with $\mathcal{K}$ satisfies the following bounds:*

*Forward Lipschitz:* $\quad \kappa_2 := \mathrm{L}(\psi) \leq 2\, r_o + r_o^2/r_i, \quad$ *Inverse Lipschitz:* $\quad \dfrac{1}{\kappa_1} := \mathrm{L}(\psi^{-1}) \leq 2/r_i.$

Therefore, to reduce the Lipschitz constant of the gauge mapping and boost the convergence of Hom-PGD, we can select a "central" interior point with large inner radius $r_i$ or small outer radius $r_o$. In practice, we may solve a convex problem by minimizing the constraint residual to find a "central" interior point approximately following [THH23] (refer to Appendix B for details).

# 4 Performance Analysis

In this section, we present a comprehensive performance analysis for Hom-PGD, including the landscape analysis, convergence rate, and run-time complexity.

**General Assumptions** (with details in Appendix C.2): The objective $f$ and constrained functions $g_i$ in $\mathbf{P}$ are continuously differentiable and smooth. The homeomorphic mapping $\psi$ is invertible with a non-singular Jacobian matrix and is $(\kappa_1, \kappa_2)$-bi-Lipschitz continuous. Additionally, the Jacobian matrix of $\psi$ (denoted as $\mathrm{J}_\psi$) exists and is Lipschitz continuous.

We remark that our theoretical results hold for *any* homeomorphism satisfying these assumptions, and we construct a specific homeomorphism, the gauge mapping, in practice. Moreover, gauge mapping meets these assumptions (with details in Appendix B.3).

## 4.1 Landscape Analysis

First, under general assumptions, the composite function $h = f \circ \psi$ in problem $\mathbf{H}$ inherits favorable properties as follows (more properties of $h$ are included Lemma D.1).

**Lemma 4.1.** *Denote $L_{h,0}, L_h, \kappa_2, L_\psi$ as the Lipschitz constant of $f, \nabla f, \psi, \mathrm{J}_\psi$ respectively. Then $h$ is $L_{h,0} := L_{f,0}\kappa_2$ Lipschitz continuous and $h$ is $L_h$-smooth with $L_h = \kappa_2^2 L_f + L_\psi L_{f,0}$.*

Next, recall that a point $\mathbf{x}^*$ is said to be a stationary point of problem $\min_{\mathbf{x} \in \mathcal{K}} f(\mathbf{x})$ with convex set $\mathcal{K}$, if $\nabla f(\mathbf{x}^*)^\top (\mathbf{x} - \mathbf{x}^*) \geq 0$ for any $\mathbf{x} \in \mathcal{K}$. It is well-known that any stationary point is a global optimum for a convex constrained optimization problem. A natural question arises: *does this property also hold for the non-convex optimization problem $\mathbf{H}$ under convex problem $\mathbf{P}$?* For unconstrained cases, this property does hold, as the function $h$ is invex with the property that every stationary point of an invex function is a global optimum [Mar85]. For the constrained case, we provide the following formal statement where the proof is deferred to Appendix D.6.

**Proposition 4.2** (Global Optimality of $\mathbf{H}$). *Suppose problem $\mathbf{P}$ is a convex optimization. If $\mathbf{z}^*$ is a stationary point of problem $\mathbf{H}$ and LICQ (Def. D.3) holds at $\mathbf{z}^*$, then $\mathbf{x}^* = \psi(\mathbf{z}^*)$ is a stationary point of problem $\mathbf{P}$ (thus a global optimum). Hence, $\mathbf{z}^*$ is a global optimum.*

We remark that the LICQ assumption is mild in our setting. For problem ($\mathbf{H}$), the only constraint is $\|\mathbf{z}\|^2 \leq 1$, so LICQ holds at any boundary point where the constraint is active. Moreover, since there are no equality constraints, LICQ trivially holds at any interior point. Moreover, we derive that there is a one-to-one correspondence for KKT points and non-degenerate stationary points between $\mathbf{P}$ and $\mathbf{H}$. The relevant definitions, formal statements, and proofs are provided in Appendix D.5.

## 4.2 Convergence Analysis

In this section, we provide a theoretical convergence analysis of Hom-PGD (Alg. 1) for solving problem **H**. Our main result demonstrates that: *Hom-PGD achieves the same convergence rate as the standard PGD for the (non-)convex problem* **P** *under mild regularity conditions, despite operating on the non-convex formulation* **H**.

Before moving on, we recall some basic definitions. An $\epsilon$-stationary point $\mathbf{x}^*$ for problem **P** is defined by $\|G(\mathbf{x}^*)\| \leq \epsilon$ where the *gradient mapping* $G(\mathbf{x}) := G_{1/\alpha}(\mathbf{x}) = \frac{1}{\alpha}[\mathbf{x} - \Pi_{\mathcal{K}}(\mathbf{x} - \alpha \nabla f(\mathbf{x}))]$. Note that in the definition, we omit the dependence of the function $G$ on $\mathcal{K}$ and $\alpha$.

### 4.2.1 (Strongly) Convex Objective $f$

The following theorem provides the convergence analysis of Hom-PGD for convex optimization **P**.

**Theorem 1.** *Suppose the strict complementary slackness condition (Def. D.4) holds for both problem* **P** *and* **H**, *and the problem* **P** *is convex with a non-degenerate[5] minimizer* $\mathbf{x}^*$. *Let* $\{\mathbf{z}_k\}$ *be the sequence generated by Hom-PGD with step-size* $\alpha \in (0, \frac{2}{L_h}]$. *For sufficient small* $\epsilon > 0$, $\{\mathbf{z}_k\}_{k=1}^K$ *with* $K = \mathcal{O}(L_h/\epsilon)$ *contains* $\mathbf{z}'$ *such that* $h(\mathbf{z}') - h^\star \leq \epsilon$.

*Proof Intuition.* Under the invertible mapping, $\mathbf{z}^* = \psi(\mathbf{x}^*)$ is also a non-degenerate point from Lemma D.10. Consequently, it satisfies local strong convexity or local PL condition, meaning that $h(\mathbf{z}) - h^* \leq \mathcal{O}(\|G(\mathbf{z})\|^2)$ holds within a sufficiently small ball centered at $\mathbf{z}^*$. As a result, it follows from Theorem 3 that PGD only requires $\mathcal{O}(1/\epsilon)$ complexity to find a $\mathcal{O}(\sqrt{\epsilon})$-stationary point $\mathbf{z}'$ such that $h(\mathbf{z}') - h(\mathbf{z}^*) \leq \mathcal{O}(\epsilon)$. The idea is motivated by [LMY23], which applies specific Hadamard parameterization to transform a simplex-constrained optimization to a sphere-constrained optimization. We extend their results to a general homeomorphic mapping $\psi$, and proof details can be found in Appendix E.1.

If the objective in **P** is strongly convex, we will have a faster (i.e., linear) convergence rate as:

**Theorem 2.** *Suppose problem* **P** *is convex with* $\mu_f$-*strongly convex objective* $f$. *Let* $\{\mathbf{z}_k\}_{k\geq 0}$ *be generated by Hom-PGD with proper constant step-size* $\alpha \in (0, \frac{2}{L_h}]$. *With* $K = \mathcal{O}(\kappa \log 1/\epsilon)$ *where* $\kappa = L_h/(\mu_f \kappa_1)$, *we have* $h(\mathbf{z}_K) - h(\mathbf{z}^*) \leq \epsilon$, *and* $\|\mathbf{z}_K - \mathbf{z}^*\| \leq \epsilon$.

*Proof Intuition.* If $f$ is a strongly convex function over a convex set, it satisfies a generalized PL condition. The homeomorphic mapping preserves this generalized PL condition such that the linear convergence can be established for **H**. Details can be found in Appendix E.2.

**Remark.** (i) The convergence rates derived in Theorem 1 and 2 match the lower bounds in the unaccelerated convex and strongly convex settings (referring to e.g., [AP23]). (ii) Theorem 1 and Theorem 2 demonstrate that the Hom-PGD algorithm not only maintains projection-free properties over the unit ball, reducing per-iteration computational complexity; but also achieves the same convergence rates as standard PGD when applied to the original convex optimization problem **P**, which is non-trivial since Hom-PGD operates on the non-convex objective in problem **H**.

### 4.2.2 Non-convex Objective $f$

For a non-convex objective $f$, a classical result (e.g., Theorem 9.15 [Bec14]) can be leveraged to show that Hom-PGD algorithms can converge to an $\epsilon$-stationary point with $\mathcal{O}(1/\epsilon^2)$ iterations.

**Theorem 3.** *Consider a problem* $\min_{\mathbf{z} \in \mathcal{Z}} h(\mathbf{z})$ *with a convex set* $\mathcal{Z}$. *Suppose* $h$ *is non-convex and differentiable with* $L_h$-*Lipschitz continuous gradient. Then the sequence* $\{\mathbf{z}_k\}_{k=0}^K$ *with* $K = \mathcal{O}(L_h/\epsilon^2)$ *generated by Hom-PGD algorithm with a constant step-size* $\alpha \in (0, \frac{2}{L_h}]$ *contains an* $\epsilon$-*stationary point* $\mathbf{z}'$, *i.e,* $\|G(\mathbf{z}')\| \leq \epsilon$ *for some* $\mathbf{z}' \in \{\mathbf{z}_k\}_{k=0}^K$.

**Remark.** (i) The convergence rate in Theorem 3 matches the optimal rate for smooth non-convex optimization problems [CDHS20]. (ii) While Theorem 3 also applies to (strongly) convex objectives, it yields slower convergence than standard PGD in these cases, as it fails to exploit the convex structure of problem **P** and the hidden convexity of problem **H** [FHH23]. This highlights the significance of

---

[5]A minimizer of an optimization problem is non-degenerate if the Hessian of the Lagrangian function is positive definite in the critical cone of the minimizer. See Appendix D.4 for details.

our results in Theorems 1 and 2, where Hom-PGD achieves the same convergence rates of standard PGD while avoiding expensive projection in the transformed domain. (iii) For non-convex objectives satisfying regularity conditions such as the *KL property* or *error bound conditions*, linear convergence rates as in Theorem 2 can also be achieved. See Remark E.5 for further discussion.

In these sections, we establish convergence results for Hom-PGD across different problem classes (Theorems 1, 2, and 3). The convergence rates depend on the Lipschitz constants of the constructed gauge mapping, specifically: (i) the forward Lipschitz constant $\kappa_2$, which relates to the parameter $L_h$ established in Lemma 4.1 and appears in Theorems 1 and 3; and (ii) the inverse Lipschitz constant $1/\kappa_1$, which appears in Theorem 2. As demonstrated in Prop. 3.3, the choice of interior point directly influences the Lipschitz constants of the gauge mapping $\psi$. Consequently, different interior points modify the Hom-PGD convergence rate by constant factors while preserving the fundamental convergence order.

Additionally, our theoretical analysis assumes access to exact gradients and gauge mappings, which is consistent with standard practice in the optimization literature (e.g., [LBGH23]). However, for general convex sets, the gauge mapping is numerically approximated using the bisection algorithm (Alg. 2) to compute both its value and gradient within a specified error tolerance $\delta$ at each iteration. This numerical approximation introduces an additional $\mathcal{O}(\delta)$ term in the optimality gap of convergence results [DGN14]. Since $\delta$ can be chosen arbitrarily small, this additional error term remains negligible and does not affect the fundamental convergence guarantees of our algorithm.

## 4.3 Complexity Analysis

In this subsection, we analyze the total run-time complexity of Hom-PGD, including initialization complexity, per-iteration complexity, and last-step complexity.

**Oracles.** We list specific oracles in Hom-PGD besides general ones (e.g., zeroth-order oracle for explicit function evaluation).

- *Membership oracle*: Given $\mathbf{x} \in \mathbb{R}^n$, this oracle $\mathcal{M}_\mathcal{K}(\mathbf{x}) := \mathbb{I}(\mathbf{x} \in \mathcal{K}) : \mathbb{R}^n \to \{0, 1\}$ returns 1 if and only if $\mathbf{x} \in \mathcal{K}$. This oracle performs only feasibility checking without requiring the solution of optimization subproblems. For common convex sets including polyhedra and second-order cones, the membership oracle can be implemented with computational complexity not exceeding $\mathcal{O}(n^2)$ [Mha22], with significantly lower computational burden than LOO or projection in practice.

- *Interior point oracle*: This oracle returns an interior point of $\mathcal{K}$ by solving a convex feasibility problem. This requirement, common in projection-free frameworks [Mha22, Gri24a, Gri24b], can be addressed using first-order methods with $\tilde{\mathcal{O}}(n^2)$[6] complexity or interior-point methods with $\tilde{\mathcal{O}}(n^{3.5})$ complexity. Notably, this oracle is only called **once** for the entire optimization algorithm.

**Basic operations in Hom-PGD** (with details in Appendix B.4). Hom-PGD requires computing the gradient of $h = f \circ \psi$ per iteration and transforming the final solution via the gauge mapping.

- *Computing gauge mapping $\psi$*: $\tilde{\mathcal{O}}(n)$. To compute the gauge mapping in the general case, one may evaluate the gauge function $\gamma_\mathcal{K}(\cdot, \mathbf{x}^\circ)$ to an accuracy $\epsilon$ with $\mathcal{O}(\log 1/\epsilon)$ membership oracle calls, plus $\mathcal{O}(n)$ operations for the scalar-vector product in Def. 3.2.

- *Computing gradient of $h$*: $\tilde{\mathcal{O}}(n^2)$. Numerical differentiation techniques (finite/automatic differentiation [BF97, BPRS18, LBGH23]) can be applied, e.g., computing each component $\nabla_i h$ ($i = 1, 2, \cdots, n$) requires $\mathcal{O}(1)$ zeroth-order oracle calls of $f$ and $\tilde{\mathcal{O}}(n)$ cost for evaluating $\psi$, yielding a total complexity of $\tilde{\mathcal{O}}(n^2)$.

**Total run-time complexity of Hom-PGD**.

- *Initialization complexity* (IC): One interior point oracle call to obtain an interior point of $\mathcal{K}$.

- *Per-iteration complexity* (PiC): Each iteration cost $\tilde{\mathcal{O}}(n^2)$, comprising gradient computation $\nabla h(\mathbf{z})$ at $\tilde{\mathcal{O}}(n^2)$ and unit ball projection at $\mathcal{O}(n)$.

- *Last-step complexity* (LsC): It costs $\tilde{\mathcal{O}}(n)$ for computing $\psi$ to map the converged solution $\mathbf{z}^*$ back to the original space via $\mathbf{x}^* = \psi(\mathbf{z}^*)$.

- *Number of iterations* (#I): Convergence rate varies from different settings, referring to Sec. 4.2.

---

[6]Here $\tilde{\mathcal{O}}(\cdot)$ hides polynomial logarithmic factors.

In conclusion, the total complexity of Hom-PGD equals to $\text{IC} + \#\text{I} \cdot \text{Pic} + \text{LsC} = \mathcal{O}(n^2 \cdot \#\text{I})$. This complexity is significantly lower than that of second-order methods, which typically incur $\mathcal{O}(n^3)$ per-iteration cost, thereby highlighting the scalability of Hom-PGD to high-dimensional problems. Moreover, our method achieves an optimal convergence rate under the first-order setting, ensuring both efficiency and theoretical soundness.

## 5  Discussions

### 5.1  Beyond Vanilla Gradient Methods

**Mirror gradient methods**: Mirror descent methods use a mirror map to transform points into a dual space, where optimization steps are performed. Essentially, these methods can be viewed as generalized projection methods [BT03]. For example, with a quadratic mirror map, it becomes standard PGD; with a negative entropy mirror map, it recovers exponential gradient descent for simplex-constrained problems. However, for general convex constraints, mirror descent often lacks explicit mirror maps and suffers from high projection complexity, despite having convergence rates comparable to ours [Bec17].

**Advanced gradient methods.** One may note that we can replace the unaccelerated gradient methods with any advanced optimizers, such as momentum-based or Nesterov-accelerated methods, or the Adam optimizer popular in deep learning [KB14]. Despite their potential for accelerating optimization of the non-convex landscape in problem **H**, analyzing the convergence behavior by utilizing the hidden convexity structure remains challenging, which opens directions for future research. We also conduct illustrative experiments comparing vanilla gradient descent and Adam in Section 6.4.

**Second-order methods.** We may also consider second-order methods with projection to optimize problem **H**. However, significant challenges arise from both computational and theoretical perspectives. From a computational standpoint, evaluating the Hessian of the composite objective $h = f \circ \psi$ requires computing a third-order tensor of the gauge mapping, which is both computationally expensive and memory intensive. From a convergence perspective, analyzing convergence over the non-convex landscape to a global optimum becomes substantially more difficult.

### 5.2  Extension to Non-Convex Constraints

**Star-shaped set.** Our framework can naturally extend to optimization over non-convex sets where an explicit homeomorphism $\psi$ exists such that $\psi(\mathcal{B}) = \mathcal{K}$. For star-shaped sets, all points are visible from a star center $\mathbf{x}^\circ$ [Lee10], allowing the construction of a gauge mapping that bijectively maps a standard ball to the set. Such star-shaped constrained problems arise in machine learning tasks such as $\ell_p$-constrained adversarial attacks in neural networks [EBMA21]. Applying Hom-PGD to such problems maintains the $\mathcal{O}(1/\epsilon^2)$ convergence rate for finding an $\epsilon$-stationary point per Theorem 3.

**Ball-homeomorphic set.** Our framework may also extend to non-convex sets that are homeomorphic to a unit ball. However, determining if a non-convex set is ball-homeomorphic requires examining its topological properties, which is challenging. While a homeomorphism exists for such sets, there is generally no explicit form for the homeomorphism. Learning-based methods for approximating such homeomorphisms [LCL23, LCL24] present promising directions for future research.

**General non-convex set.** For more general non-convex sets that may not be ball-homeomorphic (e.g., disconnected sets), our framework remains applicable but without optimality guarantees. Given an interior point $\mathbf{x}^\circ$ in $\mathcal{K}$, we can define $\mathcal{X}$ as the largest contained star-shaped set with center $\mathbf{x}^\circ$. Similar to constraint restriction methods [LNDT19], the optimality gap depends on the Hausdorff distance between $\mathcal{K}$ and $\mathcal{X}$.

## 6  Empirical Study

In this section, we conduct simulations to demonstrate the effectiveness of Hom-PGD on both convex and non-convex constrained optimization problems. The detailed problem formulations, experimental settings, algorithm hyperparameters, and supplementary experiment results are in Appendix F.

**Baselines**: (i) **PGD**: Regular projected gradient descent applied to problem **P**, where the projection operation is called at each iteration. (ii) **FW**: Frank-Wolfe methods, which solve an update direction with a linearized objective and update the decision variables. (iii) **ALM**: Augmented Lagrangian

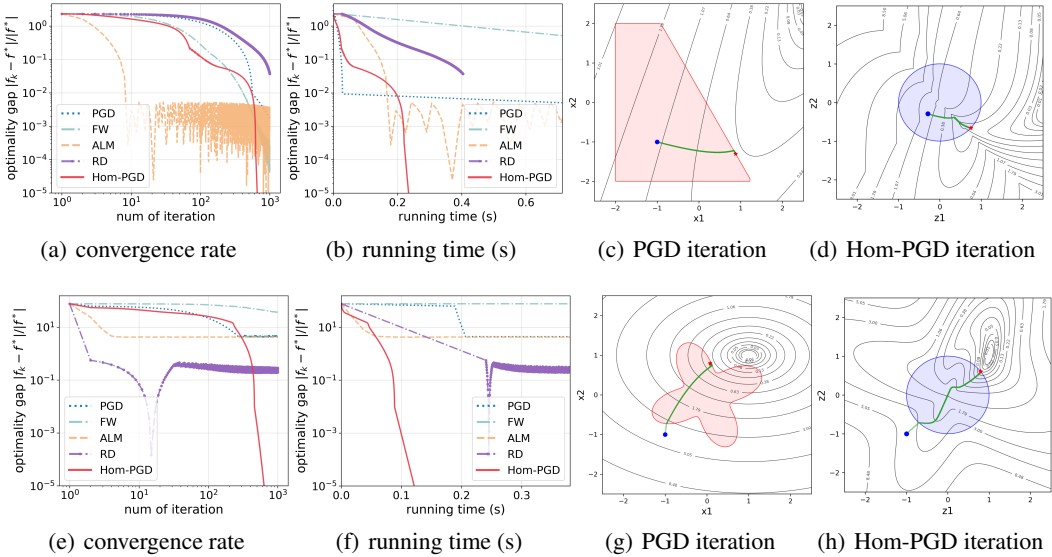

Figure 3: Performance for optimization over **polyhedron** (**a-d**) and **star-shaped** set (**e-h**), respectively. All methods are executed with the same initial points and terminated by a maximum number of iterations ($10^3$). Complete results are in Appendix G.1 (Fig. 8 - 11).

methods for problem **P** that alternately update primal and dual coefficients for the unconstrained formulation to problem **P**. (iv) **RD**: Radial-Dual framework, which applies radial-dual to formulate the constrained problem into unconstrained min-max optimization. (v) **Hom-PGD**: Projected gradient descent applied to the transformed problem **H** shown in Sec. 3.

### 6.1 Illustrative Examples: Optimization over Polyhedron and Star-shaped Set

We examine a two-dimensional illustrative optimization problem involving quadratic optimization over both a (*convex*) polyhedron and a (*non-convex*) star-shaped set to demonstrate our method's efficiency. As shown in Fig. 3, Hom-PGD outperforms other first-order algorithms in both settings. The iteration trajectories in the transformed space reveal the mechanism behind this efficiency: Hom-PGD avoids complex projections while effectively performing gradient descent in a structured landscape of problem **H** to the optimum, even though it is non-convex.

### 6.2 Solving Second-order Cone Programming (SOCP)

We next evaluate the performance of algorithms on SOCP, which encompasses fundamental convex programs (LP, QP, convex QCQP) and has widespread applications in portfolio optimization [BBV04] and optimal power flow problems [Low14a]. Problem instances are randomly generated following the CVXPY documents. As shown in Fig. 4, our method not only converges rapidly to the target error tolerance but also demonstrates significantly lower per-iteration costs (up to 3-5 orders of magnitude) compared to projection-based or Frank-Wolfe methods, since Hom-PGD does not need complex optimization oracles such as projection or LOO during iterations. Further, we also use a commercial solver, **MOSEK**, which typically applies primal-dual interior point methods to solve convex programs. Notably, the solver costs 5424 seconds to solve the 1000-dim instance, while Hom-PGD takes less than 600 seconds to reach a $10^{-3}$ objective optimality gap.

### 6.3 Solving Max-Cut Semi-Definite Programming (SDP)

We further evaluate our method on the more challenging max-cut SDP problem. While max-cut is an NP-hard combinatorial problem, SDP relaxation with randomized rounding achieves an expected approximation ratio of 0.878 [GW95]. We generate random *Erdős-Rényi* graphs as test instances [HSS08]. Since the optimum of the max-cut SDP is typically low-rank, the gauge mapping encounters non-differentiability at these solutions. To address this practical issue, we apply the smoothing techniques described in Appendix B.3.2. As shown in Figure 5, our approach demonstrates efficient optimization even in high-dimensional decision spaces ($50^2$ variables) with positive semi-definite cone constraints. Notably, Hom-PGD exhibits a *slower* convergence rate on SDP with linear objectives

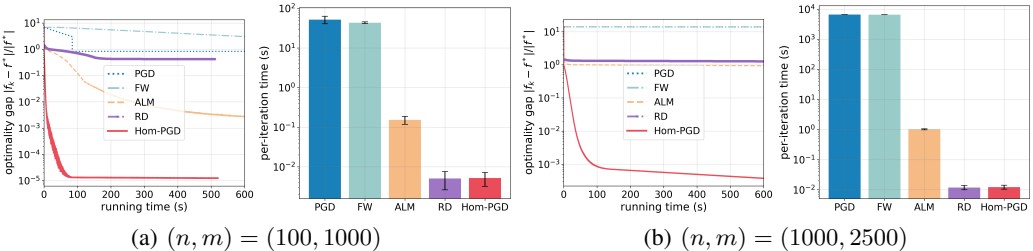

(a) $(n, m) = (100, 1000)$  (b) $(n, m) = (1000, 2500)$

Figure 4: **Performance over SOCP**: $n$ is the number of decision variables and $m$ is the number of constraints. All methods are executed with the same initial points and terminated by a maximum iterations ($10^5$) or running time (600 seconds). Complete results are in Appendix G.2 (Fig. 12-14).

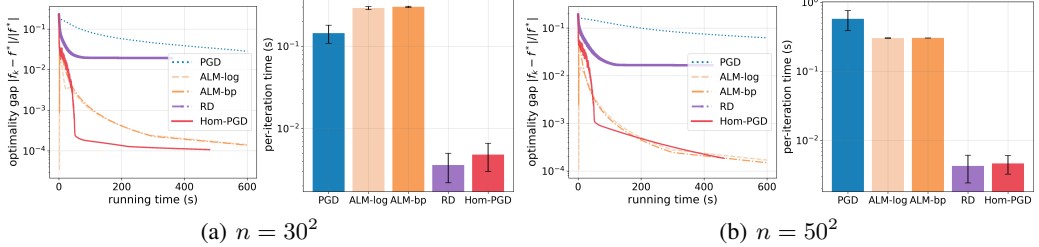

(a) $n = 30^2$  (b) $n = 50^2$

Figure 5: **Performance over SDP**: there are $n = \mathcal{O}(N^2)$ decision variables where $N$ is the graph size. All methods are executed with the same initial points and terminated by a maximum of iterations ($10^5$) or running time (600 seconds). Complete results are in Appendix G.3 (Fig. 15-19).

compared to SOCP with quadratic objectives. This behavior aligns with our theoretical analysis in Theorems 1 and 2, where strongly convex (quadratic) objectives yield faster convergence rates than convex (linear) objectives. The ALM methods solve the Burer-Monteiro SDP formulation [BM03] with log-rank ($\log N$) and Barvinok-Pataki (bp)-rank ($\sqrt{2N}$) [Bar95, Pat98, BVB20]. Despite the scalability of this low-rank formulation, it incurs violation on additional equality constraint and high iteration cost for each inner minimization. However, the per-iteration complexity of our method still outperforms other approaches, resulting in comparable convergence in terms of total running time.

### 6.4 Scalability Tests and Ablation Study

We first evaluate the scalability of gauge mapping computation across various constraints and dimensions in Fig. 6, demonstrating efficiency (less than 0.01 seconds) up to 3000-dimensional constraints. Our ablation study further examines critical framework components: (i) interior point selection (Fig. 20), confirming that central points (smaller Lipschitz) accelerate convergence as predicted by our theory analysis in Sec. 4.2; and (ii) gradient method variants (Fig. 21), revealing that advanced optimization techniques (e.g., Adam [KB14]) further enhance performance for solving non-convex problem **H**, suggesting promising directions for future research.

## 7 Conclusion and Limitations

In this work, we propose **Hom-PGD**, a projection-free method that transforms constrained optimization over general convex (and certain non-convex) sets into a ball-constrained problem via a homeomorphism. Hom-PGD achieves optimal convergence rates with $\mathcal{O}(n^2)$ per-iteration complexity without expensive projections or oracles. Numerical results show competitive convergence with significantly lower iteration costs. Despite its efficiency, there are several **limitations** to be addressed in future work: (i) Extending Hom-PGD to more general non-convex sets is non-trivial, as discussed in Sec. 5. (ii) From the convergence theory perspective, while Hom-PGD achieves optimal convergence rates under unaccelerated settings, it remains an open question whether acceleration techniques (e.g., Nesterov-style methods) can be incorporated to attain optimal accelerated rates. The challenge stems from the non-convexity of the transformed problem **H**. (iii) The gauge mapping used in this work serves as a simple and explicit homeomorphism but may not be the optimal choice in terms of conditioning or convergence behavior. Exploring alternative homeomorphisms tailored to specific problem structures could further improve performance.

## Acknowledgments

This work is supported in part by a General Research Fund from Research Grants Council, Hong Kong (Project No. 11214825), a Collaborative Research Fund from Research Grants Council, Hong Kong (Project No. C1049-24G), an InnoHK initiative, The Government of the HKSAR, Laboratory for AI-Powered Financial Technologies, a Shenzhen-Hong Kong-Macau Science & Technology Project (Category C, Project No. SGDX20220530111203026), and a Start-up Research Grant from The Chinese University of Hong Kong, Shenzhen (Project No. UDF01004086). The authors would also like to thank the anonymous reviewers for their helpful comments.

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

# Contents

# A   Related Work

We discuss related work on methods for reducing computational costs and achieving speedups in constrained optimization, organized into three parts: (i) classical projection and projection-free methods, (ii) recent advances, and (iii) work related to ball-constrained optimization.

## A.1   Classical Methods

In convex optimization, three classical approaches have been widely studied: Frank-Wolfe methods [FW⁺56], which avoids projection through linear minimization oracles over constrained sets; primal-dual methods [DFF56], which address primarily linear constraints through simultaneous updates to primal and dual variables; and penalty methods [Ber76], which incorporate constraints into the objective function using penalty functions [SCB⁺97]. Each approach has its limitations.

**Frank-Wolfe methods.** Frank-Wolfe (FW) methods are first-order optimization algorithms that offer several attractive properties: they are easy to implement, projection-free, affine-invariant [Lan13, KLLJS21, Pen23],and their iterates naturally form sparse convex combinations of extreme points in the feasible region, making them particularly valuable for various machine learning applications [ÑFSA14, JTFF14, BZK18, MHK20, THZK21]. Convergence analysis of classic FW methods have been widely studied [LP66, DR70, Dun79, GM86]. However, FW methods face two key limitations: they require an efficient linear minimization oracle [CP21], and they often exhibit slower convergence rates [BRZ24, FG16]. Specifically, under the *Wolfe's lower bound* setting [Wol70], the FW algorithm cannot achieve convergence rates better than $\mathcal{O}(\epsilon^{-1+\delta})$ for any $\delta > 0$. Overcoming this fundamental barrier requires either algorithmic modifications or additional strong assumptions [GH15, GH16, BPTW19, CP20, WKP23, WPP25]. For advanced convergence analysis, one could refer to [Pen23, WPP24, MHSY25].

**Penalty methods.** Penalty methods struggle with ill-conditioning as penalty parameters increase and perform badly, especially for complex constraints. Primal-dual methods are primarily used to handle linear constraints [Koj89, Meh92, CP11], and their second-order variants face scalability issues due to high computational complexity.

**First-order primal-dual methods.** First-order primal-dual methods iteratively update primal and dual variables using inexact gradient steps (see, e.g., [HHZ17, CP11]). However, analyzing the convergence of such methods remains a challenging problem, particularly for nonconvex objectives [ZL22]. Most existing convergence results focus on problems with linear constraints [LT93b, CP11]. A widely used technique to address this challenge is error-bound analysis [LT93a, Pan97], which has been effective in establishing convergence rates for first-order methods in the convex setting [LT92, HL17]. However, these results typically provide only local convergence guarantees—ensuring convergence only when iterates are sufficiently close to the solution set—and depend on an error-bound constant that is often unknown or difficult to estimate. Recent work [ZL20, ZL22, ZPL22] introduces the smooth augmented Lagrangian primal-dual algorithm for constrained optimization. While this method achieves an optimal convergence rate matching the lower bound of optimization complexity for nonconvex objectives, it incurs additional *projection complexity* in each iteration.

## A.2   Recent Advances

To reduce the cost and accelerate the convergence for solving (non-)convex optimization over convex sets, recent novel projection-free methods and other advanced techniques involve inexact projection, radial dual formulation, re-parameterizing optimization problems, and uncovering hidden convexity.

**Inexact projection**. In many cases, the projection operator lacks an analytic solution or is computationally expensive to compute exactly, motivating the analysis of inexact projected methods. For convex optimization, such methods achieve the same convergence rate as PGD if the cumulative projection error is bounded [SRB11, PN18], with new results derived under specific settings [PI21]. For nonconvex objectives with convex constraints, their convergence has been analyzed in [BMR03, WL06, ZWWY20]. Recent advances further generalize inexact projection operators to broader settings [FLP22, AFP23].

**Radial duality.** Beyond classical projection-free methods, recent advancements have introduced novel approaches based on gauge and radial duality theory. Radial duality theory for nonnegative

optimization problems [Gri24a, Gri24b] demonstrates that constrained optimization problems can be reformulated as unconstrained problems using the gauge of their constraints. This framework has led to the development of new families of projection-free methods with optimal convergence guarantees [LG23], as well as relaxed conditions [SG24] that enable more efficient line search operators for the reformulated unconstrained problems.

**Reparameterization.** Reparameterization optimization problems aim to mitigate challenging properties, such as non-smoothness or non-convexity, via invertible transformations while preserving equivalent optima. Parameterization is widely used in optimization and learning tasks, including semi-definite programming [Cif21], low-rank optimization [MMBS14, HLB20], and risk minimization [BRTW22]. Recent advancements include parameterizing simplex [LMY23] and polyhedron [TT24] optimization via Hadamard transformation to reduce projection complexity, smooth over-parameterization to accelerate non-smooth optimization algorithms [PP23], parameterizing discrete data as continuous for generative learning [DKP+24], and analyzing the optimization landscape under parameterization transformations in non-convex settings [LKB24].

**Hidden convexity.** Hidden convexity refers to transformations that reveal the convex structure of non-convex sets or functions, which has been exploited in problems such as rotation matrix optimization [RSW24], non-linear least squares [DP19], revenue management and inventory control [CHHY22], and quadratically constrained quadratic programming (QCQP) with Toeplitz-Hermitian quadratics [KS15]. For non-convex stochastic optimization with hidden structure, projected gradient-based algorithms can achieve the same convergence rate as in convex optimization for both strongly convex [FHH23] and convex objectives [CHHY22] under certain assumptions. Furthermore, QCQP, which is generally NP-hard, can be solved in polynomial time when hidden convexity is present [KS15].

### A.3 Ball-Constrained Optimization

To improve algorithmic performance and reduce the computational cost of constrained optimization, recent work has explored the use of ball-constrained optimization. The idea dates back to the ellipsoid method [GLS81], which iteratively encloses the feasible region in shrinking ellipsoids that contain the optimal solution. Despite its theoretical appeal and linear convergence, the ellipsoid method suffers from high computational complexity as $\mathcal{O}\left(n^4\right)$, making it impractical for large-scale problems. More recently, [CJJ+20, GLRR25] studied acceleration techniques using ball-optimization oracles for specific problem settings. Inherently, ball-constrained optimization exhibits favorable properties; for example, solving quadratic problems over a ball using a combination of bisection and Newton's method can achieve a convergence rate of $\mathcal{O}(\log \log(1/\epsilon))$ [Ye94, Ye01], which matches the fast rate of Newton's method for unconstrained problems [BBV04]. These developments highlight the potential of ball-constrained techniques in designing efficient and scalable optimization algorithms.

# B    Homeomorphism of Convex set via Gauge Mapping

In this section, we provide the omitted details in Sec. 2 and Sec. 3.

## B.1    Handling Constraint Set with Equality

We first explain how to apply our framework to handle constraints with linear equality constraints as mentioned in Sec. 2. Consider the constrained set $\mathcal{K}$ as follows

$$\mathcal{K} = \{\mathbf{x} \mid \mathbf{q}(\mathbf{x}) = \mathbf{0}, g_1(\mathbf{x}) \leq 0, \cdots, g_m(\mathbf{x}) \leq 0\}$$

where $\mathbf{q}(\cdot) = (q_1, q_2, \cdots, q_{m_{\text{eq}}})$ with $q_i : \mathbb{R}^n \rightarrow \mathbb{R}$ are linear/affine functions.

Note that the rank of $J_{\mathbf{q}}$ is constant for all $\mathbf{x}$, i.e.,

$$\text{rank}\,(J_{\mathbf{q}}(\mathbf{x})) = r, \quad \forall \mathbf{x} \in \mathcal{K}.$$

Then $\{\mathbf{q}(\mathbf{x}) = \mathbf{0}\}$ is of dimension $n - r$ by the Constant-Rank Level Set Theorem [LL12]. In other words, we can use a subset of decision variables $\mathbf{x}_1 \in \mathbb{R}^{n-r}$ and reconstruct full decision variable $[\mathbf{x}_1, \mathbf{x}_2] \in \mathbb{R}^n$ via the equality constraint, where $\mathbf{x}_2 = \phi(\mathbf{x}_1)$ and $\mathbf{q}([\mathbf{x}_1, \phi(\mathbf{x}_1)]) = \mathbf{0}$. Such a reconstruction process ensures the feasibility of the equality constraint. Then the constraint $\mathcal{K}$ can be reformulated as

$$\mathcal{K}^s = \{\mathbf{x}_1 \in \mathbb{R}^{n-r} \mid g_1(\mathbf{x}_1, \phi(\mathbf{x}_1)) \leq 0, \cdots, g_m(\mathbf{x}_1, \phi(\mathbf{x}_1)) \leq 0\}.$$

It follows from the reconstruction that

$$(\mathbf{x}_1, \mathbf{x}_2 = \phi(\mathbf{x}_2)) \in \mathcal{K} \Leftrightarrow \mathbf{x}_1 \in \mathcal{K}^s.$$

Thus, we can assume the constrained set has no inequalities without loss of generality.

## B.2    Gauge Mapping for Convex Set

We first recall the definitions of gauge function/mapping.

**Definition B.1** (Gauge/Minkowski function [BM08]). Let $\mathcal{C} \subset \mathbb{R}^n$ be a compact convex set with a non-empty interior. The gauge/Minkowski function $\gamma_{\mathcal{C}} : \mathbb{R}^n \times \text{int}(\mathcal{C}) \rightarrow \mathbb{R}_{\geq 0}$ is defined as

$$\gamma_{\mathcal{C}}(\mathbf{x}, \mathbf{x}^\circ) = \inf\{\lambda \geq 0 \mid \mathbf{x} \in \lambda(\mathcal{C} - \mathbf{x}^\circ)\},$$

where $\mathbf{x}^\circ \in \text{int}(\mathcal{C})$.

Building upon this foundation, we define the gauge mapping between two compact convex sets:

**Definition B.2** (Gauge mapping [TZ22]). Let $\mathcal{Z}, \mathcal{C} \subset \mathbb{R}^n$ be compact convex sets with interior points $\mathbf{z}^\circ \in \text{int}(\mathcal{Z})$ and $\mathbf{x}^\circ \in \text{int}(\mathcal{C})$, respectively. Then

1) the gauge mapping $\psi : \mathcal{Z} \rightarrow \mathcal{C}$ is defined as:

$$\psi(\mathbf{z}) = \frac{\gamma_{\mathcal{Z}}(\mathbf{z} - \mathbf{z}^\circ, \mathbf{z}^\circ)}{\gamma_{\mathcal{C}}(\mathbf{z} - \mathbf{z}^\circ, \mathbf{x}^\circ)}(\mathbf{z} - \mathbf{z}^\circ) + \mathbf{x}^\circ, \; \mathbf{z} \in \mathcal{Z},$$

2) and the inverse gauge mapping $\psi^{-1} : \mathcal{C} \rightarrow \mathcal{Z}$ is given by:

$$\psi^{-1}(\mathbf{x}) = \frac{\gamma_{\mathcal{C}}(\mathbf{x} - \mathbf{x}^\circ, \mathbf{x}^\circ)}{\gamma_{\mathcal{Z}}(\mathbf{x} - \mathbf{x}^\circ, \mathbf{z}^\circ)}(\mathbf{x} - \mathbf{x}^\circ) + \mathbf{z}^\circ, \; \mathbf{x} \in \mathcal{C}.$$

We have the following remarks based on the definition.

- In essence, the gauge mapping scales the boundary of a convex set from an interior point to another convex set and with translation to its interior point.
- When $\mathcal{Z}$ is a unit ball, the gauge mapping in Def. B.2 is simplified as Def. 3.2:

$$\psi(\mathbf{z}) = \frac{\|\mathbf{z}\|}{\gamma_{\mathcal{C}}(\mathbf{z}, \mathbf{x}^\circ)}\mathbf{z} + \mathbf{x}^\circ, \; \forall \mathbf{z} \in \mathcal{B}, \qquad \psi^{-1}(\mathbf{x}) = \frac{\gamma_{\mathcal{C}}(\mathbf{x} - \mathbf{x}^\circ, \mathbf{x}^\circ)}{\|\mathbf{x} - \mathbf{x}^\circ\|}(\mathbf{x} - \mathbf{x}^\circ), \; \forall \mathbf{x} \in \mathcal{C}.$$

## B.3 Properties of Gauge Mapping

The gauge function satisfies the following properties.

**Proposition B.3** (Basic properties of gauge function). *Let $\mathcal{C}$ be a compact and convex set. For all $\mathbf{x}, \mathbf{y} \in \mathbb{R}^n$ and $\alpha \geq 0$, gauge function $\gamma_{\mathcal{C}}(\cdot, \mathbf{x}^\circ)$ given an interior point $\mathbf{x}^\circ \in \text{int}(\mathcal{C})$ satisfies:*

- *Non-negativity: $\gamma_{\mathcal{C}}(\mathbf{x}, \mathbf{x}^\circ) \geq 0$.*

- *Positive homogeneity: $\gamma_{\mathcal{C}}(\alpha\mathbf{x}, \mathbf{x}^\circ) = \alpha\gamma_{\mathcal{C}}(\mathbf{x}, \mathbf{x}^\circ)$.*

- *Subadditivity: $\gamma_{\mathcal{C}}(\mathbf{x} + \mathbf{y}, \mathbf{x}^\circ) \leq \gamma_{\mathcal{C}}(\mathbf{x}, \mathbf{x}^\circ) + \gamma_{\mathcal{C}}(\mathbf{y}, \mathbf{x}^\circ)$.*

- *Convexity.*

- *Differentiability: Gauge function is twice differentiable almost everywhere.*

- *Upper/lower bounds: $\gamma_{\mathcal{C}}(\mathbf{x}, \mathbf{x}^\circ) \in [\|\mathbf{x}\|/r_{\mathrm{o}}, \|\mathbf{x}\|/r_{\mathrm{i}}]$.*

- *Lipschitz continuous: $\|\gamma_{\mathcal{C}}(\mathbf{x}, \mathbf{x}^\circ) - \gamma_{\mathcal{C}}(\mathbf{y}, \mathbf{x}^\circ)\| \leq \frac{1}{r_{\mathrm{i}}}\|\mathbf{x} - \mathbf{y}\|$.*

*Proof.* We show them one by one in the following.

1)2) The definition can directly derive non-negativity and positive homogeneity.

3) To show subadditivity, let

$$\lambda_x = \gamma_{\mathcal{C}}(\mathbf{x}, \mathbf{x}^\circ) \quad \text{and} \quad \lambda_y = \gamma_{\mathcal{C}}(\mathbf{y}, \mathbf{x}^\circ).$$

By the definition of the gauge function, there exist points $\mathbf{u}, \mathbf{v} \in \mathcal{C} - \mathbf{x}^\circ$ such that $\mathbf{x} = \lambda_x \mathbf{u}$ and $\mathbf{y} = \lambda_y \mathbf{v}$. Now, consider the sum: $\mathbf{x} + \mathbf{y} = \lambda_x \mathbf{u} + \lambda_y \mathbf{v}$. Write this sum as

$$\mathbf{x} + \mathbf{y} = (\lambda_x + \lambda_y)\left(\frac{\lambda_x}{\lambda_x + \lambda_y}\mathbf{u} + \frac{\lambda_y}{\lambda_x + \lambda_y}\mathbf{v}\right).$$

Since $\mathcal{C} - \mathbf{x}^\circ$ is convex (as a translation of the convex set $\mathcal{C}$), the convex combination

$$\frac{\lambda_x}{\lambda_x + \lambda_y}\mathbf{u} + \frac{\lambda_y}{\lambda_x + \lambda_y}\mathbf{v} \in \mathcal{C} - \mathbf{x}^\circ.$$

Thus, $\mathbf{x} + \mathbf{y} \in (\lambda_x + \lambda_y)(\mathcal{C} - \mathbf{x}^\circ)$ which implies by the definition of $\gamma_{\mathcal{C}}$ that $\gamma_{\mathcal{C}}(\mathbf{x} + \mathbf{y}, \mathbf{x}^\circ) \leq \lambda_x + \lambda_y$. Since $\lambda_x$ and $\lambda_y$ can be arbitrarily approximated by sequences converging to $\gamma_{\mathcal{C}}(\mathbf{x}, \mathbf{x}^\circ)$ and $\gamma_{\mathcal{C}}(\mathbf{y}, \mathbf{x}^\circ)$ (if the infimum is not attained exactly), we conclude $\gamma_{\mathcal{C}}(\mathbf{x} + \mathbf{y}, \mathbf{x}^\circ) \leq \gamma_{\mathcal{C}}(\mathbf{x}, \mathbf{x}^\circ) + \gamma_{\mathcal{C}}(\mathbf{y}, \mathbf{x}^\circ)$.

4) Convexity is induced by Positive homogeneity and Subadditivity.

5) Differentiability is from the fact that Convex functions over a compact set are twice differentiable almost everywhere [Eva18].

6) By the definition of gauge function (Def. B.5), we have:

$$\gamma_{\mathcal{C}}(\mathbf{x}, \mathbf{x}^\circ) = \frac{\|\mathbf{x}\|}{d_{\mathcal{C}}(\mathbf{x}^\circ, \mathbf{x}/\|\mathbf{x}\|)} \in [\|\mathbf{x}\|/r_{\mathrm{o}}, \|\mathbf{x}\|/r_{\mathrm{i}}]$$

7) By the subadditivity, we have:

$\gamma_{\mathcal{C}}(\mathbf{x}, \mathbf{x}^\circ) - \gamma_{\mathcal{C}}(\mathbf{y}, \mathbf{x}^\circ) \leq \gamma_{\mathcal{C}}(\mathbf{x} - \mathbf{y}, \mathbf{x}^\circ) + \gamma_{\mathcal{C}}(\mathbf{y}, \mathbf{x}^\circ) - \gamma_{\mathcal{C}}(\mathbf{y}, \mathbf{x}^\circ) = \gamma_{\mathcal{C}}(\mathbf{x} - \mathbf{y}, \mathbf{x}^\circ) \leq \|\mathbf{x} - \mathbf{y}\|/r_{\mathrm{i}}$
Similarly, we have:

$$\gamma_{\mathcal{C}}(\mathbf{y}, \mathbf{x}^\circ) - \gamma_{\mathcal{C}}(\mathbf{x}, \mathbf{x}^\circ) \leq \gamma_{\mathcal{C}}(\mathbf{y} - \mathbf{x}, \mathbf{x}^\circ) \leq \|\mathbf{x} - \mathbf{y}\|/r_{\mathrm{i}}$$

Thus, we have

$$\|\gamma_{\mathcal{C}}(\mathbf{x}, \mathbf{x}^\circ) - \gamma_{\mathcal{C}}(\mathbf{y}, \mathbf{x}^\circ)\| \leq \frac{1}{r_{\mathrm{i}}}\|\mathbf{x} - \mathbf{y}\|.$$

$\square$

Based on the non-negativity, positive homogeneity, and subadditivity, the gauge function generalizes the concept of a norm. For a set $\mathcal{C}$ that is symmetric about the origin, the gauge function $\gamma_{\mathcal{C}}(\mathbf{x}, \mathbf{0})$ defines a norm. In particular, when $\mathcal{C} = \mathcal{B}_p = \{\mathbf{x} \in \mathbb{R}^n \mid \|\mathbf{x}\|_p \leq 1\}$ is the unit ball of the $p$-norm, we have $\gamma_{\mathcal{B}_p}(\mathbf{x}, \mathbf{0}) = \|\mathbf{x}\|_p$.

### B.3.1 Proof of Proposition 3.3: Bi-Lipschitz Constants of Gauge Mapping

Next, we prove Proposition 3.3, stating that the gauge mapping is bi-Lipschitz continuous with constant depending on $r_{\mathrm{i}}$ and $r_{\mathrm{o}}$.

*Proof of Proposition 3.3.* We begin with the forward gauge mapping from the 2-norm ball $\mathcal{B}$ to $\mathcal{C}$ as

$$\boldsymbol{\psi}(\mathbf{z}) = \frac{\|\mathbf{z}\|}{\gamma_{\mathcal{C}}(\mathbf{z}, \mathbf{x}^{\circ})}\, \mathbf{z} + \mathbf{x}^{\circ},\ \forall \mathbf{z} \in \mathcal{B}$$

Differentiating $\boldsymbol{\psi}(\mathbf{z})$ with respect to $\mathbf{z}$ (using the product and quotient rules) yields

$$J_{\boldsymbol{\psi}}(\mathbf{z}) = \frac{\|\mathbf{z}\|}{\gamma_{\mathcal{C}}(\mathbf{z}, \mathbf{x}^{\circ})}\mathbf{I} + \frac{\mathbf{z}\mathbf{z}^{\top}}{\gamma_{\mathcal{C}}(\mathbf{z}, \mathbf{x}^{\circ})\|\mathbf{z}\|} - \frac{\|\mathbf{z}\|}{\gamma_{\mathcal{C}}(\mathbf{z}, \mathbf{x}^{\circ})^2}\, \mathbf{z}\Big(\nabla_{\mathbf{z}}\gamma_{\mathcal{C}}(\mathbf{z}, \mathbf{x}^{\circ})\Big)^{\top}.$$

Taking the operator norm and applying the triangle inequality gives

$$\left\|J_{\boldsymbol{\psi}}(\mathbf{z})\right\| \leq \frac{\|\mathbf{z}\|}{\gamma_{\mathcal{C}}(\mathbf{z}, \mathbf{x}^{\circ})} + \frac{\|\mathbf{z}\|}{\gamma_{\mathcal{C}}(\mathbf{z}, \mathbf{x}^{\circ})} + \frac{\|\mathbf{z}\|^2}{\gamma_{\mathcal{C}}(\mathbf{z}, \mathbf{x}^{\circ})^2}\left\|\nabla_{\mathbf{z}}\gamma_{\mathcal{C}}(\mathbf{z}, \mathbf{x}^{\circ})\right\|$$

$$\leq r_{\mathrm{o}} + r_{\mathrm{o}} + \frac{r_{\mathrm{o}}^2}{r_{\mathrm{i}}},$$

where in the last inequality we have used the facts that (i) for $\mathbf{z} \in \mathcal{B}$ one has $\|\mathbf{z}\| \leq 1$, (ii) the gauge function satisfies $\gamma_{\mathcal{C}}(\mathbf{z}, \mathbf{x}^{\circ}) \in [\|\mathbf{z}\|/r_{\mathrm{o}}, \|\mathbf{z}\|/r_{\mathrm{i}}]$, and (iii) $\|\nabla_{\mathbf{z}}\gamma_{\mathcal{C}}(\mathbf{z}, \mathbf{x}^{\circ})\|$ is bounded by $1/r_{\mathrm{i}}$. In summary, we obtain

$$\left\|J_{\boldsymbol{\psi}}(\mathbf{z})\right\| \leq 2r_{\mathrm{o}} + \frac{r_{\mathrm{o}}^2}{r_{\mathrm{i}}},$$

which proves that the forward Lipschitz constant of $\boldsymbol{\psi}$ satisfies

$$\mathrm{Lip}(\boldsymbol{\psi}) \leq 2r_{\mathrm{o}} + \frac{r_{\mathrm{o}}^2}{r_{\mathrm{i}}}.$$

Next, consider the inverse gauge mapping from $\mathcal{C}$ to the 2-norm ball as

$$\boldsymbol{\psi}^{-1}(\mathbf{x}) = \frac{\gamma_{\mathcal{C}}(\mathbf{x} - \mathbf{x}^{\circ}, \mathbf{x}^{\circ})}{\|\mathbf{x} - \mathbf{x}^{\circ}\|}\, (\mathbf{x} - \mathbf{x}^{\circ}),\ \forall \mathbf{x} \in \mathcal{C}.$$

Differentiating with respect to $\mathbf{x}$ gives

$$J_{\boldsymbol{\psi}^{-1}}(\mathbf{x}) = \nabla\gamma_{\mathcal{C}}(\mathbf{x} - \mathbf{x}^{\circ}, \mathbf{x}^{\circ})\left(\frac{\mathbf{x} - \mathbf{x}^{\circ}}{\|\mathbf{x} - \mathbf{x}^{\circ}\|}\right)^{\top} + \gamma_{\mathcal{C}}(\mathbf{x} - \mathbf{x}^{\circ}, \mathbf{x}^{\circ}) \cdot \frac{I - \frac{(\mathbf{x} - \mathbf{x}^{\circ})(\mathbf{x} - \mathbf{x}^{\circ})^{\top}}{\|\mathbf{x} - \mathbf{x}^{\circ}\|^2}}{\|\mathbf{x} - \mathbf{x}^{\circ}\|}.$$

Taking norms and again using the triangle inequality leads to

$$\left\|J_{\boldsymbol{\psi}^{-1}}(\mathbf{x})\right\| \leq \left\|\nabla\gamma_{\mathcal{C}}(\mathbf{x} - \mathbf{x}^{\circ}, \mathbf{x}^{\circ})\left(\frac{\mathbf{x} - \mathbf{x}^{\circ}}{\|\mathbf{x} - \mathbf{x}^{\circ}\|_2}\right)^{\top}\right\| + \gamma_{\mathcal{C}}(\mathbf{x} - \mathbf{x}^{\circ}, \mathbf{x}^{\circ})\left\|\frac{I - \frac{(\mathbf{x} - \mathbf{x}^{\circ})(\mathbf{x} - \mathbf{x}^{\circ})^{\top}}{\|\mathbf{x} - \mathbf{x}^{\circ}\|^2}}{\|\mathbf{x} - \mathbf{x}^{\circ}\|}\right\|.$$

Using the bound $\gamma_{\mathcal{C}}(\mathbf{x} - \mathbf{x}^{\circ}, \mathbf{x}^{\circ}) \in [\|\mathbf{x} - \mathbf{x}^{\circ}\|/r_{\mathrm{o}}, \|\mathbf{x} - \mathbf{x}^{\circ}\|/r_{\mathrm{i}}]$ and the projection matrix related term has norm at most $1/\|\mathbf{x} - \mathbf{x}^{\circ}\|$), we have

$$\left\|J_{\boldsymbol{\psi}^{-1}}(\mathbf{x})\right\| \leq \frac{1}{r_{\mathrm{i}}} + \frac{1}{r_{\mathrm{i}}} = \frac{2}{r_{\mathrm{i}}}.$$

Thus, the inverse Lipschitz constant is bounded by

$$\mathrm{Lip}(\boldsymbol{\psi}^{-1}) \leq \frac{2}{r_{\mathrm{i}}}.$$

This completes the proof. $\qquad\square$

### B.3.2 Smoothing Technique for Gauge Mapping

Moreover, we have the following property for gauge mapping as a corollary from Proposition B.3.

**Corollary B.4.** *The gauge mapping $\psi$ defined in Def. B.2 is twice differentiable almost everywhere.*

This corollary follows directly from Proposition B.3 and aligns with our general assumptions (see Appendix C.2). *However, the existence of a zero-measure set of non-differentiable points creates a gap between our theoretical requirements and the current formulation.* For instance, in max-cut SDP problems, low-rank optimal solutions lead to non-differentiable points on the PSD cone constraint boundary due to eigenvalue multiplicity at the maximum eigenvalue, causing convergence behavior to oscillate around the optimum. While these points may have no practical impact on most engineering problems, mathematical rigor demands that we address this discrepancy.

To bridge this gap, we introduce a smoothing technique. Recall that the gauge mapping $\psi$ from the unit ball $\mathcal{B}$ to a convex set $\mathcal{C}$ is computed (see Appendix B.4) as:

$$\psi(\mathbf{z}) = \frac{1}{\kappa_{\mathcal{C}}\left(\mathbf{x}^{\circ}, \mathbf{z}/\|\mathbf{z}\|\right)} \cdot \mathbf{z} + \mathbf{x}^{\circ}$$

where $\kappa_{\mathcal{C}}$ is the inverse distance function (Def. B.5).

In practice, most constraints we encounter are listed in Table 2. The non-smoothness of $\kappa_{\mathcal{C}}$ (and thus $\psi$) arises from the use of the max operator (including the ReLU operator $[\cdot]^{+}$) when handling multiple constraints. We define $\kappa_i(\mathbf{x}^{\circ}, \mathbf{v})$ such that $\kappa_{\mathcal{C}_i}(\mathbf{x}^{\circ}, \mathbf{v}) = [\kappa_i(\mathbf{x}^{\circ}, \mathbf{v})]^{+}$ in Table 2. Hence, each $\kappa_i$ is smooth. We then express $\kappa_{\mathcal{C}}(\mathbf{x}^{\circ}, \mathbf{v}) = \max_{1 \leq i \leq m} \{\kappa_i(\mathbf{x}^{\circ}, \mathbf{v}), 0\}$ and, letting $\kappa_{m+1} = 0$, we simplify this as

$$\kappa_{\mathcal{C}}(\mathbf{x}^{\circ}, \mathbf{v}) = \max_{1 \leq i \leq m} \{\kappa_i(\mathbf{x}^{\circ}, \mathbf{v})\},$$

with loss of generality. In the following, for convenience, we fix $\mathbf{x}^{\circ}$ and write $\kappa_i(\mathbf{v})$ instead of $\kappa_i(\mathbf{x}^{\circ}, \mathbf{v})$.

We smooth $\kappa_{\mathcal{C}}$ using the log-sum-exp approximation (also called Nesterov smoothing) with parameter $\eta > 0$ following [Gri24b, Nes05]:

$$\kappa_{\eta}(\mathbf{v}) = \eta \log \left( \sum_{i=0}^{n} \exp \left( \frac{\kappa_i(\mathbf{v})}{\eta} \right) \right),$$

with the corresponding gradient

$$\nabla \kappa_{\eta}(\mathbf{v}) = \sum_{i=1}^{m} \lambda_i \nabla \kappa_i(\mathbf{v}) \quad \text{where} \quad \lambda_i = \frac{\exp\left(\kappa_i(\mathbf{v})/\eta\right)}{\sum_{j=1}^{m} \exp\left(\kappa_j(\mathbf{v})/\eta\right)}.$$

By replacing the objective $h = f \circ \psi$ with $h_{\eta} = f \circ \psi_{\eta}$ where $\psi_{\eta}(\mathbf{z}) = \frac{1}{\kappa_{\eta}(\mathbf{x}^{\circ}, \mathbf{z}/\|\mathbf{z}\|)} \cdot \mathbf{z} + \mathbf{x}^{\circ}$, we obtain a twice-differentiable homeomorphism. The smoothing introduces an optimality gap of $\mathcal{O}(\eta)$ [Nes05, Gri24b] (here the hidden constant is dependent on the Lipschitz constant of $f$), which can be made arbitrarily small by choosing $\eta$ sufficiently small.

### B.4 Computation of Gauge Mapping

For computational purposes, we introduce a point-to-boundary distance function and its inverse.

**Definition B.5** (Point-to-boundary distance [THH23]). Let $\mathcal{C} \subset \mathbb{R}^n$ be a compact convex set and $\mathbf{x}^{\circ} \in \text{int}(\mathcal{C})$. For any unit vector $\mathbf{v} \in \mathbb{S}^{n-1} = \{\mathbf{u} \in \mathbb{R}^n \mid \|\mathbf{u}\| = 1\}$, we define the interior-point-to-boundary distance function $d_{\mathcal{C}} : \text{int}(\mathcal{C}) \times \mathbb{S}^{n-1} \to \mathbb{R}_{\geq 0}$ along direction $\mathbf{v}$ as

$$d_{\mathcal{C}}(\mathbf{x}^{\circ}, \mathbf{v}) = \sup\{\lambda \geq 0 \mid \mathbf{x}^{\circ} + \lambda \mathbf{v} \in \mathcal{C}\}.$$

The inverse distance function $\kappa_{\mathcal{C}} : \text{int}(\mathcal{C}) \times \mathbb{S}^{n-1} \to \mathbb{R}_{\geq 0}$ is defined as $\kappa_{\mathcal{C}}(\mathbf{x}^{\circ}, \mathbf{v}) := 1/d_{\mathcal{C}}(\mathbf{x}^{\circ}, \mathbf{v})$.

We then have the following rules for computing the gauge mapping with the help of the point-to-boundary distance function.

- This distance function relates to the gauge function as:
$$\gamma_{\mathcal{C}}(\mathbf{x}, \mathbf{x}^\circ) = \kappa_{\mathcal{C}}(\mathbf{x}^\circ, \mathbf{x}/\|\mathbf{x}\|) \cdot \|\mathbf{x}\| = \frac{\|\mathbf{x}\|}{d_{\mathcal{C}}(\mathbf{x}^\circ, \mathbf{x}/\|\mathbf{x}\|)}.$$

- Further, considering the positive homogeneity, we have:
$$\gamma_{\mathcal{C}}(\mathbf{x}/\|\mathbf{x}\|, \mathbf{x}^\circ) = \kappa_{\mathcal{C}}(\mathbf{x}^\circ, \mathbf{x}/\|\mathbf{x}\|) = \frac{1}{d_{\mathcal{C}}(\mathbf{x}^\circ, \mathbf{x}/\|\mathbf{x}\|)}.$$

- The gauge mapping between $\mathcal{B}$ and $\mathcal{C}$ can be simplified as:
$$\boldsymbol{\psi}(\mathbf{z}) = d_{\mathcal{C}}(\mathbf{x}^\circ, \mathbf{z}/\|\mathbf{z}\|) \cdot \mathbf{z} + \mathbf{x}^\circ, \ \forall \mathbf{z} \in \mathcal{B}$$
$$\boldsymbol{\psi}^{-1}(\mathbf{x}) = \frac{\mathbf{x} - \mathbf{x}^\circ}{d_{\mathcal{C}}(\mathbf{x}^\circ, (\mathbf{x} - \mathbf{x}^\circ)/\|\mathbf{x} - \mathbf{x}^\circ\|)}, \ \forall \mathbf{x} \in \mathcal{C}.$$

Therefore, we can compute the inverse distance function to obtain the gauge mapping. In practice, we can compute the inverse distance function $\kappa_{\mathcal{C}}(\mathbf{x}^\circ, \cdot)$ as follows.

- *Closed-form for common constrained set.* For various common constraint types, Table 2 provides closed-form expressions of the inverse distance function and Fig. 6 show the actual computational cost for it. Most matrix calculations can be computed and stored offline before being applied online given $\mathbf{v}$.

- *Bisection-based algorithm for general constrained set .* When the inverse distance function lacks a closed-form expression, we employ an efficient bisection algorithm detailed in Alg. 2. This algorithm supports batch processing, enabling efficient parallel computation for multiple inputs simultaneously.

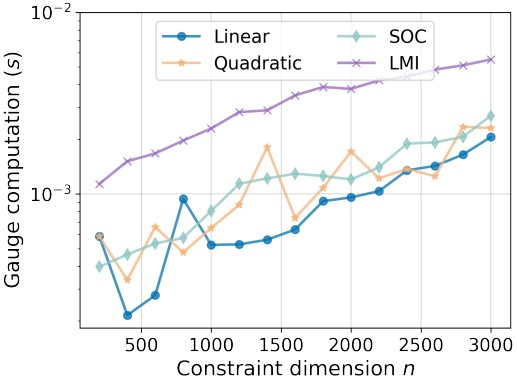

Figure 6: Illustration of gauge function calculation time as constraint dimension varies.

## B.5    Gauge Mapping in Hom-PGD

**Optimizing the interior point in Hom-PGD.** As shown in Sec. 4, one can select a central interior point with large inner radius $r_{\mathrm{i}}$ or smaller outer radius $r_{\mathrm{o}}$ to reduce the Lipschitz constant of the gauge mapping $\boldsymbol{\psi}$ thereby boosting the convergence rate of Hom-PGD. Here, we introduce two types of interior points which maximize $r_{\mathrm{i}}$ and minimize $r_{\mathrm{o}}$, respectively, below.

- Chebyshev center: maximizes the minimum distance from the point to the boundary of the set for a large inner radius $r_{\mathrm{i}}$:
$$\mathbf{x}^\circ = \arg\max_{\mathbf{x} \in \mathcal{K}} \min_{\mathbf{y} \in \partial\mathcal{K}} \|\mathbf{x} - \mathbf{y}\|.$$

- Min-max center: minimizes the maximum distance from the point to the boundary of the set for a smaller outer radius $r_{\mathrm{o}}$:
$$\mathbf{x}^\circ = \arg\min_{\mathbf{x} \in \mathcal{K}} \max_{\mathbf{y} \in \partial\mathcal{K}} \|\mathbf{x} - \mathbf{y}\|.$$

Table 2: Closed-form Expressions for Inverse Distance Functions [THH23]

| Constrained Set $\mathcal{C}$ | Formulation | Inverse Distance Function $\kappa_{\mathcal{C}}(\mathbf{x}^\circ, \mathbf{v})$ |
| --- | --- | --- |
| Linear | $\{\mathbf{x} : \mathbf{a}^\top \mathbf{x} \leq b\}$ | $\left(\frac{\mathbf{a}^\top \mathbf{v}}{b - \mathbf{a}^\top \mathbf{x}^\circ}\right)^+$ |
| Quadratic | $\{\mathbf{x} : \mathbf{x}^\top \mathbf{Q} \mathbf{x} + \mathbf{a}^\top \mathbf{x} \leq b\}$ | $(1/\mathrm{root}\,(A_\mathbf{Q}, B_\mathbf{Q}, C_\mathbf{Q}))^+$ |
| Second Order Cone | $\{\mathbf{x} : \|\mathbf{A}^\top \mathbf{x} + \mathbf{p}\|_2 \leq \mathbf{a}^\top \mathbf{x} + b\}$ | $(1/\mathrm{root}(A_\mathrm{S}, B_\mathrm{S}, C_\mathrm{S}))^+$ |
| Linear Matrix Inequality | $\{\mathbf{x} : \sum_{i=1}^n x_i \cdot \mathbf{F}_i + \mathbf{F}_0 \succeq 0\}$ | $\left(\mathrm{eig}\left(\mathbf{L}^\top (-\mathbf{S})\,\mathbf{L}\right)\right)^+$ |
| Intersections | $\mathcal{C} = \bigcap_{i=1}^m \mathcal{C}_i$ | $\max\limits_{1 \leq i \leq m} \{\kappa_{\mathcal{C}_i}(\mathbf{x}^\circ, \mathbf{v})\}$ |

[1] Notation: $\mathbf{x}, \mathbf{a} \in \mathbb{R}^n$, $b \in \mathbb{R}$, $\mathbf{Q} \in \mathbb{S}_+^n$, $\mathbf{A} \in \mathbb{R}^{n \times m}$, $\mathbf{p} \in \mathbb{R}^m$, $\mathbf{F}_0, \cdots, \mathbf{F}_n \in \mathbb{R}^{m \times m}$, $\mathbf{X} \in \mathbb{R}^{n \times n}$, $(\cdot)^+ = \max(\cdot, 0)$.

[2] $\mathrm{root}(x_1, x_2, x_3) = \frac{-x_2 \pm \sqrt{x_2^2 - 4x_1 x_3}}{2x_1}$ denotes the roots for quadratic equation.

[3] $A_\mathbf{Q} = \mathbf{v}^\top \mathbf{Q} \mathbf{v}$, $B_\mathbf{Q} = 2\mathbf{x}^{\circ\top} \mathbf{Q} \mathbf{v} + \mathbf{a}^\top \mathbf{v}$, $C_\mathbf{Q} = \mathbf{x}^{\circ\top} \mathbf{Q} \mathbf{x}^\circ + \mathbf{a}^\top \mathbf{x}^\circ - b$.

[4] $A_\mathrm{S} = (\mathbf{A}^\top \mathbf{v})^\top (\mathbf{A}^\top \mathbf{v}) - (\mathbf{a}^\top \mathbf{v})^2$, $B_\mathrm{S} = 2(\mathbf{A}^\top \mathbf{x}^\circ + \mathbf{p})^\top (\mathbf{A}^\top \mathbf{v}) - 2(\mathbf{a}^\top \mathbf{x}^\circ + b)(\mathbf{a}^\top \mathbf{v})$, $C_\mathrm{S} = (\mathbf{A}^\top \mathbf{x}^\circ + \mathbf{p})^\top (\mathbf{A}^\top \mathbf{x}^\circ + \mathbf{p}) - (\mathbf{a}^\top \mathbf{x}^\circ + b)^2$.

[5] $\mathbf{H} = \mathbf{F}_0 + \sum_{i=1}^n x_i^\circ \mathbf{F}_i$, $\mathbf{H}^{-1} = \mathbf{L}^\top \mathbf{L}$, $\mathbf{S} = \sum_{i=1}^n v_i \mathbf{F}_i$. $\mathrm{eig}(\mathbf{X}) = \lambda_1, \cdots, \lambda_n$ denotes the eigenvalues satisfying $\det(\mathbf{X} - \lambda \mathbf{I}) = 0$. Note that only the maximum eigenvalue is needed. Thus, power iteration methods can be applied to compute it efficiently.

[6] Note that all $\mathbf{v}$-independent terms can be computed only once and stored for use.

---

**Algorithm 2** Bisection Algorithm for Inverse Point-to-Boundary Distance

**Input:** A compact convex set $\mathcal{C}$, an interior point $\mathbf{x}^\circ \in \mathcal{C}$, and a unit vector $\mathbf{v}$.

1: $\alpha_l = 0$, and $\alpha_u = 1$
2: **while** $|\alpha_l - \alpha_u| \geq 10^{-3}$ **do**
3:    **if** $\mathbf{x}^\circ + \alpha_u \cdot \mathbf{v} \in \mathcal{C}$ **then**
4:       increase lower bound: $\alpha_l \leftarrow \alpha_u$
5:       double upper bound: $\alpha_u \leftarrow 2 \cdot \alpha_m$
6:    **else**
7:       bisection: $\alpha_m = (\alpha_l + \alpha_u)/2$
8:       **if** $\mathbf{x}^\circ + \alpha_m \cdot \mathbf{v} \in \mathcal{C}$ **then**
9:          increase lower bound: $\alpha_l \leftarrow \alpha_m$
10:       **else**
11:          decrease upper bound: $\alpha_u \leftarrow \alpha_m$
12:       **end if**
13:    **end if**
14: **end while**

**Output:** inverse distance: $\kappa_{\mathcal{C}}(\mathbf{x}^\circ, \mathbf{v}) \approx 1/\alpha_m$

---

According to the definitions, the Chebyshev center maximizes $r_\mathrm{i}$ and the min-max center minimizes $r_\mathrm{o}$. For certain convex sets, one can efficiently compute these centers. For instance, for a polyhedron, one can find the Chebyshev center by solving an LP (see e.g., [BBV04]). However, for general convex sets, computing the Chebyshev center entails solving a convex optimization problem that may be very hard due to the infinitely many constraints where each corresponding to a boundary point of the set.

In practice, we may minimize the constraint residual to find an approximate central interior point following [THH23]. Concretely, we solve the following convex program to obtain an interior point $\mathbf{x}^\circ \in \mathcal{K} := \{\mathbf{g}(\mathbf{x}) \leq \mathbf{0}\}$.

$$(\mathbf{x}^\circ, \epsilon) = \arg\max_{\mathbf{x}, \epsilon} \quad \epsilon$$
$$\text{s.t.} \quad \mathbf{g}(\mathbf{x}) \leq -\epsilon \mathbf{1},$$
$$\epsilon > 0.$$

Note that in the main experiments, we set $\epsilon = 10^{-3}$ as a constant to solve this convex feasibility problem to obtain an interior point (which may be close to the boundary). In the ablation study, we solve this residual maximization problem to obtain a "central" interior point for comparison.

**Computation of basic operations in Hom-PGD.** We now summarize the practical methods used to compute the key operations in Hom-PGD, including the gauge function/mapping and the gradient of $h = f \circ \psi$:

- *Computing the gauge function $\gamma_{\mathcal{K}}(\cdot, \mathbf{x}^\circ)$*: Compute the inverse distance function $\kappa_{\mathcal{K}}(\mathbf{x}^\circ, \cdot)$ via Algorithm 2 or using the closed form when available.

- *Computing the gauge mapping $\psi$*: Derived from Definition B.2, given $\gamma_{\mathcal{K}}(\cdot, \mathbf{x}^\circ)$.

- *Computing the gradient of $h = f \circ \psi$*: Approximated using numerical differentiation methods. If $h$ has a closed-form, the gradient can be computed using automatic differentiation methods (see e.g., [BPRS18]). For general cases, we adopt the finite difference method for computing the gradient of the gauge function in [LBGH23], i.e., for $i = 1, 2, \ldots, n$:

$$\nabla_i h(\mathbf{x}) = \frac{h(\mathbf{x} + \lambda \mathbf{e}_i) - h(\mathbf{x})}{\lambda}$$

given a proper small number $\lambda > 0$ where $\mathbf{e}_i$ denotes the standard basis over $\mathbb{R}^n$.

Next, we summarize the complexity for computing the above basic operations.

**Proposition B.6.** *With the zeroth-order oracle of $f$ and the membership oracle, the computational complexity of the basic operations in Hom-PGD is as follows.*

*(i) Computing gauge function $\gamma_{\mathcal{K}}(\cdot, \mathbf{x}^\circ)$ to an error $\epsilon$ costs $\mathcal{O}(\log 1/\epsilon)$.*

*(ii) Computing gauge mapping $\psi(\mathbf{x})$ costs $\tilde{\mathcal{O}}(n)$.*

*(iii) Computing the gradient of $h = f \circ \psi$ costs $\tilde{\mathcal{O}}(n^2)$.*

*Proof.* We show them one by one in the following.

(i) This is a classical result of the bisection-based algorithm. Using $\mathcal{O}(\log 1/\epsilon)$ number of calls to the membership oracle, we can get an $\epsilon_{\text{bis}}$-approximate solution to the gauge function $\gamma_{\mathcal{K}}(\cdot, \mathbf{x}^\circ)$. One could refer to, e.g., [LCL23, LBGH23, Mha22] for a detailed proof.

(ii) From Def. B.2, computing $\psi$ requires $\tilde{\mathcal{O}}(1)$ complexity to evaluate $\gamma_{\mathcal{C}}(\cdot, \mathbf{x}^\circ)$ and $\mathcal{O}(n)$ complexity to compute the scalar-vector product $d_{\mathcal{C}}(\mathbf{x}^\circ, \cdot) \cdot \mathbf{z}$.

(iii) From the finite difference method described above, computing each partial derivative $\nabla_i h$ requires evaluating $h$, which involves a zeroth-order oracle call to $f$ and a computation of the gauge mapping $\psi$. Therefore, computing each $\nabla_i h(\mathbf{x})$ $(i = 1, 2, \ldots, n)$ costs $\tilde{\mathcal{O}}(n)$, leading to a total of $\tilde{\mathcal{O}}(n^2)$ to compute the gradient mapping $\nabla h$. $\qquad\square$

# C Preliminaries for Technical Proof

In this section, we summarize the related basic concepts, notations, assumptions, and fundamental propositions and lemmas.

## C.1 Basic Concepts

We list the basic concepts used in this paper below.

- Indicator function. For a set $\mathcal{X} \subset \mathbb{R}^n$, the indicator function $\delta_{\mathcal{X}} : \mathbb{R}^n \to \mathbb{R}$ is defined as

$$\delta_{\mathcal{X}}(\mathbf{x}) := \begin{cases} 1 & \text{if } \mathbf{x} \in \mathcal{X}, \\ \infty & \text{if } \mathbf{x} \notin \mathcal{X}. \end{cases}$$

- Distance between a point and a set. For a closed set $\mathcal{X} \in \mathbb{R}^n$ and any $\mathbf{x} \in \mathbb{R}^n$, the distance between $\mathbf{x}$ and $\mathcal{X}$ is defined as $\text{dist}(\mathbf{x}, \mathcal{X}) = \inf_{\mathbf{y} \in \mathcal{X}} \|\mathbf{x} - \mathbf{y}\|$.

- Orthogonal projection. For a closed set $\mathcal{X}$, the orthogonal projection of a point $\mathbf{x} \in \mathbb{R}^n$ onto $\mathcal{X}$ is defined as $\Pi_{\mathcal{X}}(\mathbf{x}) = \arg\min_{\mathbf{y} \in \mathcal{X}} \|\mathbf{x} - \mathbf{y}\|$.

- Function convexity. For a differentiable function $f : \mathcal{X} \subseteq \mathbb{R}^n \to \mathbb{R}$, it is said to be convex if one of the following holds:

1) Jensen's inequality. For $\theta$ with $0 \le \theta \le 1$, we have $f(\theta\mathbf{x} + (1-\theta)\mathbf{y}) \le \theta f(\mathbf{x}) + (1-\theta)f(\mathbf{y})$ for all $\mathbf{x}, \mathbf{y} \in \mathcal{X}$.
2) first-order condition. $f(\mathbf{y}) \ge f(\mathbf{x}) + \langle \nabla f(\mathbf{x}), \mathbf{y} - \mathbf{x} \rangle, \forall \mathbf{x}, \mathbf{y} \in \mathcal{X}$.
3) monotone gradient. $(\nabla f(\mathbf{x}) - \nabla f(\mathbf{y}))^T(\mathbf{x} - \mathbf{y}) \ge 0$ for all $\mathbf{x}, \mathbf{y} \in \mathcal{X}$.

- $L$-Smoothness. A differentiable function $f : \mathcal{X} \subseteq \mathbb{R}^n \to \mathbb{R}$ is said be $L$-smooth if one of the following holds:
  1) zeroth-order condition. $f(\lambda\mathbf{x} + (1-\lambda)\mathbf{y}) \ge \lambda f(\mathbf{x}) + (1-\lambda)f(\mathbf{y}) - \frac{L}{2}\lambda(1-\lambda)\|\mathbf{y} - \mathbf{x}\|^2$, for all $\mathbf{x}, \mathbf{y} \in \mathcal{X}, \lambda \in [0,1]$.
  2) first-order condition. $f(\mathbf{y}) \le f(\mathbf{x}) + \langle \nabla f(\mathbf{x}), \mathbf{y} - \mathbf{x} \rangle + \frac{L}{2}\|\mathbf{y} - \mathbf{x}\|^2$, for all $\mathbf{x}, \mathbf{y} \in \mathcal{X}$.
  3) Lipschitz gradient. $\|\nabla f(\mathbf{y}) - \nabla f(\mathbf{x})\| \le L\|\mathbf{y} - \mathbf{x}\|$, for all $\mathbf{x}, \mathbf{y}$.

- $\mu$-Strong convexity. A differentiable function $f : \mathcal{X} \subseteq \mathbb{R}^n \to \mathbb{R}$ is said be $\mu$-strongly convex holds if one of the following holds:
  1) zeroth-order condition. $f(\lambda\mathbf{x} + (1-\lambda)\mathbf{y}) \le \lambda f(\mathbf{x}) + (1-\lambda)f(\mathbf{y}) - \frac{\mu}{2}\lambda(1-\lambda)\|\mathbf{y} - \mathbf{x}\|^2$, for all $\mathbf{x}, \mathbf{y} \in \mathcal{X}, \lambda \in [0,1]$.
  2) first-order condition. $f(\mathbf{y}) \ge f(\mathbf{x}) + \langle \nabla f(\mathbf{x}), \mathbf{y} - \mathbf{x} \rangle + \frac{\mu}{2}\|\mathbf{y} - \mathbf{x}\|^2$, for all $\mathbf{x}, \mathbf{y} \in \mathcal{X}$.
  3) strictly monotone gradient. $\langle \nabla f(\mathbf{y}) - \nabla f(\mathbf{x}), \mathbf{y} - \mathbf{x} \rangle \ge \mu\|\mathbf{y} - \mathbf{x}\|^2$, for all $\mathbf{x}, \mathbf{y} \in \mathcal{X}$.

- Stationary point. Consider a program $\{\min_{\mathbf{x}} f(\mathbf{x}), \text{ s.t., } \mathbf{x} \in \mathcal{X}\}$ where $f$ is differentiable and $\mathcal{X}$ is a convex set. Let $\alpha > 0$. Then a point $\mathbf{x}^*$ is called a stationary point for the program if
$$\Pi_{\mathcal{X}}(\mathbf{x} - \alpha\nabla f(\mathbf{x})) = \mathbf{x},$$
or equivalently (see e.g. [Bec14])
$$\langle \nabla f(\mathbf{x}^*), \mathbf{x} - \mathbf{x}^* \rangle \ge 0, \ \forall \mathbf{x} \in \mathcal{X}.$$

- Jacobian matrix. Suppose $\mathbf{f} : \mathbb{R}^n \to \mathbb{R}^m$ is a function such that each of its first-order partial derivatives exists on $\mathbb{R}^n$. Then the Jacobian matrix of $\mathbf{f}$, denoted $\mathrm{J}_{\mathbf{f}} \in \mathbb{R}^{m \times n}$, is defined as $\mathrm{J}_{\mathbf{f}} = (\frac{\partial f_i}{\partial x_j})_{ij}$.

- A Hessian of a function $f : \mathbb{R}^n \to \mathbb{R}$ is defined as $\nabla^2 f = (\frac{\partial^2 f}{\partial x_i \partial x_j})_{ij} \in \mathbb{R}^{n \times n}$, if its second-order partial derivatives exist. Moreover, for a mapping $\mathbf{f} : \mathbb{R}^n \to \mathbb{R}^m$ with existed second-order partial derivatives of each component $f_i$ ($i = 1, 2, \cdots, m$). The Hessian of $\mathbf{f}$ is defined as
$$\mathrm{H}(\mathbf{f}) = (\nabla^2 f_1, \cdots, \nabla^2 f_m).$$

## C.2 Basic Assumptions and Notations

In the following, we make assumptions throughout the paper.

- Assumptions on $f$ and constraints $g_i$ ($i = 1, 2, \cdots, m$) in problem $\mathbf{P}$:
  1) $f$ is $L_{f,0}$-Lipschitz continuous, i.e., $\|f(\mathbf{x}) - f(\mathbf{y})\| \le L_{f,0}\|\mathbf{x} - \mathbf{y}\|$ for any $\mathbf{x}, \mathbf{y}$.
  2) $f$ in problem $\mathbf{P}$ is differentiable and $L_f$-smooth.
  3) $f^* > -\infty$ where $f^* := \min_{\mathbf{x} \in \mathcal{K}} f(\mathbf{x})$.
  4) Each $g_i$ is $L_{g_i,0}$-Lipschitz continuous, differentiable, and $L_{g_i}$-smooth.

- Assumptions on the homeomorphic mapping $\psi : \mathbb{R}^n \to \mathbb{R}^n$:
  1) $\psi$ is differentiable with non-singular Jacobian $\mathrm{J}_{\psi}(\cdot)$,
  2) $\psi$ is $(\kappa_1, \kappa_2)$-bi-Lipschitz continuous for $\kappa_2 \ge \kappa_1 > 0$, i.e.,
$$\kappa_1\|\mathbf{u} - \mathbf{v}\| \le \|\psi(\mathbf{u}) - \psi(\mathbf{v})\| \le \kappa_2\|\mathbf{u} - \mathbf{v}\|.$$
  Then the Jacobian matrix, $\mathrm{J}_{\psi}(\cdot)$ and $\mathrm{J}_{\psi^{-1}}(\cdot)$ will satisfy
$$\|\mathrm{J}_{\psi}(\mathbf{z})\| \le \kappa_2, \ \forall \mathbf{z}, \quad \|\mathrm{J}_{\psi^{-1}}(\mathbf{x})\| \le \frac{1}{\kappa_1}, \ \forall \mathbf{x}.$$
  3) $\psi$ has $L_{\psi}$-Lipschitz continuous Jacobian matrix, i.e.,
$$\|\mathrm{J}_{\psi}(\mathbf{u}) - \mathrm{J}_{\psi}(\mathbf{v})\| \le L_{\psi}\|\mathbf{u} - \mathbf{v}\|, \ \forall \mathbf{u}, \mathbf{v}.$$
  4) $\psi$ has continuous Hessian, i.e.,
$$\mathrm{H}_{\psi}(\mathbf{z}) = (\nabla^2 \psi_1, \cdots, \nabla^2 \psi_n)$$
  exists and is continuous.

In addition, we summarize the commonly used notations in this paper in Table 3.

Table 3: Summary of Notations. The notations shown in the table is for problem **P** and we use the same type notations for problem **H**.

| Notation | Definition |
|---|---|
| $\|\cdot\|$ | $l2$-norm $\|\cdot\|_2$ |
| $\mathcal{B}$ | unit ball centered at 0 |
| $L_{f,0}$ | Lipschitz constant of $f$ |
| $L_f$ | $L_f$-smooth property of $f$ |
| $\mu_f$ | $\mu_f$-strong convexity of $f$ |
| $\kappa_1, \kappa_2$ | bi-Lipschitz constant of $\psi$ |
| $D$ | distortion of $\psi$, i.e., $\kappa_2/\kappa_1$ |
| $L_\psi$ | Lipschitz constant of $\mathrm{J}_\psi$ |
| $\mathrm{int}(\mathcal{K}), \partial\mathcal{K}$ | the interior,boundary of $\mathcal{K}$ |

## C.3 Basic Facts

In this section, we list the fundamental facts we will use in this paper.

**Proposition C.1** (Global optimality condition, see e.g., [BBV04, Bec17]). *For convex constrained optimization $\{\min_{\mathbf{x}} f(\mathbf{x}), \; s.t. \; \mathbf{x} \in \mathcal{X}\}$, $\mathbf{x}^*$ is a global optimum if and only if $\mathbf{x}^*$ is a stationary point of the problem, i.e.,*

$$\langle \nabla f(\mathbf{x}^*), \mathbf{x} - \mathbf{x}^* \rangle \geq 0.$$

**Proposition C.2** (Properties of orthogonal projection, see e.g., [Bec14]). *The projection operator $\Pi_{\mathcal{C}}$ over a closed and convex set $\mathcal{C}$ satisfies the following properties.*

1) *Optimality condition:* $\forall \mathbf{y} \in \mathcal{C}, \; \langle \mathbf{x} - \Pi_{\mathcal{C}}(\mathbf{x}), \mathbf{y} - \Pi_{\mathcal{C}}(\mathbf{x}) \rangle \leq 0.$

2) *Non-Expansiveness:* $\|\Pi_{\mathcal{C}}(\mathbf{x}) - \Pi_{\mathcal{C}}(\mathbf{y})\| \leq \|\mathbf{x} - \mathbf{y}\|.$

3) *Monotonicity:* $\langle \Pi_{\mathcal{C}}(\mathbf{x}) - \Pi_{\mathcal{C}}(\mathbf{y}), \mathbf{x} - \mathbf{y} \rangle \geq 0.$

We have the following lemma related to $\psi$ to help with the computation.

**Lemma C.3.** *Suppose $\mathrm{J}_\psi$ is $L_\psi$ Lipschitz, i.e., $\|\mathrm{J}_\psi(\mathbf{u}) - \mathrm{J}_\psi(\mathbf{z})\| \leq L_\psi \|\mathbf{u} - \mathbf{z}\|$ for any $\mathbf{u}$ and $\mathbf{z}$. Then, we have*

$$\|\psi(\mathbf{u}) - \psi(\mathbf{z}) - \mathrm{J}_\psi(\mathbf{z})(\mathbf{u} - \mathbf{z})\| \leq \frac{L_\psi \|\mathbf{u} - \mathbf{z}\|^2}{2}, \; \forall \mathbf{u}, \mathbf{z}.$$

One can refer to Lemma 1.2.3 [N$^+$18] for the proof.

Next, we list the following rules for basic computation:

- Jacobian equivalence: $\mathrm{J}_{\psi^{-1}}(\mathbf{x}) = \mathrm{J}_\psi^{-1}(\mathbf{z})$ for $\mathbf{z} = \psi(\mathbf{x})$.

- Chain rule for computing gradient of $h = f \circ \psi$:

$$\nabla h(\mathbf{z}) = \mathrm{J}_\psi(\mathbf{z})^\top \nabla f(\psi(\mathbf{z})) = \mathrm{J}_\psi(\mathbf{z})^\top \nabla f(\mathbf{x}).$$

- Chain rule for computing gradient of $f$:

$$\nabla f(\mathbf{x}) = \mathrm{J}_{\psi^{-1}}(\mathbf{x})^\top \nabla h(\mathbf{z}) = \mathrm{J}_\psi^{-1}(\mathbf{z})^\top \nabla h(\mathbf{z}).$$

- Chain rule for computing Hessian of $h = f \circ \psi$:

$$\nabla^2 h(\mathbf{z}) = \mathrm{J}_\psi(\mathbf{z})^\top \nabla^2 f(\psi(\mathbf{z})) \mathrm{J}_\psi(\mathbf{z}) + \sum_{i=1}^n \frac{\partial f}{\partial \mathbf{x}_i}(\psi(\mathbf{z})) \nabla^2 \psi_i(\mathbf{z}).$$

# D   Landscape Analysis

In this section, we provide landscape analysis to understand important relationships between problem **P** and **H**.

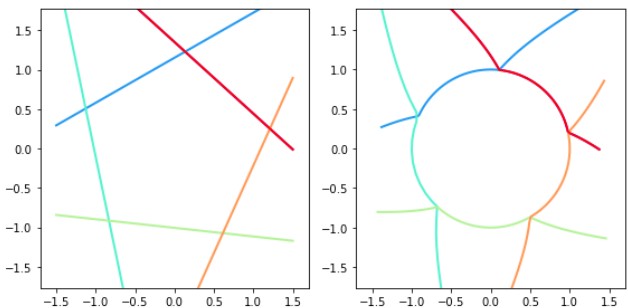

Figure 7: Illustration of the action of homeomorphism on a polyhedron. The left figure is the polyhedral constraints of problem $\mathbf{P}$. Each color of line represents a constraint inequality $\{\mathbf{a}_i^\top \mathbf{x} \leq b_i\}$ for some $i$. Under the homeomorphic mapping $\boldsymbol{\psi}$, the constrained set is transformed to a ball (right figure). Each constraint inequality $\{G_i(\mathbf{z}) \leq 0\}$ (colored differently) is non-convex in general.

### D.1  Action of Homeomorphism on a Constrained Set

Recall that the constrained set is $\mathcal{K} = \{\mathbf{x} \in \mathbb{R}^n \mid \mathbf{g}(\mathbf{x}) \leq 0\}$ with $\mathbf{g} = (g_1, g_2, \cdots, g_m)$ where $g_i$ ($i = 1, 2, \cdots, m$) is a convex function. For problem $\mathbf{H}$,

$$\mathcal{B} = \boldsymbol{\psi}^{-1}(\mathcal{K}) = \{\mathbf{z} \in \mathbb{R}^n \mid \boldsymbol{\psi}(\mathbf{z}) \in \mathcal{K}\} = \{\mathbf{z} \in \mathbb{R}^n \mid \mathbf{G}(\mathbf{z}) := \mathbf{g}(\boldsymbol{\psi}(\mathbf{z})) \leq \mathbf{0}\}$$

where $G_i$ might be non-convex even for convex $g_i$. However, $\mathcal{B}$ is assumed to be convex (actually a ball set) in this paper. One can refer to Fig. 7 for an illustration.

Moreover, we assume there are no redundant inequalities in $\mathcal{K}$, i.e., there is no $g_i$ such that it can be represented as the positive linear combination of other inequalities. In this case, any feasible point $\mathbf{x}$ satisfying $g_i(\mathbf{x}) = 0$ for some $i$ is on the boundary of the set $\mathcal{K}$. Moreover, we have

$$\{\mathbf{x} \in \mathcal{K} \mid g_j(\mathbf{x}) = 0, g_k(\mathbf{x}) \neq 0\} \bigcap \{\mathbf{x} \in \mathcal{K} \mid g_k(\mathbf{x}) = 0, g_j(\mathbf{x}) \neq 0\} = \emptyset$$

for any $k \neq j$. Note $\mathcal{B} = \{\mathbf{z} \mid G_i(\mathbf{z}) \leq 0, i = 1, 2, \cdots, m\} = \{\mathbf{z} \mid \|\mathbf{z}\|^2 \leq 1\}$. Moreover, $\{G_i(\mathbf{z}) \leq 0, i = 1, 2, \cdots, m\}$ also has no redundant constraints by the non-singularity of the Jacobian of $\boldsymbol{\psi}$ and similarly,

$$\{\mathbf{z} \in \mathcal{B} \mid G_j(\mathbf{z}) = 0, G_k(\mathbf{z}) \neq 0\} \bigcap \{\mathbf{z} \in \mathcal{B} \mid G_k(\mathbf{z}) = 0\} = \emptyset$$

for any $j \neq k$. Hence if $\mathbf{z} \in \mathcal{B}$ satisfies $G_i(\mathbf{z}) = 0$ for some $i$, it lies on the boundary of $\mathcal{B}$. Clearly, we have

$$G_i(\mathbf{z}) = \|\mathbf{z}\|^2 - 1 \quad \text{at} \quad \mathbf{z}' \in \mathcal{B}, G_i(\mathbf{z}') = 0, \tag{2}$$

and

$$\nabla G_i(\mathbf{z}) = 2\mathbf{z}, \nabla^2 G_i(\mathbf{z}) = 2\mathbf{I}_n \quad \text{at} \quad \mathbf{z}' \in \mathcal{B}, G_i(\mathbf{z}') = 0. \tag{3}$$

where $\mathbf{I}_n$ is the identity matrix of $n$ by $n$.

### D.2  Properties of Function $h = f \circ \boldsymbol{\psi}$

**Lemma D.1** (Properties of $h = f \circ \boldsymbol{\psi}$). *Under the general assumptions C.2, $h = f \circ \boldsymbol{\psi}$ has the following properties.*

1) *$h$ is $L_{h,0} := L_{f,0}\kappa_2$ Lipschitz continuous.*

2) *$h$ is $L_h$-smooth with $L_h = \kappa_2^2 L_f + L_{\boldsymbol{\psi}} L_{f,0}$.*

3) *If $f$ is convex, then $h$ is $\ell_h$-weakly convex with $\ell_h = L_{f,0} L_{\boldsymbol{\psi}}$.*

4) *If $f$ is $\mu_f$-strongly convex. Then $h$ satisfies the quadratic growth condition with $m_h = \frac{\mu_f \kappa_1}{2}$ on the ball set $\mathcal{B}$, i.e.,*

$$h(\mathbf{z}) - h^* \geq m_h \|\mathbf{z} - \mathbf{z}^*\|^2, \ \forall \mathbf{z} \in \mathcal{B}. \tag{4}$$

*Proof.* We prove them one by one in the following.

1) We can directly derive from basic definitions:

$$\|h(\mathbf{u}) - h(\mathbf{v})\| \leq \|f(\boldsymbol{\psi}(\mathbf{u})) - f(\boldsymbol{\psi}(\mathbf{v}))\|$$
$$\leq L_{f,0} \|\boldsymbol{\psi}(\mathbf{u}) - \boldsymbol{\psi}(\mathbf{v})\|$$
$$\leq L_{f,0} L_{\boldsymbol{\psi}} \|\mathbf{u} - \mathbf{v}\|.$$

2) From $L_f$-smoothness of $f$, we have

$$\|\nabla f(\mathbf{x}) - \nabla f(\mathbf{y})\| \leq L_f \|\mathbf{x} - \mathbf{y}\|. \tag{5}$$

Then we derive with $\mathbf{x} = \boldsymbol{\psi}(\mathbf{z}), \mathbf{v} = \boldsymbol{\psi}(\mathbf{y})$,

$$\|\nabla h(\mathbf{z}) - \nabla h(\mathbf{v})\| = \left\| \mathrm{J}_{\boldsymbol{\psi}}(\mathbf{z})^\top \nabla f(\mathbf{x}) - \mathrm{J}_{\boldsymbol{\psi}}(\mathbf{v})^\top \nabla f(\mathbf{y}) \right\|$$
$$= \left\| \mathrm{J}_{\boldsymbol{\psi}}(\mathbf{z})^\top (\nabla f(\mathbf{x}) - \nabla f(\mathbf{y})) + (\mathrm{J}_{\boldsymbol{\psi}}(\mathbf{z}) - \mathrm{J}_{\boldsymbol{\psi}}(\mathbf{v}))^\top \nabla f(\mathbf{y}) \right\|$$
$$\leq \left\| \mathrm{J}_{\boldsymbol{\psi}}(\mathbf{z})^\top (\nabla f(\mathbf{x}) - \nabla f(\mathbf{y})) \right\| + \left\| (\mathrm{J}_{\boldsymbol{\psi}}(\mathbf{z}) - \mathrm{J}_{\boldsymbol{\psi}}(\mathbf{v}))^\top \nabla f(\mathbf{y}) \right\|$$
$$\leq \kappa_2 L_f \|\boldsymbol{\psi}(\mathbf{z}) - \boldsymbol{\psi}(\mathbf{v})\| + L_{\boldsymbol{\psi}} L_{f,0} \|\mathbf{z} - \mathbf{v}\|$$
$$\leq \left( \kappa_2^2 L_f + L_{\boldsymbol{\psi}} L_{f,0} \right) \|\mathbf{z} - \mathbf{v}\|.$$

Let $L_h = \kappa_2^2 L_f + L_{\boldsymbol{\psi}} L_{f,0}$. We have the conclusion.

3) One hope to show $h(\cdot) + \frac{\ell_h}{2} \| \cdot + \mathbf{v}\|^2$ is a convex function, i.e.,

$$h(\mathbf{v}) + \frac{\ell_h}{2} \|\mathbf{v}\|^2 \geq h(\mathbf{z}) + \frac{\ell_h}{2} \|\mathbf{z}\|^2 + \langle \nabla h(\mathbf{z}) + \ell_h \mathbf{z}, \mathbf{v} - \mathbf{z} \rangle, \ \forall \mathbf{z}, \mathbf{v}.$$

This is equivalent to show

$$h(\mathbf{v}) + \frac{\ell_h}{2} \|\mathbf{v} - \mathbf{z}\|^2 \geq h(\mathbf{z}) + \langle \nabla h(\mathbf{z}), \mathbf{v} - \mathbf{z} \rangle, \ \forall \mathbf{z}, \mathbf{v}.$$

We drive with $\mathbf{x} = \boldsymbol{\psi}(\mathbf{z}), \mathbf{y} = \boldsymbol{\psi}(\mathbf{v})$ as follows,

$$\langle \nabla h(\mathbf{z}), \mathbf{v} - \mathbf{z} \rangle = \langle \nabla \mathrm{J}_{\boldsymbol{\psi}}(\mathbf{z})^\top f(\mathbf{x}), \mathbf{v} - \mathbf{z} \rangle$$
$$= \langle \nabla f(\mathbf{x}), \mathrm{J}_{\boldsymbol{\psi}}(\mathbf{z})(\mathbf{v} - \mathbf{z}) \rangle$$
$$= \langle \nabla f(\mathbf{x}), -\boldsymbol{\psi}(\mathbf{v}) + \boldsymbol{\psi}(\mathbf{z}) + \mathrm{J}_{\boldsymbol{\psi}}(\mathbf{z})(\mathbf{v} - \mathbf{z}) \rangle + \langle \nabla f(\mathbf{x}), \boldsymbol{\psi}(\mathbf{v}) - \boldsymbol{\psi}(\mathbf{z}) \rangle$$
$$\leq \|\nabla f(\mathbf{x})\| \cdot \|\boldsymbol{\psi}(\mathbf{v}) - \boldsymbol{\psi}(\mathbf{z}) - \mathrm{J}_{\boldsymbol{\psi}}(\mathbf{z})(\mathbf{v} - \mathbf{z})\| + \langle \nabla f(\mathbf{x}), \mathbf{y} - \mathbf{x} \rangle$$
$$\leq L_{f,0} L_{\boldsymbol{\psi}} \|\mathbf{z} - \mathbf{v}\|^2 + f(\mathbf{y}) - f(\mathbf{x})$$
$$= L_{f,0} L_{\boldsymbol{\psi}} \|\mathbf{z} - \mathbf{v}\|^2 + h(\mathbf{v}) - h(\mathbf{z})$$

where the first inequality is from triangular inequality and the second inequality is from Lemma C.3 and the convexity of $f$.

4) Note that optimality implies stationarity. By the strong convexity of $f$, for any $\mathbf{x} \in \mathcal{K}$ and $\mathbf{x}^* \in \mathcal{K}^*$, we have

$$f(\mathbf{x}) \geq f(\mathbf{x}^*) + \langle \nabla f(\mathbf{x}^*), \mathbf{x} - \mathbf{x}^* \rangle + \frac{\mu_f}{2} \|\mathbf{x} - \mathbf{x}^*\|^2 \geq f(\mathbf{x}^*) + \frac{\mu_f}{2} \|\mathbf{x} - \mathbf{x}^*\|^2.$$

Here the last inequality is from the global optimality condition C.1.

With $\mathbf{x} = \boldsymbol{\psi}(\mathbf{z})$, we have

$$h(\mathbf{z}) \geq h(\mathbf{z}^*) + \frac{\mu_f}{2} \|\boldsymbol{\psi}(\mathbf{z}) - \boldsymbol{\psi}(\mathbf{z}^*)\|^2 \geq h(\mathbf{z}^*) + \frac{\mu_f}{2} \kappa_1 \|\mathbf{z} - \mathbf{z}^*\|^2.$$

$\square$

## D.3 Stationary Points and KKT Points

Recall that the stationary point $\mathbf{x}^\star$ of a set $\mathcal{K}$ is defined by the variational inequality:

$$\langle \nabla f(\mathbf{x}^*), \mathbf{x} - \mathbf{x}^\star \rangle \geq 0, \forall \mathbf{x} \in \mathcal{K}. \tag{6}$$

Besides, a tuple $(\mathbf{x}, \boldsymbol{\lambda})$ is said to satisfy the Karush–Kuhn–Tucker (KKT) condition of problem $\mathbf{P}$ if the following holds

$$
\begin{aligned}
\nabla f(\mathbf{x}) + \sum_{i=1}^{m} \lambda_i \nabla g_i(\mathbf{x}) &= \mathbf{0}, \\
g_i(\mathbf{x}) &\leq 0, \quad i = 1, 2, \cdots, m \\
\boldsymbol{\lambda} \geq \mathbf{0}, \; \lambda_i g_i(\mathbf{x}) &= 0, \quad i = 1, 2, \cdots, m
\end{aligned}
\tag{7}
$$

where $\boldsymbol{\lambda}$ is the dual variable corresponding to inequality constraints.

**Definition D.2** (KKT stationary point). A point $\mathbf{x}^*$ is said to be a KKT stationary point of $\mathbf{P}$ if there exists $\boldsymbol{\lambda}^*$ such that $(\mathbf{x}^*, \boldsymbol{\lambda}^*)$ satisfies KKT condition (7).

Actually, these two definitions are equivalent under mild conditions, which will be discussed later. Before moving on, we introduce related definitions.

**Definition D.3** (Constraints qualification). Let $\bar{\mathbf{x}}$ be feasible for a problem with constraint set

$$\{\mathbf{x} \mid g_i(\mathbf{x}) \leq 0, q_j(\mathbf{x}) = 0, \text{ for } i = 1, 2, \cdots, m, \text{ and } j = 1, 2, \cdots, p\}$$

and put $\mathcal{A}(\bar{\mathbf{x}}) := \{i \mid g_i(\bar{\mathbf{x}}) = 0\}$. We say that

1) the linear independence constraint qualification (LICQ) holds at $\bar{\mathbf{x}}$ (and write $\mathrm{LICQ}(\bar{\mathbf{x}})$ ) if the gradients

$$\nabla g_i(\bar{\mathbf{x}})(i \in \mathcal{A}(\bar{\mathbf{x}})), \quad \nabla q_j(\bar{\mathbf{x}})(j = 1, \ldots, p)$$

are linearly independent.

2) the Mangasarian-Fromovitz constraint qualification (MFCQ) holds at $\bar{\mathbf{x}}$ (and write $\mathrm{MFCQ}(\bar{\mathbf{x}})$ ) if the gradients

$$\nabla q_j(\bar{\mathbf{x}})(j = 1, \ldots, p)$$

are linearly independent and there exists a vector $d \in \mathbb{R}^n$ such that

$$\nabla g_i(\bar{\mathbf{x}})^\top d < 0(i \in \mathcal{A}(\bar{\mathbf{x}})), \quad \nabla q_j(\bar{\mathbf{x}})^\top d = 0(j = 1, \ldots, p).$$

**Definition D.4** (Strict complementary slackness). It is said that the strict complementary slackness condition holds for problem $\mathbf{P}$, if

$$\lambda_i^* > 0 \quad \text{for} \quad g_i(\mathbf{x}^*) = 0,$$

and for problem $\mathbf{H}$, if

$$\nu_* > 0 \quad \text{for} \quad \|\mathbf{z}^*\| = 1.$$

**Definition D.5** (Slater's condition). It is said that the Slater's condition holds for a problem with constrained set $\mathcal{K} := \{g_i(\mathbf{x}) \leq 0, i = 1, 2, \cdots, m\}$, if there exists $\mathbf{x}_0$ such that for all $i = 1, 2, \cdots, m$,

$$g_i(\mathbf{x}_0) < 0.$$

It is established that the variational inequality (6) and the KKT system (7) coincide with a strong relationship if certain constraints qualification holds. One could refer to [HP90, FFK98, Tob86] for detailed discussion. We summarize the conclusion in the following proposition.

**Proposition D.6.** *We have the following relationship between (6) and (7).*

*1) If $\mathbf{x}^*$ is a solution of the variational inequality (6) and a certain constraint qualification (e.g., LICQ, MFCQ, see Def. D.3) holds at $\mathbf{x}^*$, then there exists $\boldsymbol{\lambda}^*$ such that the tuple $(\mathbf{x}^*, \boldsymbol{\lambda}^*)$ satisfy the KKT system (7).*

*2) Suppose $f$ and $g_i$ $(i = 1, 2, \cdots, m)$ are convex and Slater's condition D.5 is satisfied. If $\mathbf{x}^*$ is a solution of the variational inequality (6), then there exists $\boldsymbol{\lambda}^*$ such that the tuple $(\mathbf{x}^*, \boldsymbol{\lambda}^*)$ satisfy the KKT system (7).*

*3) If $(\mathbf{x}^*, \boldsymbol{\lambda}^*)$ satisfies the KKT system (7) and $g$ is convex, then $\mathbf{x}^*$ satisfy the variational inequalities (6).*

*Proof.* One can refer to e.g. [Tob86, HP90] for item (1) and (3) and refer to e.g. Theorem 3.78 [Bec17] for item (2). $\qquad \square$

## D.4 KKT Conditions of Problem P and H

We first introduce the following definition.

- The normal cone $N_S(\mathbf{x})$ of a closed and convex set $\mathcal{K}$ at $\mathbf{x} \in \mathcal{K}$ is defined as
$$N_\mathcal{K}(\mathbf{x}) = \{\mathbf{y} : \langle \mathbf{y}, \mathbf{z} - \mathbf{x} \rangle \le 0 \text{ for any } \mathbf{z} \in \mathcal{K}\}.$$

- The tangent cone $T_\mathcal{K}(\mathbf{x})$ of a closed and convex set $\mathcal{K}$ at $\mathbf{x} \in \mathcal{K}$ is defined as
$$T_\mathcal{K}(\mathbf{x}) = \text{cl}\{\mathbf{y} : \exists \lambda > 0 \text{ s.t } \mathbf{x} + \lambda \mathbf{y} \in \mathcal{K}\}$$
where $\text{cl}(\cdot)$ denotes the closure of a set.

- The critical cone $C_\mathcal{K}(\mathbf{x})$ of a closed and convex set $\mathcal{K}$ at $\mathbf{x} \in \mathcal{K}$ with related to a objective function $f$ is defined as
$$C_\mathcal{K}(\mathbf{x}) = \left\{\mathbf{d} \in T_\mathcal{K}(\mathbf{x}) : \nabla f(\mathbf{x})^\top \mathbf{d} = 0\right\}.$$

To define the second-order KKT condition for the optimization problems, we first recall that the critical cone of problem $\mathbf{P}$ can be written as [NW99]

$$\mathbf{d} \in C_\mathcal{K}(\mathbf{x}^*) \Leftrightarrow \begin{cases} \nabla g_i(\mathbf{x}^*)^\top \mathbf{d} = 0, & \text{for all } i \in \mathcal{A}(\mathbf{x}^*) \text{ with } \lambda_i^* > 0, \\ \nabla g_i(\mathbf{x}^*)^\top \mathbf{d} \ge 0, & \text{for all } i \in \mathcal{A}(\mathbf{x}^*) \text{ with } \lambda_i^* = 0. \end{cases}$$

Here $\boldsymbol{\lambda}^*$ is the Lagrangian multiplier of inequality constraints $g_i$ and $\mathcal{A}(\mathbf{x}^*)$ is the index of active constraints. Moreover, if *strict complementary slackness* holds, the critical cone is simplified as

$$C_\mathcal{K}(\mathbf{x}^*) = \{\mathbf{d} \in \mathbb{R}^n \mid \nabla g_i(\mathbf{x}^*)^\top \mathbf{d} = 0, \text{ for all } i \in \mathcal{A}(\mathbf{x}^*)\}.$$

Suppose *strict complementary slackness* holds for problem $\mathbf{P}$ and $\mathbf{H}$. Then, we can write KKT conditions for problem $\mathbf{P}$ and $\mathbf{H}$ in the following.

*First-order KKT conditions* on $\mathbf{x}^*$. The Lagrangian of $\mathbf{P}$ is

$$\mathcal{L}_\mathrm{P}(\mathbf{x}, \boldsymbol{\lambda}) = f(\mathbf{x}) + \sum_{i=1}^m \lambda_i g_i(\mathbf{x}).$$

The first-order KKT conditions of $\mathbf{P}$ are: there exists $\boldsymbol{\lambda}^*$ such that

$$\nabla f(\mathbf{x}^*) + \sum_{i=1}^m \lambda_i^* \nabla g_i(\mathbf{x}^*) = \mathbf{0}, \tag{8a}$$

$$g_i(\mathbf{x}^*) \le 0, \quad i = 1, 2, \cdots, m \tag{8b}$$

$$\boldsymbol{\lambda}^* \ge \mathbf{0}, \ \lambda_i^* g_i(\mathbf{x}^*) = 0, \quad i = 1, 2, \cdots, m. \tag{8c}$$

*Second-order KKT conditions* on $\mathbf{x}^*$. It adds the following condition

$$\mathbf{w}^\top \nabla_\mathbf{x}^2 \mathcal{L}_\mathrm{P}(\mathbf{x}^*, \boldsymbol{\lambda}^*) \mathbf{w} \ge 0 \tag{9}$$

for any $\mathbf{w}$ satisfying $\mathbf{w}^\top \nabla g_i(\mathbf{x}^*) = 0$ with $i \in \mathcal{A}(\mathbf{x}^*)$.

*First-order KKT conditions* on $\mathbf{z}^*$. The Lagrangian of $\mathbf{H}$ is

$$\mathcal{L}_\mathrm{H}(\mathbf{z}, \nu) = h(\mathbf{z}) + \nu(\|\mathbf{z}\|^2 - 1).$$

The first-order KKT conditions of $\mathbf{H}$ are: there exists $\nu^*$ such that

$$\nabla h(\mathbf{z}^*) + 2\nu^* \mathbf{z}^* = \mathbf{0}, \tag{10a}$$

$$\|\mathbf{z}^*\|^2 \le 1, \tag{10b}$$

$$\nu^* \ge 0, \ \nu^*(\|\mathbf{z}^*\|^2 - 1) = 0. \tag{10c}$$

*Second-order KKT condition* on $\mathbf{z}^*$. It will add the following condition.

$$\mathbf{d}^\top \nabla_\mathbf{z}^2 \mathcal{L}_\mathrm{H}(\mathbf{z}^*, \nu^*) \mathbf{d} \ge 0 \tag{11}$$

for any $\mathbf{d} \in C_\mathcal{B}(\mathbf{z}^*)$. Here recall that

$$C_\mathcal{B}(\mathbf{z}^*) = \begin{cases} \mathbb{R}^n, & \text{if } \mathbf{z}^* \in \text{int}(\mathcal{B}), \\ \{\mathbf{d} : \mathbf{d}^\top \mathbf{z}^* = 0\}, & \text{if } \mathbf{z}^* \in \partial\mathcal{B}. \end{cases}$$

## D.5 Relationships of KKT Points between Problem P and H

**Lemma D.7.** *We have that $\mathbf{x}^*$ is a KKT stationary point of $\mathbf{P}$ if and only if $\mathbf{z}^*$ is also a KKT stationary point of $\mathbf{H}$.*

*Proof.* 1) First, we assume that $\mathbf{z}^*$ is a KKT stationary point of $\mathbf{P}$. By assumption, there exists $\boldsymbol{\lambda}^*$ such that the KKT condition holds (8) holds. Then we have

$$\mathbf{J}_{\boldsymbol{\psi}}(\mathbf{z}^*)^\top \nabla f(\mathbf{x}^*) + \sum_{i=1}^m \lambda_i^* \mathbf{J}_{\boldsymbol{\psi}}(\mathbf{z}^*)^\top \nabla g_i(\mathbf{x}^*) = \mathbf{0},$$

$$g_i(\boldsymbol{\psi}(\mathbf{z}^*)) \leq 0, \quad i = 1, 2, \cdots, m$$

$$\boldsymbol{\lambda}^* \geq \mathbf{0}, \ \lambda_i^* g_i(\boldsymbol{\psi}(\mathbf{z}^*)) = 0, \quad i = 1, 2, \cdots, m.$$

This is equivalent to

$$\nabla h(\mathbf{z}^*) + \sum_{i=1}^m \lambda_i^* \nabla G_i(\mathbf{z}^*) = \mathbf{0}, \tag{12a}$$

$$G_i(\mathbf{z}^*) \leq 0, \quad i = 1, 2, \cdots, m \tag{12b}$$

$$\boldsymbol{\lambda}^* \geq \mathbf{0}, \ \lambda_i^* G_i(\mathbf{z}^*) = 0, \quad i = 1, 2, \cdots, m. \tag{12c}$$

Let $\nu^* = \sum_{i=1} \lambda_i^*$. According to the eq. (2,3), eq. (12a) is actually

$$\nabla h(\mathbf{z}^*) + 2\nu^* \mathbf{z}^* = \mathbf{0}.$$

By assumption, eq. (12b) is equivalent to

$$\|\mathbf{z}^*\|^2 \leq 1.$$

Note that if $G_i(\mathbf{z}^*) < 0$ for all $i$, then $\boldsymbol{\lambda}^* = 0$ and thus $\nu^* = 0$. In this case, $\nu^*(\|\mathbf{z}^*\|^2 - 1) = 0$. If $\mathbf{z}^*$ makes at least one $G_i(\mathbf{z}^*) = 0$, then we have $\|\mathbf{z}^*\|^2 = 1$. In this case, we also have $\nu^*(\|\mathbf{z}^*\|^2 - 1) = 0$. Hence, eq. (12c) implies

$$\nu^* \geq 0, \nu^*(\|\mathbf{z}^*\|^2 - 1) = 0.$$

In conclusion, there exists $\mathbf{z}^*, \nu^*$ such that the KKT condition holds.

2) Now, we assume $\mathbf{z}^*, \nu^*$ satisfy KKT condition for problem $\mathbf{H}$, i.e.,

$$\nabla h(\mathbf{z}^*) + 2\nu^* \mathbf{z}^* = \mathbf{0},$$

$$\|\mathbf{z}^*\|^2 \leq 1,$$

$$\nu^* \geq 0, \ \nu^*(\|\mathbf{z}^*\|^2 - 1) = 0.$$

If $\mathbf{z}^* \in \text{int}(\mathcal{B})$, then $G_i(\mathbf{z}^*) < 0$ for all $i$ and $\nu^* = 0$. In this case, there exists $\boldsymbol{\lambda}^* = 0$ such that the KKT condition (10) of problem $\mathbf{P}$ holds at $\mathbf{x}^* = \boldsymbol{\psi}(\mathbf{z}^*), \boldsymbol{\lambda}^*$.

If $\mathbf{z}^* \in \partial\mathcal{B}$, then there exists at least one $i \in \{1, 2, \cdots, m\}$ such that $G_i(\mathbf{z}^*) = 0$ and $\nu^* > 0$ from strict complementary slackness. Denote $\mathcal{A} = \{i : G_i(\mathbf{z}^*) = 0\}$. Note we define $\lambda_i^* = 0$ if $i \notin \mathcal{A}$ and $\lambda_i^* = \nu^*/|\mathcal{A}|$. Then we have $\mathbf{z}^*, \boldsymbol{\lambda}^*$ such that eq. 12 holds which implies $\mathbf{x}^* = \boldsymbol{\psi}(\mathbf{z}^*), \boldsymbol{\lambda}^*$ make the KKT condition of problem $\mathbf{P}$ hold.

$\square$

**Lemma D.8.** *Suppose strict complementary slackness condition holds for both problem $\mathbf{P}$ and $\mathbf{H}$. Then $\mathbf{x}^*$ is a second-order KKT stationary point of $\mathbf{P}$ if and only if $\mathbf{z}^* = \boldsymbol{\psi}^{-1}(\mathbf{x}^*)$ is also a second-order KKT stationary point of $\mathbf{H}$.*

*Proof.* From Lemma D.7, there exists $\boldsymbol{\lambda}^*$ and $\nu^*$ such that $(\mathbf{x}^*, \boldsymbol{\lambda}^*)$ holds for first-order KKT condition of $\mathbf{P}$ if and only if $(\mathbf{z}^*, \nu^*)$ holds for first-order KKT condition of $\mathbf{H}$. Hence, it suffices to show the equivalence of condition 11 and 9.

1) Let's first suppose $\mathbf{x}^*$ is a second-order KKT stationary point, i.e., eq. (9) holds.

Note

$$\nabla^2_{\mathbf{z}} \mathcal{L}_{\mathrm{H}}(\mathbf{z}^*, \nu^*) = \nabla^2 h(\mathbf{z}^*) + 2\nu^* \mathbf{I}_n,$$

where $\mathbf{I}_n$ is identity matrix of size $n \times n$. We just need to show $\mathbf{d}^\top \nabla \mathcal{L}_\mathrm{H}(\mathbf{z}^*, \nu^*)\mathbf{d} \geq 0$ for any $\mathbf{d} \in \mathrm{C}_\mathcal{B}(\mathbf{z}^*)$. Recall that

$$\nabla^2 h(\mathbf{z}^*) = \mathrm{J}_{\boldsymbol{\psi}}(\mathbf{z}^*)^\top \nabla^2 f(\boldsymbol{\psi}(\mathbf{z}^*))\mathrm{J}_{\boldsymbol{\psi}}(\mathbf{z}^*) + \sum_{i=1}^{n} \frac{\partial f}{\partial \mathbf{x}_i}(\boldsymbol{\psi}(\mathbf{z}^*))\nabla^2 \psi_i(\mathbf{z}^*),$$

and

$$\nabla^2 G_i(\mathbf{z}^*) = \mathrm{J}_{\boldsymbol{\psi}}(\mathbf{z}^*)^\top \nabla^2 g_i(\boldsymbol{\psi}(\mathbf{z}^*))\mathrm{J}_{\boldsymbol{\psi}}(\mathbf{z}^*) + \sum_{k=1}^{n} \frac{\partial g_i}{\partial \mathbf{x}_k}(\boldsymbol{\psi}(\mathbf{z}^*))\nabla^2 \psi_k(\mathbf{z}^*), \quad i = 1, 2, \cdots, m.$$

From eq. (2), note that

$$\nabla^2 G_k(\mathbf{z}^*) = 2\mathbf{I}_n, \forall k \in \mathcal{A}(\mathbf{x}^*) \cap \{k : G_k(\mathbf{z}^*) = 0\}.$$

From Lemma D.7, $\nu^* = \sum_i \lambda_i^*$. Then we have

$$\nabla^2 \mathcal{L}_\mathrm{H}(\mathbf{z}^*, \nu^*) = \nabla^2 h(\mathbf{z}^*) + \sum_{i=1}^{m} \lambda_i^* \nabla^2 G_i(\mathbf{z}^*) \tag{13a}$$

$$= \mathrm{J}_{\boldsymbol{\psi}}(\mathbf{z}^*)^\top \nabla^2 f(\boldsymbol{\psi}(\mathbf{z}^*))\mathrm{J}_{\boldsymbol{\psi}}(\mathbf{z}^*) + \sum_{i=1}^{m} \mathrm{J}_{\boldsymbol{\psi}}(\mathbf{z}^*)^\top \lambda_i^* \nabla^2 g_i(\boldsymbol{\psi}(\mathbf{z}^*))\mathrm{J}_{\boldsymbol{\psi}}(\mathbf{z}^*) \tag{13b}$$

$$+ \sum_{k=1}^{n} \frac{\partial f}{\partial \mathbf{x}_k}(\boldsymbol{\psi}(\mathbf{z}^*))\nabla^2 \psi_k(\mathbf{z}^*) + \sum_{k=1}^{n}\sum_{i=1}^{m} \lambda_i^* \frac{\partial g_i}{\partial \mathbf{x}_k}(\boldsymbol{\psi}(\mathbf{z}^*))\nabla^2 \psi_k(\mathbf{z}^*). \tag{13c}$$

From first-order KKT stationarity of $\mathbf{P}$, i.e.,

$$\nabla f(\mathbf{x}^*) + \sum_{i=1}^{m} \lambda_i^* \nabla g_i(\mathbf{x}^*) = \mathbf{0},$$

We have

$$\frac{\partial f}{\partial \mathbf{x}_k}(\mathbf{x}^*) + \sum_{i=1}^{m} \lambda_i^* \frac{\partial g_i}{\partial \mathbf{x}_k}(\mathbf{x}^*) = 0.$$

Hence for any $\mathbf{d} \in \mathrm{C}_\mathcal{B}(\mathbf{z}^*)$, we have the second term (13c) is equal to 0.

Now we note it's trivial that $\mathrm{C}_\mathcal{K}(\mathbf{x}^*) = \mathrm{C}_\mathcal{B}(\mathbf{z}^*) = \mathbb{R}^n$ if $\mathbf{z}^* \in \mathrm{int}(\mathcal{K})$ where $\mathbf{x}^* = \boldsymbol{\psi}(\mathbf{z}^*)$. Hence in this case if $\mathbf{d} \in \mathrm{C}_\mathcal{B}(\mathbf{z}^*)$, we will have $\mathrm{J}_{\boldsymbol{\psi}}(\mathbf{z}^*)\mathbf{d} \in \mathrm{C}_\mathcal{K}(\mathbf{x}^*)$

If $\mathbf{x}^* \in \partial\mathcal{K}$, then $\mathcal{A}(\mathbf{x}^*) \neq \emptyset$. For $\mathbf{d} \in \mathrm{C}_\mathcal{B}(\mathbf{z}^*)$, i.e., $\mathbf{d}^\top\mathbf{z}^* = 0$, we have

$$(\mathrm{J}_{\boldsymbol{\psi}}(\mathbf{z}^*)\mathbf{d})^\top \nabla g_i(\mathbf{x}^*) = \mathbf{d}^\top \mathrm{J}_{\boldsymbol{\psi}}(\mathbf{z}^*)^\top \nabla g_i(\mathbf{x}^*) = \mathbf{d}^\top G_i(\mathbf{z}^*) = 2\mathbf{d}^\top\mathbf{z}^* = 0, \quad \text{for } i \in \mathcal{A}(\mathbf{x}^*),$$

or $\mathrm{J}_{\boldsymbol{\psi}}(\mathbf{z}^*)\mathbf{d} \in \mathrm{C}_\mathcal{K}(\mathbf{x}^*)$.

So for $\mathbf{d} \in \mathrm{C}_\mathcal{B}(\mathbf{z}^*)$, we have the following holds about the first term of $\nabla^2 \mathcal{L}_\mathrm{H}(\mathbf{z}^*, \nu^*)$.

$$(\mathrm{J}_{\boldsymbol{\psi}}(\mathbf{z}^*)\mathbf{d})^\top \nabla^2 f(\boldsymbol{\psi}(\mathbf{z}^*))\mathrm{J}_{\boldsymbol{\psi}}(\mathbf{z}^*)\mathbf{d} + (\mathrm{J}_{\boldsymbol{\psi}}(\mathbf{z}^*)\mathbf{d})^\top \left(\sum_{i=1}^{m} \lambda_i^* \nabla^2 g_i(\boldsymbol{\psi}(\mathbf{z}^*))\right)\mathrm{J}_{\boldsymbol{\psi}}(\mathbf{z}^*)\mathbf{d} \geq 0$$

where the last '$\geq$' is from the assumption that $\mathbf{x}^*$ is the second-order KKT stationary point of $\mathbf{P}$. Hence, we have $\mathbf{d}^\top \nabla^2 \mathcal{L}_\mathrm{H}(\mathbf{z}^*, \nu^*)\mathbf{d} \geq 0$ for any $\mathbf{d} \in \mathrm{C}_\mathcal{B}(\mathbf{z}^*)$, i.e., $\mathbf{z}^* = \boldsymbol{\psi}^{-1}(\mathbf{x}^*)$ is also a second-order KKT stationary point.

2) Let's suppose $\mathbf{z}^*$ is a second-order KKT stationary point and show that $\mathbf{x}^*$ is a second-order KKT stationary point.

If $\mathbf{z}^* \in \mathrm{int}(\mathcal{B})$, the proof is trivial because $\nu^* = 0$ according to the similar analysis. So we assume $\mathbf{z}^* \in \partial\mathcal{B}$. Define $\mathcal{A}(\mathbf{z}^*) = \{i : G_i(\mathbf{z}^*) = 0\}$, and $\lambda_i^* = 0$ for $i \notin \mathcal{A}(\mathbf{z}^*)$, $\lambda_i^* = \nu^*/|\mathcal{A}(\mathbf{z}^*)|$ for $i \in \mathcal{A}(\mathbf{z}^*)$.

Note for any $\mathbf{w} \in \mathrm{C}_\mathcal{K}(\mathbf{x}^*)$, we have

$$0 = \mathbf{w}^\top \nabla g_i(\mathbf{x}^*) = \mathbf{w}^\top \mathrm{J}_{\boldsymbol{\psi}}^{-1}(\mathbf{z}^*)^\top \nabla G_i(\mathbf{z}^*) = (\mathrm{J}_{\boldsymbol{\psi}}^{-1}(\mathbf{z}^*)\mathbf{w})^\top \mathbf{z}^*, \text{ for } i \in \mathcal{A}(\mathbf{x}^*) = \mathcal{A}(\mathbf{z}^*).$$

Hence $J_{\psi}^{-1}(\mathbf{z}^*)\mathbf{w} \in C_{\mathcal{B}}(\mathbf{z}^*)$. Then for any $\mathbf{w} \in C_{\mathcal{K}}(\mathbf{x}^*)$,

$$\mathbf{w}^\top \nabla_{\mathbf{x}}^2 \mathcal{L}_{\mathrm{P}}(\mathbf{x}^*, \boldsymbol{\lambda}^*)\mathbf{w}$$

$$= \mathbf{w}^\top \nabla^2 f(\mathbf{x}^*)\mathbf{w} + \mathbf{w}^\top \sum_{i=1}^m \lambda_i^* \nabla^2 g_i(\mathbf{x}^*)\mathbf{w}$$

$$= (J_{\psi}^{-1}(\mathbf{z}^*)\mathbf{w})^\top J_{\psi}(\mathbf{z}^*)\nabla^2 f(\mathbf{x}^*)J_{\psi}(\mathbf{z}^*)J_{\psi}^{-1}(\mathbf{z}^*)\mathbf{w}$$

$$+ (J_{\psi}^{-1}(\mathbf{z}^*)\mathbf{w})^\top J_{\psi}(\mathbf{z}^*)(\sum_{i=1}^m \lambda_i^* \nabla^2 g_i(\mathbf{x}^*))J_{\psi}(\mathbf{z}^*)J_{\psi}^{-1}(\mathbf{z}^*)\mathbf{w}$$

$$+ (J_{\psi}^{-1}(\mathbf{z}^*)\mathbf{w})^\top [\sum_{k=1}^n \frac{\partial f}{\partial \mathbf{x}_k}(\boldsymbol{\psi}(\mathbf{z}^*))\nabla^2 \psi_k(\mathbf{z}^*) + \sum_{k=1}^n \sum_{i=1}^m \lambda_i^* \frac{\partial g_i}{\partial \mathbf{x}_k}(\boldsymbol{\psi}(\mathbf{z}^*))\nabla^2 \psi_k(\mathbf{z}^*)]J_{\psi}^{-1}(\mathbf{z}^*)\mathbf{w}$$

$$= (J_{\psi}^{-1}(\mathbf{z}^*)\mathbf{w})^\top \mathcal{L}_{\mathrm{H}}(\mathbf{z}^*, \nu^*)J_{\psi}^{-1}(\mathbf{z}^*)\mathbf{w} \geq 0$$

where the sum of last term of the second '=' is exactly 0 and the last '$\geq$' is from the assumption that $\mathbf{z}^*$ is a second-order KKT stationary point.

$\square$

**Definition D.9** (Non-degenerate KKT stationary point). A second-order KKT point $\mathbf{x}^*$ of $\mathbf{P}$ is said to be non-degenerate if there exists $\boldsymbol{\lambda}^*$ such that

$$\mathbf{d}^\top \nabla^2 \mathcal{L}(\mathbf{x}^*, \boldsymbol{\lambda}^*)\mathbf{d} > 0$$

for all $0 \neq \mathbf{d} \in C_{\mathcal{K}}(\mathbf{x}^*)$. Here the Lagrangian function is

$$\mathcal{L}(\mathbf{x}, \boldsymbol{\lambda}) = f(\mathbf{x}) + \sum_{i=1}^m \lambda_i g_i(\mathbf{x}).$$

**Lemma D.10.** *Suppose strict complementary slackness holds for problem $\mathbf{P}$ and $\mathbf{H}$. Then $\mathbf{x}^\star$ is a non-degenerate KKT point of optimization $\mathbf{P}$ if and only if $\mathbf{z}^*$ satisfying $\mathbf{x}^\star = \boldsymbol{\psi}(\mathbf{z}^*)$ is also a non-degenerate KKT point of problem $\mathbf{H}$.*

*Proof.* 1) Suppose $\mathbf{x}^*$ is a non-degenerate KKT stationary point. Note that for $\mathbf{d} \in C_{\mathcal{B}}(\mathbf{z}^*)$, we have $J_{\psi}(\mathbf{z}^*)\mathbf{d} \in C_{\mathcal{K}}(\mathbf{x}^*)$ from the proof of Lemma D.8. Moreover, from $J_{\psi}(\mathbf{z}^*) \neq 0$ we have $J_{\psi}(\mathbf{z}^*)\mathbf{d} \neq 0$ if and only if $\mathbf{d} \neq 0$. Then the conclusion is trivial from eq. (13) in the proof of Lemma D.8.

2) Now, we suppose $\mathbf{z}^*$ is a non-degenerate KKT stationary point. It follows from the proof of Lemma D.8 that for any $\mathbf{w} \in C_{\mathcal{K}}(\mathbf{x}^*)$, we have $J_{\psi}^{-1}(\mathbf{z}^*)\mathbf{w} \in C_{\mathcal{B}}(\mathbf{z}^*)$. Hence, the conclusion is also trivial from the proof of item (2) of Lemma D.8.

$\square$

## D.6 Global Optimality Property of Optimization Problem $\mathbf{H}$

**Proposition 4.2.** *Suppose problem $\mathbf{P}$ is a convex optimization. Then the following holds.*

*1) If $\mathbf{x}^*$ is a stationary point (thus a global optimum) in problem $\mathbf{P}$, then $\mathbf{z}^* = \boldsymbol{\psi}^{-1}(\mathbf{x}^*)$ is a global optimum in problem $\mathbf{H}$ and $\langle \nabla h(\mathbf{z}^*), \mathbf{z} - \mathbf{z}^* \rangle \geq 0$.*

*2) If $\mathbf{z}^*$ is a stationary point of problem $\mathbf{H}$ and LICQ holds at $\mathbf{z}^*$, then $\mathbf{x}^*$ is a stationary point of problem $\mathbf{P}$ (thus a global optimum). Hence, $\mathbf{z}^*$ is a global optimum.*

*Proof.* 1) Due to the global optimality of $\mathbf{x}^*$, we have

$$f(\mathbf{x}^*) \leq f(\mathbf{x}), \forall \mathbf{x} \in \mathcal{K}.$$

Hence with the homeomorphism $\boldsymbol{\psi}$ and $\mathbf{x} = \boldsymbol{\psi}(\mathbf{z})$, we have

$$f(\boldsymbol{\psi}(\mathbf{z}^*)) \leq f(\boldsymbol{\psi}(\mathbf{z})), \forall \mathbf{z} \in \mathcal{B},$$

that is

$$h(\mathbf{z}^*) \le h(\mathbf{z}), \forall \mathbf{z} \in \mathcal{B}.$$

Therefore, $\langle \nabla h(\mathbf{z}^*), \mathbf{z} - \mathbf{z}^* \rangle \ge 0$ referring to Prop. C.1.

2) By assumption, LICQ holds at $\mathbf{z}^*$. Hence by Proposition D.6, there exists a tuple $(\mathbf{z}^*, \nu^*)$ such that the KKT condition of problem $\mathbf{H}$ holds. It follows from Lemma D.7, there exists $\boldsymbol{\lambda}^*$ such that $(\mathbf{x}^*, \boldsymbol{\lambda}^*)$ also satisfy the KKT condition of problem $\mathbf{P}$. Then it follows from the convexity of $\mathbf{g} = (g_1, \cdots, g_m)$ that $\mathbf{x}^*$ is a global optimum of $\mathbf{P}$. From the above conclusion (item (1)), $\mathbf{z}^*$ is also a global optimum. $\qquad \square$

# E  Convergence Analysis

In this section, we will provide omitted details in Section 4.2.

## E.1  Proof of Theorem 1: Convex Case

Define for any $\mathbf{z}$, $\mathrm{T}(\mathbf{z}) := \mathrm{T}_{\mathcal{S}^{n-1}}(\mathbf{z}) = \{\mathbf{d} : \mathbf{d}^\top \mathbf{z} = 0\}$ and denote $\mathrm{Proj}_{\mathbf{z}}(\mathbf{d}) = \mathrm{Proj}_{\mathrm{T}(\mathbf{z})}(\mathbf{d}) = \mathbf{d} - (\mathbf{d}^\top \mathbf{z})\mathbf{z}$. We define

$$\mathrm{grad}\, h(\mathbf{z}) := \mathrm{Proj}_{\mathbf{z}}(\nabla h(\mathbf{z}))$$

and

$$\mathrm{hess}\, h(\mathbf{z}) = \mathrm{Proj}_{\mathbf{z}} \circ (\nabla^2 h(\mathbf{z}) - \nabla h(\mathbf{z})^\top \mathbf{z} \cdot I_n) \circ \mathrm{Proj}_{\mathbf{z}}.$$

Moreover, we define $\nabla_{\mathcal{B}} h(\mathbf{z}) = \nabla h(\mathbf{z})$ if $\mathbf{z} \in \mathrm{int}(\mathcal{B})$ and $\nabla_{\mathcal{B}} h(\mathbf{z}) = \mathrm{grad}\, h(\mathbf{z})$ if $\mathbf{z} \in \partial \mathcal{B}$. Similarly, define $\nabla_{\mathcal{B}}^2 h(\mathbf{z}) = \nabla^2 h(\mathbf{z})$ if $\mathbf{z} \in \mathrm{int}(\mathcal{B})$ and $\nabla_{\mathcal{B}}^2 h(\mathbf{z}) = \mathrm{hess}\, h(\mathbf{z})$ if $\mathbf{z} \in \partial \mathcal{B}$.

With these notations, the definition of the second-order KKT stationary point $\mathbf{z}^*$ for $\mathbf{H}$ is equivalent to the following (see e.g. [LMY23])

$$\nabla_{\mathcal{B}}\, h(\mathbf{z}^*) = 0, \min \mathrm{eig}(\nabla_{\mathcal{B}}^2 h(\mathbf{z}^*)) \ge 0,$$

where $\min \mathrm{eig}(\cdot)$ represents the minimum eigenvalue. Moreover, $\mathbf{z}^*$ is non-degenerate if in addition $\min \mathrm{eig}(\nabla_{\mathcal{B}}^2 h(\mathbf{z}^*)) > 0$.

We first give some help lemmas in the following.

**Lemma E.1** (Local PL condition). *Suppose $h$ is twice continuously differentiable.*

*1) If $\mathbf{z}^* \in \partial \mathcal{B}$ is a non-degenerate minimizer for $\mathbf{H}$, there exists $\delta := \delta(\mathbf{z}^*)$ and $\tau := \tau(\mathbf{z}^*)$ such that PL inequality holds locally, i.e.,*

$$h(\mathbf{z}) - h(\mathbf{z}^*) \le \frac{1}{2\tau}\|\mathrm{grad}\, h(\mathbf{z})\|^2$$

*for any $\mathbf{z} \in \mathcal{B}(\mathbf{z}^*, \delta) \cap \partial \mathcal{B}$.*

*2) If $\mathbf{z}^* \in \mathrm{int}(\mathcal{B})$ is a non-degenerate minimizer for $\mathbf{H}$, there exists $\delta := \delta(\mathbf{z}^*)$ and $\tau := \tau(\mathbf{z}^*)$ such that PL inequality holds at the ball $\mathcal{B}(\mathbf{z}^*, \delta)$, i.e.,*

$$h(\mathbf{z}) - h(\mathbf{z}^*) \le \frac{1}{2\tau}\|\nabla h(\mathbf{z})\|^2$$

*for any $\mathbf{z} \in \mathcal{B}(\mathbf{z}^*, \delta)$.*

*Proof.* For item (1), $\mathbf{z}^*$ is on the boundary of $\mathcal{B}$. We can consider $h : \partial \mathcal{B} = \mathcal{S}_{n-1} \to \mathbb{R}$. Because $\mathbf{z}^*$ is a non-degenerate minimizer, by continuity there exists a ball $\mathcal{B}(\mathbf{z}^*, \delta)$ and upon $\mathbf{z} \in \mathcal{B}(\mathbf{z}^*, \delta) \cap \partial \mathcal{B}$ we have $\mathrm{hess}\, h(\mathbf{z})$ is positive definite, i.e., $h$ is $\tau$-strongly convex over the ball. Hence PL inequality holds over $\mathbf{z} \in \mathcal{B}(\mathbf{z}^*, \delta) \cap \partial \mathcal{B}$ (refer to e.g. Lemma 11.28 [Bou23]), i.e.,

$$h(\mathbf{z}) - h(\mathbf{z}^*) \le \frac{1}{2\tau}\|\mathrm{grad}\, h(\mathbf{z})\|^2$$

for any $\mathbf{z} \in \mathcal{B}(\mathbf{z}^*, \delta) \cap \partial \mathcal{B}$.

It's easy to show item (2). Actually, by continuity there exists a ball $\mathcal{B}(\mathbf{z}^*, \delta)$ upon which $\nabla^2 h(\mathbf{z})$ is positive definite, i.e., $h$ is $\tau$-strongly convex over the ball. Hence PL inequality holds over the ball (refer to e.g. [KNS16]). $\qquad \square$

**Definition E.2** (Approximate stationary point). A point $\mathbf{x}^*$ is called $\epsilon$-stationary point for problem $\min_{\mathbf{x} \in \mathcal{K}} f(\mathbf{x})$ with convex set $\mathcal{K}$, if the gradient norm mapping

$$G(\mathbf{x}) := G_{1/\alpha} = \frac{1}{\alpha}[\mathbf{x} - \Pi_\mathcal{K}(\mathbf{x} - \alpha \nabla f(\mathbf{x}))]$$

satisfies $\|G(\mathbf{x})\| \le \epsilon$ for proper $\alpha > 0$.

**Lemma E.3.** *Suppose $\mathbf{z}$ and $\mathbf{z}^+ = \Pi_\mathcal{B}(\mathbf{z} - \alpha \nabla h(\mathbf{z}))$ for some $\alpha > 0$ are both on the boundary of $\mathcal{B}$. If the gradient norm mapping*

$$G(\mathbf{z}) := G_{1/\alpha}(\mathbf{z}) = \frac{1}{\alpha}[\mathbf{z} - \mathbf{z}^+]$$

*satisfies $\|G(\mathbf{z})\| \le \epsilon$, then $\|\mathrm{grad}\, h(\mathbf{z})\| \le \mathcal{O}(\epsilon)$.*

*Proof.* From properties of the orthogonal projection C.2, we have

$$\mathbf{z} - \alpha \nabla h(\mathbf{z}) - \mathbf{z}^+ = \mu \mathbf{z}^+$$

for some $\mu \ge 0$, or

$$\nabla h(\mathbf{z}) = G(\mathbf{z}) - \alpha \mu G(\mathbf{z}) + \frac{\mu}{\alpha}\mathbf{z} = c_1 G(\mathbf{z}) + c_2 \mathbf{z}$$

where $c_1 = 1 - \alpha\mu > 0$ (for small enough $\alpha$) and $c_2 = \mu/\alpha$. We have then

$$\begin{aligned}
\mathrm{grad}\, h(\mathbf{z}) &= \nabla h(\mathbf{z}) - \langle \nabla h(\mathbf{z}), \mathbf{z} \rangle \mathbf{z} \\
&= c_1 G(\mathbf{z}) - \langle c_1 G(\mathbf{z}), \mathbf{z} \rangle \mathbf{z}.
\end{aligned}$$

Thus,

$$\|\mathrm{grad}\, h(\mathbf{z})\| \le 2c_1 \|G(\mathbf{z})\| \le \mathcal{O}(\epsilon).$$

$\square$

**Lemma E.4.** *Suppose $h$ is twice continuously differentiable and $\mathbf{z}^*$ is a unique minimizer (stationary point) for $\mathbf{H}$. Then for any $\delta > 0$, there exists an $\epsilon_\delta > 0$ such that $\mathbf{z}$ is an $\epsilon_\delta$ stationary point can imply $\|\mathbf{z} - \mathbf{z}^*\| < \delta$.*

*Proof.* We suppose there does not exist such $\epsilon_\delta > 0$. Then for any $\epsilon := 1/k$, we can find an $\epsilon$ stationary point $\mathbf{z}_k$ such that $\|\mathbf{z}_k - \mathbf{z}^*\| \ge \delta$. As $\mathcal{B}$ is compact, we may assume $\mathbf{z}_k \to \bar{\mathbf{z}} \in \mathcal{B}$. Then $\bar{\mathbf{z}}$ is a stationary point by continuity of the gradient norm mapping:

$$\|G(\bar{\mathbf{z}})\| = \|G(\lim_{k \to \infty} \mathbf{z}_k)\| = \lim_{k \to \infty} \|G(\mathbf{z}_k)\| \le \lim_{k \to \infty} \epsilon_k = 0.$$

This contradicts $\|\mathbf{z}_k - \mathbf{z}^*\| \ge \delta$. $\square$

**Theorem 1.** *Suppose the strict complementary slackness condition holds for both problem $\mathbf{P}$ and $\mathbf{H}$ and we suppose the problem $\mathbf{P}$ is convex and has a non-degenerate minimizer $\mathbf{x}^*$. Let $\{\mathbf{z}_k\}$ be the sequence generated by Hom-PGD with step-size $\alpha \in (0, \frac{2}{L_h}]$. For sufficient small $\epsilon > 0$, $\{\mathbf{z}_k\}_{k=1}^K$ with $K = \mathcal{O}(L_h/\epsilon)$ contains $\mathbf{z}'$ such that*

$$h(\mathbf{z}') - h^\star \le \epsilon.$$

*Proof.* From Lemma D.10, $\mathbf{z}^\star$ is also a non-degenerate stationary point.

1) We first assume $\mathbf{z}^* \in \partial \mathcal{B}$.

By Lemma E.1, there exists $\tau^* > 0$ and $\delta^*$ such that for any $\mathbf{z} \in \mathcal{B}(\mathbf{z}^\star, \delta^\star) \cap \partial \mathcal{B}$,

$$h(\mathbf{z}) - h(\mathbf{z}^*) \le \frac{1}{2\tau^*} \| \mathrm{grad}\, h(\mathbf{z}) \|^2.$$

Moreover, we assume this $\delta^*$ is small enough to satisfy the condition that for any $\mathbf{z} \in \mathcal{B}(\mathbf{z}^\star, \delta^\star)$, $\mathbf{z} - \alpha \nabla h(\mathbf{z})$ is outside the ball, which implies $\mathbf{z}^+ = \Pi_\mathcal{B}(\mathbf{z} - \alpha h(\mathbf{z})) \in \partial \mathcal{B}$. This $\delta^*$ exists by the continuity of $\nabla h$ and note $\nabla h(\mathbf{z}^*) = \mu \mathbf{z}^*$ for some $\mu^* > 0$ (where '>' is from strict complementary slackness condition). Hence if PGD can find an approximate stationary point $\mathbf{z}' \in \mathcal{B}(\mathbf{z}^\star, \delta^\star)$, we can assume $\mathbf{z}' \in \partial \mathcal{B}$.

From Lemma E.4, there exists $\epsilon_{\delta^*} > 0$ such that if $\mathbf{z}$ is an $\epsilon_{\delta^*}$ stationary point then $\mathbf{z} \in \mathcal{B}(\mathbf{z}^\star, \delta^*)$.

So, suppose $\epsilon > 0$ is small enough that $c\sqrt{\epsilon} < \epsilon_{\delta^*}$ where $c > 0$ is a fixed constant determined later. By Theorem 3, PGD finds a $c\sqrt{\epsilon}$-stationary point $\mathbf{z}' \in \partial \mathcal{B}$, within $K = \mathcal{O}\left(L_h/(\sqrt{\epsilon})^2\right) = \mathcal{O}\left(1/\epsilon\right)$ iterations, i.e., $\|G(\mathbf{z}')\| \le c\sqrt{\epsilon}$. From Lemma E.3, we can choose proper $c$ such that $\|\mathrm{grad}\, h(\mathbf{z}')\| \le \sqrt{2\tau^*\epsilon}$.

As $c\sqrt{\epsilon} < \epsilon_{\delta^*}$ we know $\mathbf{z}'$ is also an $\epsilon_{\delta^*}$-stationary point within $B(\mathbf{z}^\star, \delta^*)$.

$$h(\mathbf{z}') - h(\mathbf{z}^\star) \le \frac{1}{2\tau^*}\|\mathrm{grad}\, h(\mathbf{z}')\|^2 \le \frac{1}{2\tau^*}(\sqrt{2\tau^*\epsilon})^2 \le \epsilon.$$

2) The case when $\mathbf{z}^* \in \mathrm{int}(\mathcal{B})$ is similar.

By Lemma E.1, there exists $\tau^* > 0$ and $\delta^* > 0$ such that for any $\mathbf{z} \in \mathcal{B}(\mathbf{z}^\star, \delta^\star)$,

$$h(\mathbf{z}) - h(\mathbf{z}^*) \le \frac{1}{2\tau^*}\|\nabla h(\mathbf{z})\|^2.$$

From Lemma E.4, there exists $\epsilon_{\delta^*} > 0$ such that if $\mathbf{z}$ is an $\epsilon_{\delta^*}$ stationary point then $\mathbf{z} \in \mathcal{B}(\mathbf{z}^\star, \delta)$.

Similarly, suppose $\epsilon > 0$ is small enough that $c\sqrt{\epsilon} < \epsilon_{\delta^*}$ where $c > 0$ is a fixed constant determined later. By Theorem 3, PGD finds a $c\sqrt{\epsilon}$-stationary point $\mathbf{z}'$, within $K = \mathcal{O}\left(1/(\sqrt{\epsilon})^2\right) = \mathcal{O}\left(1/\epsilon\right)$ iterations, i.e., $\|G(\mathbf{z}')\| \le c\sqrt{\epsilon}$. Because $\mathbf{z}^*$ is in the interior of $\mathcal{B}$, hence when $\mathbf{z}'$ is close enough to $\mathbf{z}^*$ ($\delta$ is sufficient small), the gradient norm mapping $G(\mathbf{z}')$ is exactly $\nabla h(\mathbf{z}')$. So we can choose proper $c$ such that $\|\nabla h(\mathbf{z}')\| \le \sqrt{2\tau^*\epsilon}$.

As $c\sqrt{\epsilon} < \epsilon_{\delta^*}$ we know $\mathbf{z}'$ is also an $\epsilon_{\delta^*}$ stationary point within $B(\mathbf{z}^\star, \delta^*)$. Then we have

$$h(\mathbf{z}') - h(\mathbf{z}^\star) \le \frac{1}{2\tau^*}\|\mathrm{grad}\, h(\mathbf{z}')\|^2 \le \frac{1}{2\tau^*}(\sqrt{2\tau^*\epsilon})^2 \le \epsilon.$$

$\square$

## E.2 Proof of Theorem 2: Strongly Convex Case

In this section, we show Theorem 2. Actually, Theorem 2 is a corollary of the following theorem.

**Theorem 4.** *Suppose $f$ is $\mu_f$-strongly convex and $\mathcal{K}$ is a convex set. Then the updating sequence $\{\mathbf{z}_k\}$ by Hom-PGD algorithm with a constant step size $\alpha \in (0, \frac{2}{L_h}]$, converges to a global optimum point $\mathbf{z}^\star$ linearly, i.e.,*

$$h(\mathbf{z}_K) - h(\mathbf{z}^*) \le \sigma^K (h(\mathbf{z}_0) - h(\mathbf{z}^*)), \mathrm{dist}(\mathbf{z}_K, \mathcal{B}^*) \le \frac{2}{\mu_f \kappa_1}\sigma^K (h(\mathbf{z}_0) - h(\mathbf{z}^*))$$

*where $\sigma = 1 - \frac{\mu_f \kappa_1}{L_h} \in (0, 1)$.*

*Proof.* By strong convexity of $f$, and convexity of $\mathcal{K}$, one can show that $f$ satisfy proximal PL condition over $\mathcal{K}$ (see e.g. Appendix G [KNS16]), i.e.,

$$\mathcal{D}_{\delta_\mathcal{K}}(\mathbf{x}, \mu_f; f) \ge 2\mu_f(f(\mathbf{x}) - f(\mathbf{x}^*))$$

where

$$\mathcal{D}_{\delta_\mathcal{K}}(\mathbf{x}, \lambda; f) = -2\lambda \min_{y \in \mathcal{K}}\left[\langle \nabla f(\mathbf{x}), \mathbf{y} - \mathbf{x}\rangle + \frac{\lambda}{2}\|\mathbf{y} - \mathbf{x}\|^2\right].$$

Next, we show that $h = f \circ \psi$ also satisfies the proximal-PL condition over $\mathcal{B}$. We derive with $\mathbf{x} = \psi(\mathbf{z}), \mathbf{y} = \psi(\mathbf{u})$ as follows

$$\langle \nabla h(\mathbf{z}), \mathbf{u} - \mathbf{z} \rangle + \frac{L_{f,0} L_\psi}{2} \|\mathbf{u} - \mathbf{z}\|^2$$

$$= \langle \mathrm{J}_\psi(\mathbf{z})^\top \nabla f(\psi(\mathbf{z})), \mathbf{u} - \mathbf{z} \rangle + \frac{L_{f,0} L_\psi}{2} \|\mathbf{u} - \mathbf{z}\|^2$$

$$= \langle \nabla f(\psi(\mathbf{z})), \mathrm{J}_\psi(\mathbf{z})(\mathbf{u} - \mathbf{z}) \rangle + \frac{L_{f,0} L_\psi}{2} \|\mathbf{u} - \mathbf{z}\|^2$$

$$= \langle \nabla f(\psi(\mathbf{z})), -\psi(\mathbf{u}) + \psi(\mathbf{z}) + \mathrm{J}_\psi(\mathbf{z})(\mathbf{u} - \mathbf{z}) \rangle + \langle \nabla f(\psi(\mathbf{z})), \psi(\mathbf{u}) - \psi(\mathbf{z}) \rangle + \frac{L_{f,0} L_\psi}{2} \|\mathbf{u} - \mathbf{z}\|^2$$

$$\leq \frac{L_{f,0} L_\psi}{2} \|\mathbf{u} - \mathbf{z}\|^2 + \langle \nabla f(\mathbf{x}), \mathbf{y} - \mathbf{x} \rangle + \frac{L_{f,0} L_\psi}{2} \|\mathbf{u} - \mathbf{z}\|^2$$

$$= \langle \nabla f(\mathbf{x}), \mathbf{y} - \mathbf{x} \rangle + \frac{2 L_{f,0} L_\psi}{2} \left\| \psi^{-1}(\mathbf{y}) - \psi^{-1}(\mathbf{x}) \right\|^2$$

$$\leq \langle \nabla f(\mathbf{x}), \mathbf{y} - \mathbf{x} \rangle + \frac{2 L_{f,0} L_\psi}{2 \kappa_1} \|\mathbf{y} - \mathbf{x}\|^2,$$

where
the 2-nd line is from the chain rule of the gradient,
the 3-rd line is from property of inner product, i.e., for vector $\mathbf{a}, \mathbf{b}$ and matrix $A$, $\langle A^\top \mathbf{a}, \mathbf{b} \rangle = \langle \mathbf{a}, A\mathbf{b} \rangle$,
the 4-th line is based on simple caculation,
the 5-th line is from Lemma C.3, and the transformation $\mathbf{z} = \psi(\mathbf{x}), \mathbf{y} = \psi(\mathbf{u})$,
the 6-th line is from the inverse transformation $\psi^{-1}$,
and the last line is from the bi-Lipschitz property of $\psi$.

Briefly, we get

$$\langle \nabla h(\mathbf{z}), \mathbf{u} - \mathbf{z} \rangle + \frac{L_{f,0} L_\psi}{2} \|\mathbf{u} - \mathbf{z}\|^2 \leq \langle \nabla f(\mathbf{x}), \mathbf{y} - \mathbf{x} \rangle + \frac{2 L_{f,0} L_\psi}{2 \kappa_1} \|\mathbf{y} - \mathbf{x}\|^2 \qquad (14)$$

Next, we assume $c_1 := \frac{2 L_{f,0} L_\psi}{\kappa_1} \geq \mu_f$ without loss of generality. This is because if $c_1 < \mu_f$, we have

$$\langle \nabla h(\mathbf{z}), \mathbf{u} - \mathbf{z} \rangle + \frac{L_{f,0} L_\psi}{2} \|\mathbf{u} - \mathbf{z}\|^2 \leq \langle \nabla f(\mathbf{x}), \mathbf{y} - \mathbf{x} \rangle + \frac{\mu_f}{2} \|\mathbf{y} - \mathbf{x}\|^2$$

or

$$-2 \min_{\mathbf{u} \in \mathcal{B}} \left\{ \langle \nabla h(\mathbf{z}), \mathbf{u} - \mathbf{z} \rangle + \frac{L_{f,0} L_\psi}{2} \|\mathbf{u} - \mathbf{z}\|^2 \right\} \geq -2 \min_{\mathbf{y} \in \mathcal{K}} \left\{ \langle \nabla f(\mathbf{x}), \mathbf{y} - \mathbf{x} \rangle + \frac{\mu_f}{2} \|\mathbf{y} - \mathbf{x}\|^2 \right\}$$

$$\geq 2 \left( f(\mathbf{x}) - f^* \right) = 2 \left( h(\mathbf{z}) - h^* \right).$$

That is, if $c_1 < \mu_f$, we directly get the conclusion that $h$ satisfy proximal PL condition over $\mathcal{B}$:

$$\mathcal{D}_{\delta_\mathcal{B}}(\mathbf{z}, L_{f,0} L_\psi; h) \geq 2 L_{f,0} L_\psi \left( h(\mathbf{z}) - h(\mathbf{z}^*) \right).$$

In the following, we assume $c_1 \geq \mu_f$. By Lemma 1 in [KNS16], for any convex set $\mathcal{K}$ and differentiable function $f$, $\mathcal{D}_{\delta_\mathcal{K}}(\mathbf{x}, \mu; f)$ is monotone increasing in $\mu$, i.e., $\mathcal{D}_{\delta_\mathcal{K}}(\mathbf{x}, \mu_1; f) \geq \mathcal{D}_{\delta_\mathcal{K}}(\mathbf{x}, \mu_2; f)$ given $\mu_2 \geq \mu_1 > 0$. Hence, we have

$$\mathcal{D}_{\delta_\mathcal{K}}(\mathbf{x}, c_1; f) \geq \mathcal{D}_{\delta_\mathcal{K}}(\mathbf{x}, \mu_f; f) \geq 2 \mu_f (f(\mathbf{x}) - f^*).$$

Then it following from Eq. (14) that

$$\frac{1}{L_{f,0} L_\psi} \mathcal{D}_{\delta_\mathcal{B}}(\mathbf{z}, L_{f,0} L_\psi; h) \geq \frac{1}{c_1} \mathcal{D}_{\delta_\mathcal{K}}(\mathbf{x}, c_1; f) \geq \frac{1}{c_1} \mathcal{D}_{\delta_\mathcal{K}}(\mathbf{x}, \mu_f; f) \geq 2 \frac{\mu_f}{c_1} (f(\mathbf{x}) - f^*).$$

Hence we have $h$ satisfy proximal PL condition over $\mathcal{B}$:

$$\mathcal{D}_{\delta_\mathcal{B}}(\mathbf{z}, L_{f,0} L_\psi; h) \geq 2 L_{f,0} L_\psi \frac{\mu_f}{c_1} (h(\mathbf{z}) - h(\mathbf{z}^*)) = 2 \frac{\mu_f \kappa_1}{2} (h(\mathbf{z}) - h(\mathbf{z}^*)). \qquad (15)$$

Finally, we show the linear convergence rate of the projected gradient descent algorithm. We derive

$$
\begin{aligned}
\mathbf{z}_{k+1} &= \Pi_{\mathcal{B}}\left(\mathbf{z}_k - \alpha \nabla h\left(\mathbf{z}_k\right)\right) \\
&= \arg\min_{\mathbf{u}\in\mathcal{B}} \left\|\mathbf{u} - \left(\mathbf{z}_k - \alpha \nabla h\left(\mathbf{z}_k\right)\right)\right\|^2 \\
&= \arg\min_{\mathbf{u}\in\mathcal{B}} \left\{\alpha \left\langle \nabla h\left(\mathbf{z}_k\right), \mathbf{u} - \mathbf{z}_k \right\rangle + \left\|\mathbf{u} - \mathbf{z}_k\right\|^2\right\} \\
&= \arg\min_{\mathbf{u}\in\mathcal{B}} \left\{\left\langle \nabla h\left(\mathbf{z}_k\right), \mathbf{u} - \mathbf{z}_k \right\rangle + \frac{1}{\alpha}\left\|\mathbf{u} - \mathbf{z}_k\right\|^2\right\},
\end{aligned}
$$

where the 2-nd line is from the definition of orthogonal projection $\Pi$, the 3-rd and last line is from simple calculation.

This implies

$$
-\frac{\alpha}{2}\mathcal{D}_{\delta_{\mathcal{B}}}(\mathbf{z}, \frac{2}{\alpha}; h) = \langle \nabla h(\mathbf{z}_k), \mathbf{z}_{k+1}-\mathbf{z}_k\rangle + \frac{1}{\alpha}\|\mathbf{z}_{k+1}-\mathbf{z}_k\|^2 \geq \langle h(\mathbf{z}_k), \mathbf{z}_{k+1} - \mathbf{z}_k\rangle + \frac{L_h}{2}\|\mathbf{z}_{k+1}-\mathbf{z}_k\|^2 \tag{16}
$$

where the last inequality is from the selection of stepsize $\alpha \in (0, \frac{2}{L_h}]$ and recall $L_h = \kappa_2^2 L_{f,1} + L_\psi L_{f,0}$ from Prop. D.1.

Then the iterative of PGD satisfies,

$$
\begin{aligned}
h(\mathbf{z}_{k+1}) &\leq h(\mathbf{z}_k) + \langle \nabla h(\mathbf{z}_k), \mathbf{z}_{k+1} - \mathbf{z}_k\rangle + \frac{L_h}{2}\|\mathbf{z}_{k+1} - \mathbf{z}_k\|^2 \\
&\leq h(\mathbf{z}_k) - \frac{\alpha}{2}\mathcal{D}_{\delta_{\mathcal{B}}}(\mathbf{z}_k, \frac{2}{\alpha}; h) \\
&\leq h(\mathbf{z}_k) - \frac{\alpha}{2}\mathcal{D}_{\delta_{\mathcal{B}}}(\mathbf{z}_k, L_{f,0}L_\psi; h) \\
&\leq h(\mathbf{z}_k) - \frac{\alpha \mu_f \kappa_1}{2}(h(\mathbf{z}_k) - h(\mathbf{z}^*)),
\end{aligned}
$$

where
the 1-st line is from the $L_h$-smoothness of $h$,
the 2-nd line is from Eq. (16),
the 3-rd line is from monotone increasing property of $\mathcal{D}_{\delta_{\mathcal{B}}}(\mathbf{z}, \cdot; h)$ and the choice of $\alpha \in (0, \frac{2}{L_h}]$,
and the last line is from the proximal PL condition Eq. (15) of $h$.

Take $\alpha = \frac{2}{L_h}$,

$$
h(\mathbf{z}_{k+1}) - h(\mathbf{z}^*) \leq (1 - \frac{\mu_f \kappa_1}{L_h})(h(\mathbf{z}_k) - h(\mathbf{z}^*)).
$$

Then we rewrite

$$
h(\mathbf{z}_K) - h(\mathbf{z}^*) \leq \sigma^K(h(\mathbf{z}_0) - h(\mathbf{z}^*)), \quad \text{where} \quad \sigma = 1 - \frac{\mu_f \kappa_1}{L_h}.
$$

From quadratic growth condition D.1, we have

$$
\mathrm{dist}(\mathbf{z}_K, \mathcal{B}^*) \leq \frac{2}{\mu_f \kappa_1}(h(\mathbf{z}_K) - h(\mathbf{z}^*)) \leq \frac{2}{\mu_f \kappa_1}\sigma^K(h(\mathbf{z}_0) - h(\mathbf{z}^*)).
$$

Let $\sigma^K(h(\mathbf{z}_0) - h(\mathbf{z}^*)) \leq \epsilon$. We have

$$
K = \mathcal{O}(\frac{\log 1/\epsilon}{\log 1/\sigma}) = \mathcal{O}\left((1 - \sigma)^{-1}\log 1/\epsilon\right)
$$

where the last '=' holds when $\sigma$ is closed to 1. □

*Remark* E.5. It follows directly from the proof that the assumption of strong convexity can be relaxed. It suffices for the objective to satisfy the proximal-PL condition over the *convex* constraint set $\mathcal{K}$ in order to achieve the convergence rate of $\mathcal{O}(\log 1/\epsilon)$. Moreover, there exist many alternative and equivalent assumptions for the proximal-PL condition. We list some commonly used equivalent assumptions below, where the proof of the equivalence can be referred to, e.g, [KNS16].

- Proximal error bounds (Proximal-EB): There exists $c > 0$ such that

$$\|\mathbf{x} - \mathbf{x}_{\mathcal{K}^*}\| \leq c \left\| \mathbf{x} - \text{prox}_{\frac{1}{L_f}\delta_{\mathcal{K}}} \left( \mathbf{x} - \frac{1}{L_f}\nabla f(\mathbf{x}) \right) \right\|$$

where $\mathbf{x}_{\mathcal{K}^*}$ is the orthogonal projection of $\mathbf{x}$ onto the optimal solution set $\mathcal{K}^*$ and $\text{prox}_g(\mathbf{x}) = \arg\min_{\mathbf{u}} \left\{ g(\mathbf{u}) + \frac{1}{2}\|\mathbf{u} - \mathbf{x}\|^2 \right\}$. Moreover, note that for $\lambda > 0$, we have $\text{prox}_{\lambda\delta_{\mathcal{K}}} = \Pi_{\mathcal{K}}$.

- Kurdyka-Łojasiewicz (KL) condition: There exists $\mu_f > 0$ such that

$$\min_{s \in \partial F(\mathbf{x})} \|s\|^2 \geq 2\mu_f \left( F(\mathbf{x}) - F^* \right)$$

where $F(\mathbf{x}) = f(\mathbf{x}) + \delta_{\mathcal{K}}(\mathbf{x})$ and $\partial F(\mathbf{x})$ is the Frechet subdifferential [RW09]. In this case,

$$\partial F(\mathbf{x}) = \{\nabla f(\mathbf{x}) + \boldsymbol{\xi} \mid \boldsymbol{\xi} \in \partial\delta_{\mathcal{K}}(\mathbf{x})\}$$

where $\partial\delta_{\mathcal{K}}(\mathbf{x})$ can be simlified as $N_{\mathcal{K}}(\mathbf{x})$, with $N_{\mathcal{K}}(\mathbf{x}) = \{\mathbf{y} : \langle \mathbf{y}, \mathbf{z} - \mathbf{x}\rangle \leq 0, \forall \mathbf{z} \in \mathcal{K}\}$ for $\mathbf{x} \in \mathcal{K}$ and $N_{\mathcal{K}}(\mathbf{x}) = \emptyset$ for $\mathbf{x} \notin \mathcal{K}$. Note that KL condition can imply PL condition ($\|\nabla f(\mathbf{x})\|^2 \geq 2\mu_f(f(\mathbf{x}) - f^*)$) but the converse does not hold in general.

# F  Experiments Setting

## F.1  Problem Formulations and Instance Generation

**Optimization over polyhedron**: We first consider a two-dimensional optimization over a convex polyhedron to illustrate the effectiveness of our methods. The problem is defined as:

$$\min_{\mathbf{L} \leq \mathbf{x} \leq \mathbf{U}} \quad \sum_{i=1}^{2} w_i(x_i - 1)^2 \qquad \text{s.t.} \quad \mathbf{a}_i^\top \mathbf{x} \leq b_i, \quad i = 1, \ldots, n_{\text{lin}} \tag{17}$$

where $\mathbf{x} \in \mathbb{R}^2$ is the decision variable, $w_i$ is the positive coefficients, and $\mathbf{L}, \mathbf{U} \in \mathbb{R}^2$ represent the lower and upper bounds on the variables. $\mathbf{a}_i \in \mathbb{R}^2$ and $b_i \in \mathbb{R}$ represents the coefficients in $n_{\text{lin}}$ linear constraints. The homeomorphic counterparts are derived by a closed-form gauge mapping as discussed in B.4.

**Optimization over star-shaped set**: We then consider a two-dimensional optimization over a non-convex star-shaped set to illustrate the effectiveness of our methods. The problem and its homeomorphic counterpart are defined as:

$$\begin{array}{cc} \min_{x} & \sum_{i=1}^{2} w_i(x_i - 1)^2 \\ \text{s.t.} & \|\mathbf{x}\| \leq \Gamma_{\alpha,n}(\mathbf{x}) \end{array} \quad \begin{array}{c} \stackrel{\boldsymbol{\psi}^{-1}}{\Longrightarrow} \\ \stackrel{\longleftarrow}{\boldsymbol{\psi}} \end{array} \quad \begin{array}{cc} \min_{z} & \sum_{i=1}^{2} w_i(z_i \cdot \Gamma_{\alpha,n}(\mathbf{z}) - 1)^2 \\ \text{s.t.} & \|\mathbf{z}\| \leq 1 \end{array} \tag{18}$$

$$\boldsymbol{\psi}(\mathbf{z}) = [z_1 \cdot \Gamma_{\alpha,n}(\mathbf{z}), z_2 \cdot \Gamma_{\alpha,n}(\mathbf{z})], \quad \boldsymbol{\psi}^{-1}(\mathbf{x}) = [x_1/\Gamma_{\alpha,n}(\mathbf{x}), x_2/\Gamma_{\alpha,n}(\mathbf{x})], \tag{19}$$

where $\mathbf{z}, \mathbf{x} \in \mathbb{R}^2$ and $\Gamma_{\alpha,n}$ is a non-linear function with parameters $\alpha > 0$ and $n \in \mathbb{Z}^+$, defined as $\Gamma_{\alpha,n}([x_1, x_2]) := 1 + \alpha\sin(n\arctan(x_2/x_1))$. Under the homeomorphic mapping $\boldsymbol{\psi}$, the non-convex-constrained optimization problem can be transformed into a ball-constrained non-convex optimization. We then compared different iterative algorithms over this problem.

**Second-order cone programming**: We then consider convex second-order cone programming (SOCP), which encompasses linear programming (LP), quadratic programming (QP), and convex quadratically constrained quadratic programming (QCQP) problems. This formulation has wide applications in portfolio optimization and optimal power flow problems [Low14a, Low14b].

$$\min_{\mathbf{L} \leq \mathbf{x} \leq \mathbf{U}} \quad \frac{1}{2}\mathbf{x}^\top \mathbf{Q}\mathbf{x} + \mathbf{p}^\top \mathbf{x} \qquad \text{s.t.} \quad \|\mathbf{G}_i\mathbf{x} + \mathbf{h}_i\| \leq \mathbf{c}_i^\top \mathbf{x} + d_i, \quad i = 1, \ldots, n_{\text{soc}} \tag{20}$$

where $\mathbf{x} \in \mathbb{R}^n$ is the decision variable, $\mathbf{Q} \in \mathbb{R}^{n \times n}$ is a symmetric positive semidefinite matrix, $\mathbf{p} \in \mathbb{R}^n$ is a vector of linear cost coefficients, and $\mathbf{L}, \mathbf{U} \in \mathbb{R}^n$ represent the lower and upper bounds on the variables. For the second-order cone constraints, $\mathbf{G}_i \in \mathbb{R}^{m_i \times n}$ and $\mathbf{h}_i \in \mathbb{R}^{m_i}$ define the affine function inside the norm, while $\mathbf{c}_i \in \mathbb{R}^n$ and $d_i \in \mathbb{R}$ define the affine function on the right-hand side.

The parameter $n_{\mathrm{soc}}$ represents the number of second-order cone constraints in the problem. The total number of constraints also includes the upper/lower bound on the decision variables.

**Max-cut semidefinite programming**: We consider an important class of SDP in the max-cut problem. Given a graph $G = \{\mathcal{N}, \mathcal{E}\}$ with node set $i \in \mathcal{N}$ and edge set $(i, j) \in \mathcal{E}$, the max-cut SDP problem is formulated as:

$$\max_{-1 \leq \mathbf{X} \leq 1} \quad \sum_{(i,j) \in \mathcal{E}} (1 - x_{ij})/2 \tag{21}$$

$$\text{s.t.} \quad x_{ii} = 1, \quad i = 1, \cdots, n \tag{22}$$

$$\mathbf{X} \succeq \mathbf{0}, \tag{23}$$

where $\mathbf{X} \succeq \mathbf{0}$ indicates that $\mathbf{X}$ is positive semidefinite. Define the upper triangle off-diagonal elements in $\mathbf{X}$ as $\mathbf{y} \in \mathbb{R}^{(N^2 - N)/2}$, then the SDP can be equivalently reformulated in Linear Matrix Inequality (LMI)-based form as:

$$\max_{-1 \leq \mathbf{y} \leq 1} \quad \sum_{k=(i,j) \in \mathcal{E}} (1 - y_k)/2 \tag{24}$$

$$\text{s.t.} \quad \mathbf{I} + \sum_{k} y_k \cdot \mathbf{A}_k \succeq \mathbf{0}, \tag{25}$$

where $\mathbf{A}_k$ is a symmetric matrix with zeros on the diagonal and with a 1 in the $(i, j)$ and $(j, i)$ positions corresponding to the $k$-th off-diagonal entry, and zeros elsewhere. Given such an LMI-based formulation, we can construct the homeomorphic counterpart based on a closed gauge mapping as discussed in B.4. Note that a "central" interior point for such a PSD cone is naturally the zero vector $\mathbf{y}^\circ = \mathbf{0}$.

We also consider solving the well-known Burer-Monteiro (BM) factorization-based semidefinite program via augmented Lagrangian methods [BM03]. Let $\mathbf{X} = \mathbf{V}\mathbf{V}^{\mathbf{T}}$, where $\mathbf{V} \in \mathbb{R}^{N \times r}$ and $r$ is the selected rank, then we have the following low-rank SDP:

$$\max_{\mathbf{X} = \mathbf{V}\mathbf{V}^{\mathbf{T}}} \quad \sum_{(i,j) \in \mathcal{E}} (1 - x_{ij})/2 \tag{26}$$

$$\text{s.t.} \quad x_{ii} = 1, \quad i = 1, \cdots, n \tag{27}$$

$$-\mathbf{1} \leq \mathbf{X} \leq \mathbf{1} \tag{28}$$

In our experiments, we consider both log-rank ($r = \log(N)$) and Barvinok-Pataki (bp)-rank ($r = \sqrt{2N}$) factorization-based SDP methods [Bar95, Pat98, BVB20].

### F.2 Baseline Algorithms and Hyper-Parameters

We implement the baselines as follows:

- **PGD**:

$$\mathbf{x}_{k+1} = \Pi_{\mathcal{K}}(\mathbf{x}_k - \alpha_k \nabla f(\mathbf{x}_k)) \tag{29}$$

where $\Pi_{\mathcal{K}}$ denotes the orthogonal projection onto the feasible set $\mathcal{K}$, $\alpha_k > 0$ is the step size at iteration $k$, and $\nabla f(\mathbf{x}_k)$ is the gradient of the objective function at point $\mathbf{x}_k$. The quadratic projection problem is solved via MOSEK for convex problems and by ALM for non-convex problems.

- **FW**:

$$\mathbf{s}_k = \arg\min_{\mathbf{s} \in \mathcal{K}} \langle \nabla f(\mathbf{x}_k), \mathbf{s} \rangle, \tag{30}$$

$$\mathbf{x}_{k+1} = (1 - \alpha)\mathbf{x}_k + \alpha_k \mathbf{s}_k, \tag{31}$$

where $\alpha_k \in [0, 1]$ is the step size at iteration $k$. The linear minimization problem is solved via MOSEK for convex problems and by ALM for non-convex problems.

- **ALM**:

$$\mathbf{x}_{k+1} = \arg\min_{\mathbf{x}} \{ f(\mathbf{x}) + \boldsymbol{\lambda}_k^T \mathbf{g}(\mathbf{x}) + \rho_k \cdot \mathbf{1}^T [\mathbf{g}(\mathbf{x})]_+^2 \}, \tag{32}$$

$$\boldsymbol{\lambda}_{k+1} = [\boldsymbol{\lambda}_k + \rho_k \cdot \mathbf{g}(\mathbf{x}_{k+1})]_+, \tag{33}$$

where $\boldsymbol{\lambda}_k$ is the Lagrange multipliers, $g(\mathbf{x})$ represents the constraint functions, and $\rho_k > 0$ is the dual step size. The inner unconstrained optimization problem is solved by gradient descent.

- **RD**:

$$\mathbf{y}_{k+1} = \mathbf{y}_k - \alpha_k \nabla \max\{f^\Gamma(\mathbf{y}_k), \gamma_\mathcal{K}(\mathbf{y}_k)\} \tag{34}$$

where $f^\Gamma$ is the radial dual of the objective function and $\gamma_\mathcal{K}$ is the gauge function [Gri24b]. The solution is mapped to the original space after convergence as $\mathbf{x}^* = \mathbf{y}^*/f^\Gamma(\mathbf{y}^*)$ via radial dual.

- **Hom-PGD**:

$$\mathbf{z}_{k+1} = \Pi_\mathcal{B}(\mathbf{z}_k - \alpha_k \nabla f(\boldsymbol{\psi}(\mathbf{z}_k))) \tag{35}$$

where $\Pi_\mathcal{B}$ denotes the projection onto unit ball $\mathcal{B}$, and $\boldsymbol{\psi}$ is the homeomorphism. The solution is mapped to the original space after convergence as $\mathbf{x}^* = \boldsymbol{\psi}(\mathbf{z}^*)$.

- **MOSEK**: A commercial interior-point optimizer that solves conic optimization problems efficiently using highly optimized primal-dual interior-point methods with predictor-corrector techniques and sparse linear algebra. Note that we use an Academic license for MOSEK.

**Gradient calculation**: For simple quadratic objective functions, gradients are calculated via closed-form formulations. Other non-trivial gradient calculations across the various algorithms are implemented using auto-differentiation in PyTorch. We note that replacing auto-differentiation with closed-form gradient implementations could further improve the computational efficiency of the algorithms.

**Step-size**: Theoretically, different algorithms employ their own step size selection strategies, such as explicit dependence on smoothness and convexity parameters, or implicit step sizes that depend on the optimal objective value [Gri24b]. For practical implementation, we initialize a fixed step size (e.g., $10^{-3}$) and decay it by a factor of 0.999 if the objective value does not decrease, which helps identify a sufficient step size for convergence. Although more sophisticated backtracking line search methods or adaptive step size schemes could accelerate convergence, we do not implement these for the sake of fair comparison.

**Computational Environment**: We conduct our experiments across two computational platforms to accommodate different problem scales. For small-scale illustrative examples, we execute algorithm comparisons on a MacBook Pro 2023. For larger-scale SOCP and SDP experiments, we implement all algorithms in PyTorch and execute them on an Ubuntu server equipped with an NVIDIA A800 GPU and an AMD EPYC 7763 64-Core Processor.

# G    Supplementary Experiments Results

## G.1    Illustrative Examples

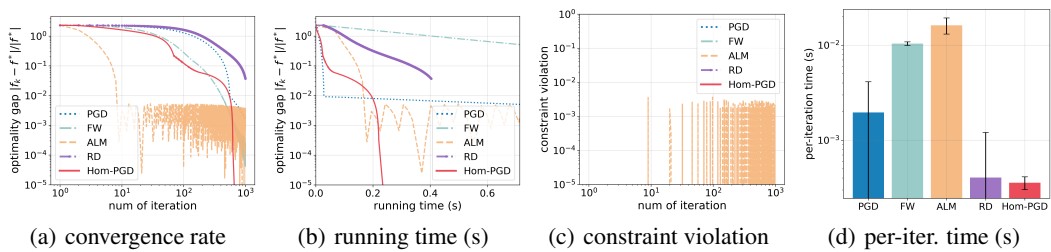

(a) convergence rate     (b) running time (s)     (c) constraint violation     (d) per-iter. time (s)

Figure 8: Convergence performance for optimization over polyhedron.

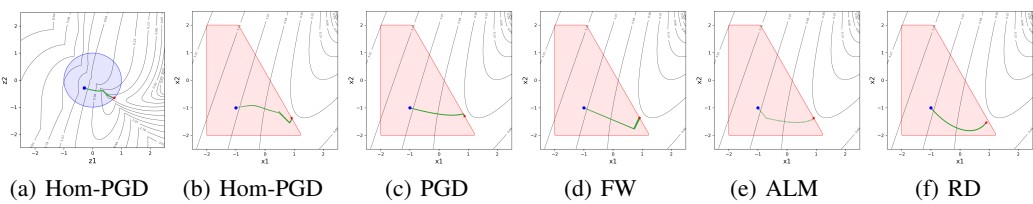

(a) Hom-PGD    (b) Hom-PGD    (c) PGD    (d) FW    (e) ALM    (f) RD

Figure 9: Iteration trajectory for optimization over polyhedron. Hom-PGD (a) and RD (f) are also mapped to the original space to visualize their trajectories.

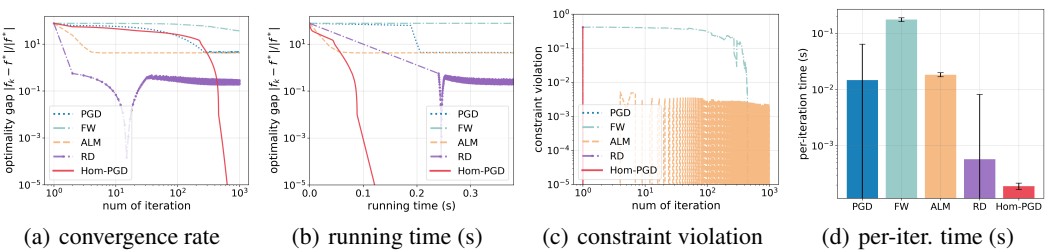

(a) convergence rate     (b) running time (s)     (c) constraint violation     (d) per-iter. time (s)

Figure 10: Convergence performance for optimization over a star-shaped set.

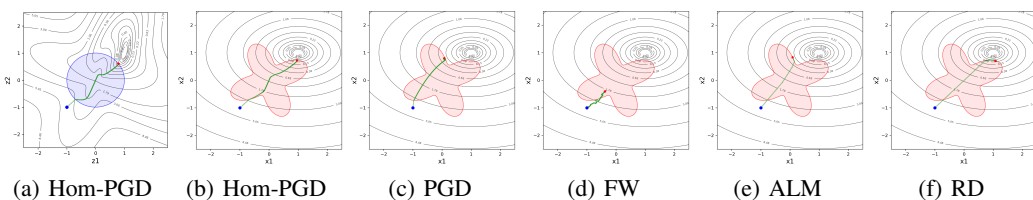

(a) Hom-PGD    (b) Hom-PGD    (c) PGD    (d) FW    (e) ALM    (f) RD

Figure 11: Iteration trajectory for optimization over a star-shaped set. Hom-PGD (a) and RD (f) are also mapped to the original space to visualize their trajectories.

## G.2 Second-order Cone Programming

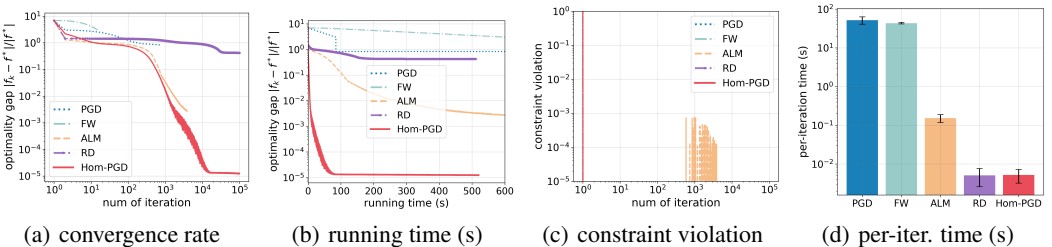

Figure 12: Convergence performance over SOCP with $(n, m) = (100, 1000)$.

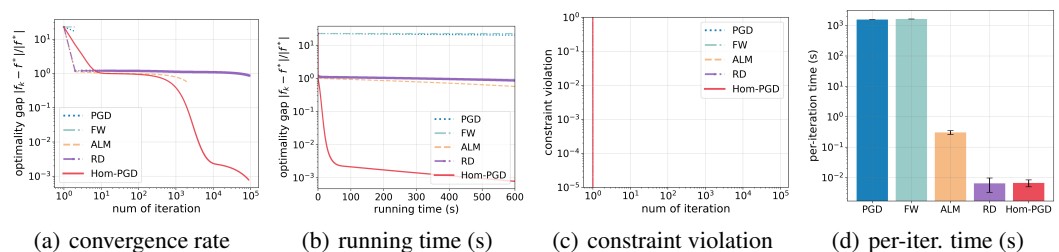

Figure 13: Convergence performance over SOCP with $(n, m) = (500, 1500)$.

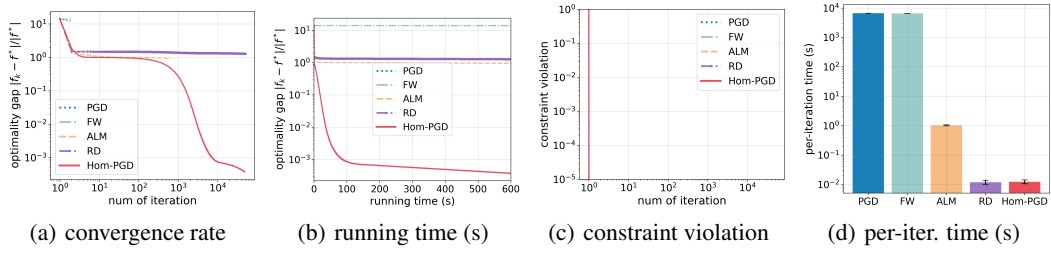

Figure 14: Convergence performance over SOCP with $(n, m) = (1000, 2500)$.

## G.3 Max-Cut Semidefinite Programming

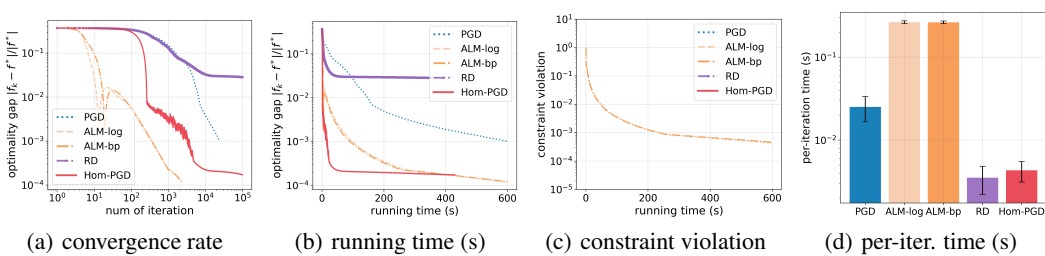

Figure 15: Convergence performance over SDP with $n = 10^2$.

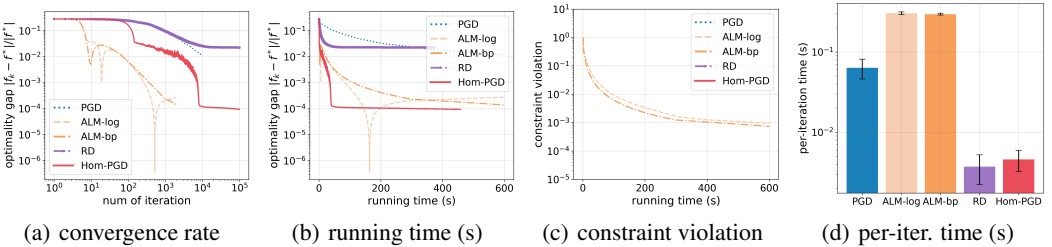

(a) convergence rate      (b) running time (s)      (c) constraint violation      (d) per-iter. time (s)

Figure 16: Convergence performance over SDP with $n = 20^2$.

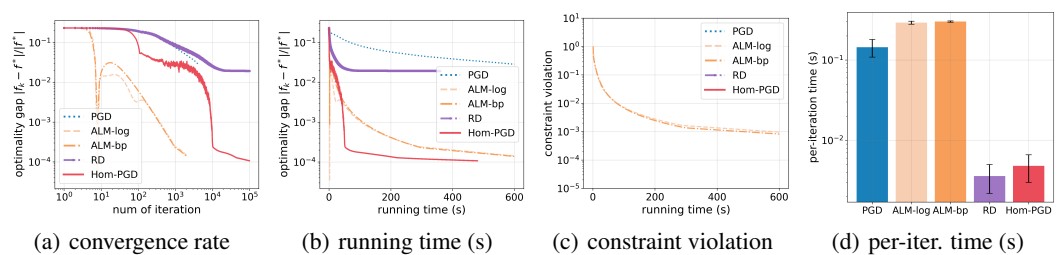

(a) convergence rate      (b) running time (s)      (c) constraint violation      (d) per-iter. time (s)

Figure 17: Convergence performance over SDP with $n = 30^2$.

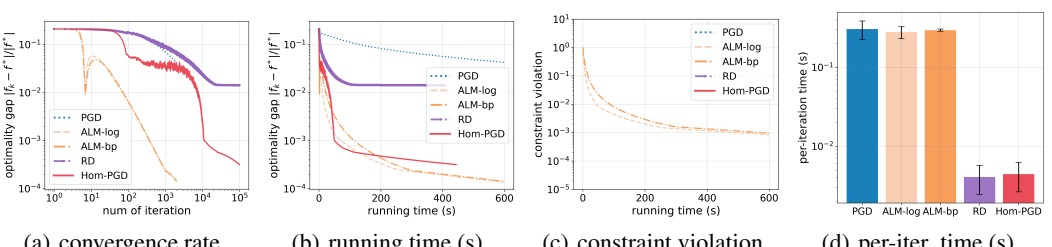

(a) convergence rate      (b) running time (s)      (c) constraint violation      (d) per-iter. time (s)

Figure 18: Convergence performance over SDP with $n = 40^2$.

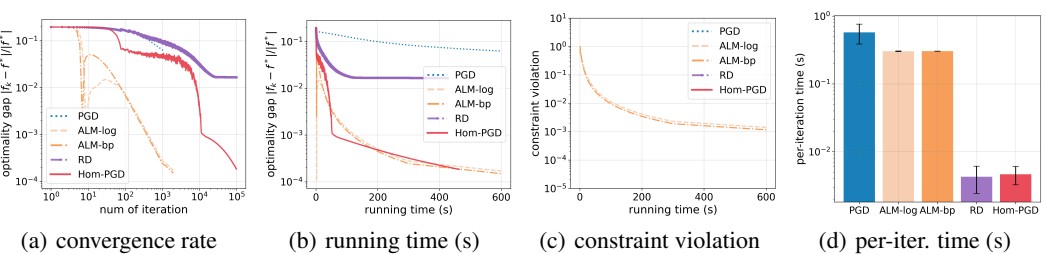

(a) convergence rate      (b) running time (s)      (c) constraint violation      (d) per-iter. time (s)

Figure 19: Convergence performance over SDP with $n = 50^2$.

## G.4 Ablation Study

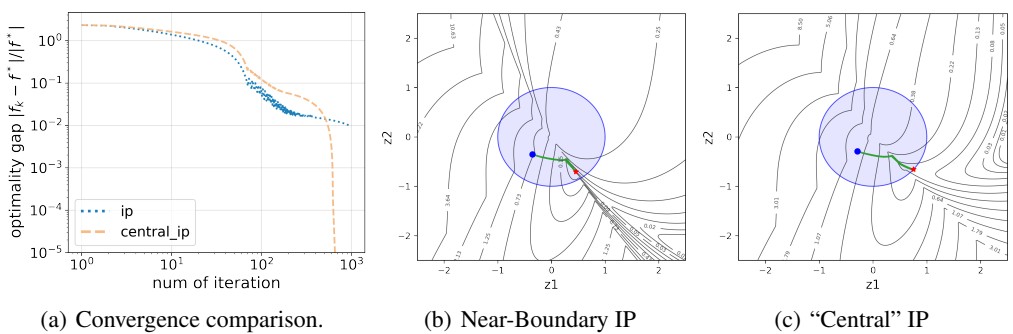

(a) Convergence comparison.  (b) Near-Boundary IP  (c) "Central" IP

Figure 20: Effect of **interior point** (IP) selection on convergence behavior: Gauge mapping with a near-boundary IP results in larger Bi-Lipschitz constants, which distorts the landscape of the transformed problem **H**, consequently reducing convergence speed in practice.

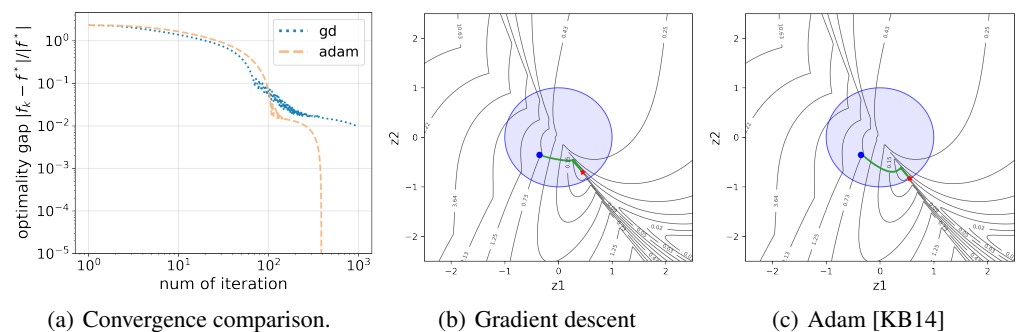

(a) Convergence comparison.  (b) Gradient descent  (c) Adam [KB14]

Figure 21: Comparison of **gradient methods**: We evaluate Hom-PGD under Gauge mapping with a near-boundary IP. In this non-convex landscape, standard gradient descent exhibits slower convergence, while the Adam optimizer demonstrates superior performance due to its momentum acceleration and adaptive step size adjustment.

