# OpenReview forum: "Fast Projection-Free Approach (without Optimization Oracle) for Optimization over Compact Convex Set"
_NeurIPS.cc/2025/Conference — NeurIPS 2025 spotlight_

### Official Review · Reviewer_ERsh · 2025-06-27

**Clarity:** 3
**Significance:** 2
**Originality:** 3
**Rating:** 5
**Confidence:** 3

**Summary:**

This paper proposes a novel method for solving constrained convex optimization problems. The main idea is to construct a homeomorphism between the original problem and a transformed problem with a unit ball constraint. The transformed problem can be solved without using expensive LOO step or complicated projection steps in PGD. The paper provides convergence guarantees for the proposed method in both convex and non-convex settings.

**Questions:**

- Why is the proposed method called "projection-free"? The transformed problem still involves a constraint (even though it is a unit ball with closed closed-form solution) and requires projection.

- In Figure 1, why is the initial point located outside the constrained set?

- The paper only evaluates toy examples. Could the method also be applied to practical settings such as adversarial attacks in neural networks, especially with some non-ball constraints? Can the author discuss under which setting the proposed method might be useful for deep learning or neural network training?

**Ethical Concerns:**

["NO or VERY MINOR ethics concerns only"]

**Final Justification:**

The detailed rebuttal has solved my concerns, and I will update the score.

**Limitations:**

yes

**Quality:**

3

**Strengths And Weaknesses:**

# Strengths
- The method is novel and effective. The homeomorphism and Gauge Mapping is well constructed and the connection between the Global Optimality of the original problem and the stationary point of transformed problem is good. (Proposition 5.2)

- The paper presents strong theoretical results to support the proposed method and shows that it can achieve the same optimal convergence rate as raw gradient descent.

- The empirical results are promising, showing faster convergence compared to LOO and projected gradient descent.

- The paper is well-written and well-structured, I do not find any typos, the visualization is good


# Weaknesses:
- Very minor problem: line 34 could be clarified. Specifically, the case “O(1/epsilon) in the convex setting” requires the gradient to be Lipschitz continuous, and the “O(1/epsilon^2) in the non-convex setting” seems to need additional assumption, i.e., gradient Lipschitz continuity.
-- Section 4.2 is more important than the theoretical analysis in Section 4.5 in my view because it discuss with the construction of the homeomorphism, which is crucial for practical implementation. However, Algorithm 1, Table 2, and some revalant discussion are deferred to the appendix. It would improve the paper to move them into the main body.
- The paper claims that a key benefit over FW (or LOO) is avoiding expensive linear minimization. It would be useful to include concrete examples of problems where this is indeed costly, to enhance the paper’s self-consistency. Additionally, the limitation section can be improved by considering the following question.: Are there any case where FW (or LOO) is more efficient than the proposed method? For instance, when the algorithm is slow, or when the closed-form expression in Table 2 is hard to derive while LOO is is easily to solve?

---

> ### Author Rebuttal · Authors · 2025-07-30
>
> ### Dear Reviewer ERsh,
>
> Thank you for your time and effort in reviewing our paper, and for your encouraging and insightful review. We have addressed your questions as follows one by one.
>
> ### Comments/Questions in *Weakness* section
> ---
>
> > **C1: Very minor problem: line 34 could be clarified. Specifically, the case "O(1/epsilon) in the convex setting" requires the gradient to be Lipschitz continuous, and the "O(1/epsilon$^2$) in the non-convex setting" seems to need additional assumptions, i.e., gradient Lipschitz continuity.**
>
> ---
>
>
> We thank the reviewer for this careful observation. You are correct that these complexity results require the gradient Lipschitz continuity assumption.
>
> In the revised version, we will modify line 34 to explicitly state those conditions. Thank you for helping us improve the precision of our presentation.
>
> ---
>
> > **C2: Section 4.2 is more important than the theoretical analysis in Section 4.5 in my view because it discuss with the construction of the homeomorphism, which is crucial for practical implementation. However, Algorithm 1, Table 2, and some relevant discussion are deferred to the appendix. It would improve the paper to move them into the main body.**
>
> ---
>
>
> Thank you for the helpful suggestion. We agree that the construction of the homeomorphism in Section 4.2 is central to the practical implementation and deserves more prominence. Due to space constraints, we initially placed Algorithm 1, Table 2, and the associated discussion in the appendix. However, since the final version allows for an additional page, we will move these elements into the main body to improve clarity and accessibility.
>
> ---
>
> > **C3: The paper claims that a key benefit over FW (or LOO) is avoiding expensive linear minimization. It would be useful to include concrete examples of problems where this is indeed costly, to enhance the paper's self-consistency. Additionally, the limitation section can be improved by considering the following question: Are there any case where FW (or LOO) is more efficient than the proposed method? For instance, when the algorithm is slow, or when the closed-form expression in Table 2 is hard to derive while LOO is easily to solve?**
>
> ---
>
>
>
> Thank you for this insightful comment and the suggestions for improving the limitations discussion.
>
> Regarding the cost of linear minimization, a concrete example is semidefinite programming (SDP). In this case, the linear optimization oracle (LOO) required by Frank-Wolfe (FW) corresponds to solving an SDP itself, which defeats the purpose of using FW as a low-cost alternative. Therefore, FW is not applicable in such settings.
>
> - This limitation is also reflected in our experimental benchmarks, where all tasks involve constraint sets for which LOO is expensive or impractical. To enhance the paper's self-consistency, we will include a small toy example in the introduction that highlights a case where linear minimization is prohibitively expensive.
>
> Regarding the reviewer's follow-up question on when FW (or LOO-based methods) might be more efficient than our approach, we offer the following considerations:
>
> - **Computational Cost:** Our method relies only on a *membership oracle* (one just needs to check whether a point is feasible), which is generally cheaper than a *linear optimization oracle* (one needs to solve a linear optimization over the feasible set) for general convex sets.
>     - Intuitively, solving the LOO at least requires evaluating the constraint/objective function, and the membership oracle only needs to evaluate the constraint to check feasibility.
>
> - **Convergence Behavior:** Under standard assumptions (e.g., smoothness and convexity), the convergence rates of FW and our method are comparable, even though there are slightly different conditions for the two settings.
>
> Therefore, our proposed method has a lower run-time cost than FW methods overall.
>
> We will revise the limitations section to include this discussion in the final version of our paper.
>
> ---
>
> ### Questions
>
> ---
>
> > **Q1: Why is the proposed method called "projection-free"? The transformed problem still involves a constraint (even though it is a unit ball with closed closed-form solution) and requires projection.**
>
> ---
>
>
> The term "projection-free" specifically refers to avoiding expensive projections onto original complex constraint sets. Hence while in our work, we use the term "projection-free" due to the closed and trivial projection operator, which is a usage that aligns with standard conventions in the literature e.g.:
>
> - Li Q, McKenzie D, Yin W. *From the simplex to the sphere: faster constrained optimization using the Hadamard parametrization*. Information and Inference: A Journal of the IMA, 2023, 12(3): 1898–1937.
> - Lu Z, Brukhim N, Gradu P, et al. Projection-free adaptive regret with membership oracles[C]//International Conference on Algorithmic Learning Theory. PMLR, 2023: 1055-1073.
>
> We will clarify this in the revised version to prevent any misunderstanding.
>
> ---
>
> > **Q2: In Figure 1, why is the initial point located outside the constrained set?**
>
> ---
>
>
> For the PGD algorithm, the initial point is randomly generated and does not need to be feasible because the projection operator will make each iteration point feasible. In Fig. 1, we randomly generate an initial point, which is located outside coincidentally.
>
> ---
>
> > **Q3: The paper only evaluates toy examples. Could the method also be applied to practical settings such as adversarial attacks in neural networks, especially with some non-ball constraints? Can the author discuss under which setting the proposed method might be useful for deep learning or neural network training?**
>
> ---
>
>
>
> We thank the reviewer for this excellent suggestion to connect our work to deep learning applications.
>
> While our experiments include large-scale SOCP and SDP problems with practical importance, we agree that demonstrating applications in deep learning would strengthen the paper's impact. Our method is indeed applicable for adversarial robustness and constrained neural network training.
>
> - **Adversarial Attack**: Standard PGD-based attacks use simple $\ell_2$ or $\ell_\infty$ ball constraints where projection is trivial. However, recent work has shown that more sophisticated constraint sets (e.g., non-uniform bounds) can be more effective. For these cases, projection becomes computationally expensive, making our approach advantageous.
>
>     - For instance, when perturbations are constrained to non-uniform balls [1], Hom-PGD can efficiently handle these constraints via gauge mapping.
>     - We include experiments comparing standard PGD, Non-orthogonal PGD [1], and Hom-PGD on MNIST with various non-ball perturbation constraints:
>
>
> |               | $\|Wv\|_{0.5} \le 1$ | $\|Wv\|_4 \le 0.1$ |
> |---------------|----------------------|--------------------|
> | PGD           | N/A (non-convex)     | N/A (polynomial)   |
> | Non-orthogonal-PGD | 0%, 0.0032s      | 12.7%, 0.0030s     |
> | Hom-PGD       | 0.1%, 0.0034s        | 12.2%, 0.0033s     |
>
> *Table: Attack success rate and per-iteration cost of PGD, Non-orthogonal-PGD, and Hom-PGD for trained CNN under 1,000 input samples with Non-uniform Perturbations [1] in MNIST dataset*
>
>
>
>
>
> - **Constrained NN Policy**: Our method also applies to safety-critical applications where neural network outputs must satisfy convex constraints [2]. By transforming the output space via gauge mapping, we can train networks with guaranteed constraint satisfaction while maintaining efficient gradient updates.
>
> We will add those discussions to Sections 5 and 6 in the final version, including deep learning applications with experimental validation, demonstrating the practical relevance of our approach beyond traditional convex optimization problems.
>
> **Reference:**
>
> - [1] Erdemir E, Bickford J, Melis L, et al. *Adversarial robustness with non-uniform perturbations*. NeurIPS 2021.
> - [2] Tabas D and Zhang B. Computationally efficient safe reinforcement learning for power562systems. ACC 2022

---

> > ### Comment · Reviewer_ERsh · 2025-08-01
> > **thanks for the rebuttal**
> >
> > The detailed rebuttal has solved my concerns, and I will update the score.

---

> > > ### Author Response · Authors · 2025-08-06
> > > **Thanks for your acklowedgments!**
> > >
> > > Dear Reviewer ERsh
> > >
> > > We appreciate your careful consideration of our rebuttal and are pleased that it addressed your concerns.
> > >
> > > Your suggestions regarding adversarial attack experiments have broadened the application scenarios of our work, and we will incorporate these valuable insights into the final version.
> > >
> > > We are grateful for your thoughtful feedback throughout this process, as it has significantly improved the quality of our research.
> > >
> > > Best
> > > Authors

---

### Official Review · Reviewer_HSQR · 2025-06-29

**Clarity:** 4
**Significance:** 3
**Originality:** 3
**Rating:** 5
**Confidence:** 3

**Summary:**

The paper investigates projection-free first-order methods for optimization over a compact and convex set. The key contribution of the paper is to propose an approach, Hom-PGD, which contrary to the popular Frank-Wolfe algorithm does not require a linear minimization oracle (LOO). The main insight is to transform the convex set into a ball, and then solve the transformed problem. The approach achieves optimal convergence rates while enjoying low per-iteration complexity. The experiments show a speed up of 3-6 orders of magnitude in the per-iteration runtime for some problem.

**Questions:**

- Which sets are best handled by the proposed approach versus existing algorithms, and which sets admit a closed-form gauge mapping?
- Does the speed up demonstrated in the experiments hold even for complicated sets that do not admit a closed-form gauge mapping?
- The behavior of the ALM approach in Figure 4 seems unexpected. Is the large optimality gap due to the suboptimality gap in the SDP relaxation? Otherwise, I would expect Burer Monteiro to converge fairly quickly?

**Ethical Concerns:**

["NO or VERY MINOR ethics concerns only"]

**Final Justification:**

The paper presents a strong and interesting contribution. The proposed idea is elegant and effective and the authors have addressed my remaining concerns.

**Limitations:**

yes

**Quality:**

4

**Strengths And Weaknesses:**

Strengths:
- The paper is very clear and easy to follow. Abstract and introduction provide compelling problem statement and motivations.
- The idea is elegant in its simplicity: transform the convex set into a ball and optimize over the unit ball instead!Proposition 5.2 shows that while this transformation might break convexity, stationary points of the nonconvex problem can be mapped to global optima of the original problem.
- The convergence analysis is comprehensive (e.g., it covers the convex, strongly convex and non-convex case) and reports optimal convergence rates.
- The experiments show impressive results, with a 4-6 orders of magnitude per-iteration runtime reduction compared to baselines.

Weaknesses:
- In general, the paper remains vague about which sets are best handled by the proposed approach versus existing algorithms, and which sets admit a closed-form gauge mapping. This prevents the reader from fully understanding the impact of the proposed method.
- The experiments would benefit from adding a breakdown of the runtime for the initialization and the last-step computation for Hom-PGD.
- A main concern is that the experiments do not clarify if the considered problem instances admit a closed-form gauge mapping. In other words, does the speed up demonstrated in the experiments hold even for complicated sets that do not admit a closed-form gauge mapping?
- The behavior of the ALM approach in Figure 4 seems unexpected. Is the large optimality gap due to the suboptimality gap in the SDP relaxation? Otherwise, I would expect Burer Monteiro to converge fairly quickly?

Minor weaknesses:
- Line 31: “Despite their slower convergence rate, …”: this sentence is not grammatically correct as written.
- Right after stating problem (P), it would be useful to provide an example of problems where existing methods do not perform well. This would help the reader visualize the type of constraints that make the problem challenging.
- Line 90: The paper structure feels a bit redundant and could be merged into the overview of contributions at the end of Section 1.
- Footnote 2 on page 3 should refer to Appendix B.1 instead of B.2 I think.

---

> ### Author Rebuttal · Authors · 2025-07-30
>
> ### Dear Reviewer HSQR,
>
> Thank you for your time and effort in reviewing our paper, and for your encouraging and insightful review. We have addressed your questions as follows one by one.
>
> **We note that the comments in the Weakness section cover the questions.** Therefore, we list the comments in Weakness section and give the responses in the following.
>
> ---
>
> > **C1: In general, the paper remains vague about which sets are best handled by the proposed approach versus existing algorithms, and which sets admit a closed-form gauge mapping. This prevents the reader from fully understanding the impact of the proposed method.**
>
> ---
>
>
>
> Thank you for the insightful comment.  We agree that clarifying which constraint sets are best suited for our approach is important for understanding its practical impact.
>
> Our approach is particularly effective for constraint sets that admit closed-form gauge mappings, which encompasses a broad class of important convex optimization problems, including: QP, SOCP, and SDP. Besides, it also works for general compact convex sets with bisection-induced gauge mapping.
> - We provide a comprehensive treatment of these cases in Appendix B.4, with specific examples and closed-form expressions summarized in Table 2. (Due to space limitations, this discussion was placed in the appendix.)
>
> The key advantage of our method over existing algorithms lies in its ability to handle **general compact convex** constraint sets where projection oracles or linear minimization oracles are computationally expensive or unavailable. While we briefly mentioned this in the introduction, we agree it deserves more prominent placement.
>
> We appreciate the reviewer's guidance in improving the clarity and impact of our presentation.
>
> ---
>
> > **C2: The experiments would benefit from adding a breakdown of the runtime for the initialization and the last-step computation for Hom-PGD.**
>
> ---
>
>
> Thank you for the suggestion.
>
> We thank the reviewer for this constructive suggestion. While the initialization (*interior point computation*) and final-step (*gauge map computation*) computation costs are minor compared to the total iteration cost, we agree that providing this breakdown would enhance transparency and completeness of our experimental analysis.
>
> - For example, as shown in the table, the initialization and final-step computations together account for less than 0.1% of the total runtime, confirming that the computational overhead of our method is minimal.
>
> | Problem     | initialization     | total iteration     | final-step     |
> |-------------|-----------|-----------|----------|
> | SOCP        | 0.060%    | 99.93%    | 0.0003%  |
> | MaxCut-SDP  | 0 (explicit interior point) | 99.99% | 0.0002%  |
> ||
>
> *Table: Percentage of initialization/iteration/last-step running time for 50-dim SOCP and 50×50 MAX-CUT SDP*
>
> We will also include these breakdown tables alongside our main experimental results for problems of different dimensions in the final version of our paper to enhance the completeness. Thanks for your suggestion.
>
>
> ---
>
> > **C3: A main concern is that the experiments do not clarify if the considered problem instances admit a closed-form gauge mapping. In other words, does the speed up demonstrated in the experiments hold even for complicated sets that do not admit a closed-form gauge mapping?**
>
> ---
>
>
> Thank you for raising this important point.
>
> - All problem instances in our experiments (SOCP and SDP) do admit **closed-form** gauge mappings. We selected these problems because they represent widely-used convex optimization classes with significant practical importance in systems processing and machine learning applications. Specifically, SDP is one of the most general convex program classes that still admits efficient closed-form gauge mappings.
>
> - While in the **general convex** case, the gauge map can be efficiently computed via bisection (see *Appendix B.4* for details) and our theoretical results guarantee that the proposed method remains applicable, we agree that it is valuable to evaluate performance in such scenarios.
>    - For example, we test polynomial constraints using the sum-of-squares formulation (common in robotic control): for such $m$ polynomial inequalities with $n$ variables and degree $d$ (with total terms of $\binom{n+d/2}{d/2}^2$), the gauge function computation via bisection (tolerance of $10^{-5}$) on a MacBook Pro 2023 costs is presented in the follow table:
>
> | $m$ | $n$   | $d$  | $\binom{n+d/2}{d/2}^2$ | Gauge calculation (s) |
> |-----|-------|------|-------------------------|------------------------|
> | 10  | 10    | 4    | 4,356                   | 0.0250                 |
> | 10  | 50    | 4    | 1,758,276               | 0.4919                 |
> | 10  | 100   | 4    | 26,532,801              | 5.5915                 |
> ||
>
> We will further include additional experiments on such polynomial-constrained optimization problem instances without closed-form gauge mappings in the revised version to further support the generality and robustness of our approach.
>
>
> ---
>
> > **C4: The behavior of the ALM approach in Figure 4 seems unexpected. Is the large optimality gap due to the suboptimality gap in the SDP relaxation? Otherwise, I would expect Burer Monteiro to converge fairly quickly?**
>
> ---
>
>
> Thank you for the insightful comment.
>
> The large optimality gap is indeed due to the non-convexity introduced by the Burer-Monteiro (BM) reformulation, not the SDP relaxation itself.
> - When the rank parameter is small (e.g., $\log(d)$ in our experiments following the initialization setting of [1]), the BM approach can converge to poor local minima, which explains the behavior observed in Figure 4. While BM can converge quickly to a local minimum, the quality of this solution depends critically on the rank parameter choice.
>
> We will also explore adjusting the rank parameter to improve the performance and fairness of this comparison.
>
> - It's worth noting the fundamental difference between these approaches: BM methods solve a non-convex reformulation and provide optimality guarantees only in over-parameterized settings, while our method operates on a reparameterized problem via a homeomorphic mapping, guaranteeing global optimality. We include BM as a baseline because it represents the current state-of-the-art for large-scale SDP solving, making it an important reference point despite these fundamental differences.
>
>
> [1] Han, Qiushi, Zhenwei Lin, Hanwen Liu, Caihua Chen, Qi Deng, Dongdong Ge, and Yinyu Ye. "Accelerating low-rank factorization-based semidefinite programming algorithms on GPU." arXiv preprint arXiv:2407.15049 (2024).
>
> ---
>
> ### Minor Comments
>
> ---
>
> > **C5: Line 31: "Despite their slower convergence rate, ...": this sentence is not grammatically correct as written.**
>
> ---
>
>
> Thank you for pointing this out. We will revise the sentence to correct the grammar in the final version of the paper.
>
> ---
>
> > **C6: Right after stating problem (P), it would be useful to provide an example of problems where existing methods do not perform well. This would help the reader visualize the type of constraints that make the problem challenging.**
>
> ---
>
>
> Thank you for the suggestion. We agree that providing a concrete example early on helps clarify the motivation. Initially, we included a toy example in Figure 1 that illustrates how existing general methods can struggle with general constrained problems due to the high cost of the projection operator. To improve clarity and flow, we will provide an illustrative example where existing methods do not perform well closer to the statement of problem (P) in the revised version.
>
> ---
>
> > **C7: Line 90: The paper structure feels a bit redundant and could be merged into the overview of contributions at the end of Section 1.**
>
> ---
>
>
> Thank you for the helpful suggestion.
>
> We agree that the structure summary in Line 90 may be redundant. In the revised version, we will merge it into the overview of contributions at the end of Section 1 to improve clarity and reduce repetition.
>
> ---
>
> > **C8: Footnote 2 on page 3 should refer to Appendix B.1 instead of B.2 I think.**
>
> ---
>
>
> Thank you for pointing this out. We will revise it in the final version of the paper.

---

> > ### Comment · Reviewer_HSQR · 2025-08-02
> > **followup**
> >
> > I appreciated the authors' informative and comprehensive rebuttal. I do recommend improving the comparison with Burer-Monteiro, but otherwise I consider this a very good paper and recommend acceptance.

---

### Official Review · Reviewer_w43j · 2025-07-01

**Clarity:** 4
**Significance:** 3
**Originality:** 3
**Rating:** 5
**Confidence:** 3

**Summary:**

In this paper, the authors propose **Hom-PGD**, a new projection-free first-order method for solving convex optimization problems over general compact convex sets. Hom-PGD reparameterizes the original problem into an equivalent optimization problem constrained to a Euclidean ball, applies standard PGD to solve the reparameterized problem, and then maps the solution back to the original domain. The method achieves optimal convergence rates in the unaccelerated setting and maintains a low per-iteration complexity of $\mathcal{O}(n^2)$.

**Questions:**

See the Weaknesses section.

One minor question: Hom-PGD still requires projection onto a ball. Given this, it is unclear whether "projection-free" is the most appropriate term to describe the method.

**Ethical Concerns:**

["NO or VERY MINOR ethics concerns only"]

**Final Justification:**

The authors provided detailed rebuttal and addressed my concerns and questions. Therefore, I raised my score.

**Limitations:**

Yes

**Quality:**

4

**Strengths And Weaknesses:**

**Strengths:**

1. The paper presents a thorough literature review of projection-free methods and clearly communicates the theoretical results. Overall, it is well-written, demonstrating both quality and clarity.

2. The proposed algorithm is the first projection-free, first-order framework capable of solving optimization problems over general compact convex sets. It achieves the optimal convergence rate in the unaccelerated setting without relying on expensive oracles, highlighting its significance and originality.

**Weaknesses**

1. The LICQ assumption in Proposition 5.2 appears to be somewhat strong and may limit the applicability of the proposed algorithm. Is it possible to relax this assumption by adopting a weaker constraint qualification?

2. Similarly, the strict complementary slackness assumption in Theorem 1 could be restrictive. Can this assumption be relaxed?

3. In line 261, the authors claim that the proposed algorithm has significantly lower complexity compared to second-order methods. However, is this always the case for first-order methods? It may be beneficial to include a section comparing the overall computational cost with both other first-order and second-order methods.

---

> ### Author Rebuttal · Authors · 2025-07-30
>
> ### Dear Reviewer w43j,
>
> Thank you for the time and effort in giving our paper a thorough review. We are very happy to receive your comments and suggestions. For your concerns and questions, we address them one by one as follows.
>
> ---
>
> > **Q1: The LICQ assumption in Proposition 5.2 appears to be somewhat strong and may limit the applicability of the proposed algorithm. Is it possible to relax this assumption by adopting a weaker constraint qualification?**
>
> ---
>
>
> Thank you for the thoughtful question.
>
> We acknowledge that the Linear Independence Constraint Qualification (LICQ) can be strong in general. However, it is a standard assumption widely adopted in the optimization literature to ensure the validity of KKT conditions and local optimality results.
>
> - In our specific setting, the LICQ assumption is not restrictive. For problem $\mathcal{H}$, the only constraint is $||z||^2 - 1 \leq 0$, which defines the unit Euclidean ball. By definition, LICQ holds at any boundary point $z^*$ where the constraint is active, as the gradient of the constraint is non-zero. Therefore, LICQ is automatically satisfied in our case.
>
> - More generally, the LICQ assumption can indeed be relaxed. In Appendix D, we show that our results hold under the weaker Mangasarian-Fromovitz Constraint Qualification (MFCQ) (see Definition D.3 and Proposition D.6). Here, one should note LICQ can imply MFCQ thereby MFCQ is a weaker condition. Additionally, other constraint qualifications like Slater's condition may be appropriate in convex settings (if problem H is also convex). Exploring weaker or alternative constraint qualifications for broader applicability is a valuable direction for future work.
>
> We will update the paper to clarify these points in the final version.
>
> ---
>
> > **Q2: Similarly, the strict complementary slackness assumption in Theorem 1 could be restrictive. Can this assumption be relaxed?**
>
> ---
>
>
> Thank you for raising this important point.
>
> While the strict complementary slackness assumption can be strong in general, it is a standard condition commonly used in the optimization literature to ensure uniqueness and stability of KKT multipliers. Its use allows for cleaner theoretical analysis and has precedent in many related works.
>
> Relaxing this assumption within our current proof framework is non-trivial, as it plays a key role in the analysis of KKT conditions and directional optimality. Addressing this would require more delicate treatment of degenerate cases and potentially different analytical tools. We view this as an interesting open problem and a promising direction for future work.
>
> We will revise the paper to clarify the role of this assumption and discuss its potential relaxation in the final version.
>
> ---
>
> > **Q3: In line 261, the authors claim that the proposed algorithm has significantly lower complexity compared to second-order methods. However, is this always the case for first-order methods? It may be beneficial to include a section comparing the overall computational cost with both other first-order and second-order methods.**
>
> ---
>
>
> Thanks for the insightful question. While first-order methods generally have lower per-iteration complexity than second-order methods (avoiding expensive Hessian computations), a more nuanced analysis would indeed be valuable.
>
> -  In our submitted manuscript (lines 306–308), we indeed compare the run-time complexity with a second-order method empirically in a 1000-dim SOCP.
> - Specifically, we compare our methods to a commercial solver, MOSEK (which uses highly optimized second-order interior point methods).
> - MOSEK costs 5424 seconds to solve the 1000-dim instance, while Hom-PGD takes less than 600 seconds to reach a similar optimality gap.
>
> We agree that a more comprehensive comparison of different problems would further strengthen the paper. We will include a new table in the final version that provides comprehensive run-time complexity for both first-order and second-order baselines.
>
> ---
>
> > **Q4: Hom-PGD still requires projection onto a ball. Given this, it is unclear whether "projection-free" is the most appropriate term to describe the method.**
>
> ---
> Thanks for the insightful question.
>
> While Hom-PGD involves projection onto a Euclidean ball, this operation has a closed form and is trivial. The term "projection-free" specifically refers to avoiding expensive projections onto original complex constraint sets—a usage that aligns with standard conventions in the literature, e.g., [1,2].
>
> We will clarify this in the revised version to prevent any misunderstanding.
>
> - [1] Li Q, McKenzie D, Yin W. *From the simplex to the sphere: faster constrained optimization using the Hadamard parametrization*. Information and Inference: A Journal of the IMA, 2023, 12(3): 1898-1937.
> - [2] Lu Z, Brukhim N, Gradu P, et al. *Projection-free adaptive regret with membership oracles*. In *International Conference on Algorithmic Learning Theory*. PMLR, 2023: 1055-1073.

---

> > ### Comment · Reviewer_w43j · 2025-08-06
> >
> > I thank the authors for the detailed rebuttal. Since it mostly addressed my concerns, I am happy to raise my rating.

---

### Official Review · Reviewer_4NvD · 2025-07-03

**Clarity:** 4
**Significance:** 4
**Originality:** 4
**Rating:** 6
**Confidence:** 3

**Summary:**

This paper presents a new, simple approach to solving a constrained convex optimization problem. The idea is to compute a homeomorphism from the convex constraint set to the unit ball, and solve the “homeomorphic constrained optimization” problem. To do so, they consider standard projected gradient descent (PGD) in the homeomorphic optimization problem. They provide convergence analysis, complexity analysis, and experiments. They suggest a specific example of a homeomorphism (the gauge/Minkowski function), and provide details about how to efficiently compute/approximate it.

**Questions:**

I do not really have additional questions beyond the ones I listed above. Some questions that I'm curious about, but are not relevant to how I score the paper, nor do they necessarily need a response:

1. If we only care about the convergence rate in terms of iterations of projected gradient descent, do you think there would be any advantage in recomputing the gauge map centered at a point $\mathbf{x}^\circ$ close to the iterates? Or, is it better to just choose $\mathbf{x}^\circ$ to be some sort of centroid of $\mathcal{K}$?

2. You mentioned this approach builds upon [LMY23], which focuses on optimization on the probability simplex. In the case the objective is the simplex, another approach to deal with constraints is to run mirror descent (where the dual space is now unconstrained). The cost here is computing the mirror map versus the gauge map. Do you have a sense for which approach would be favorable?

3. How amenable do you think this homeomorphism approach is to second-order methods?

**Ethical Concerns:**

["NO or VERY MINOR ethics concerns only"]

**Final Justification:**

The paper is a great contribution.

**Limitations:**

yes

**Quality:**

4

**Strengths And Weaknesses:**

**Strengths.**
This paper was really a pleasure to read, and introduces a very natural approach to solving constrained convex problems. The theory seems pretty complete. The experiments looked convincing, with quite significant speed-ups in terms of wall-clock time.

**Weaknesses.**
I do not see any significant weaknesses in this paper. Perhaps it would be nice to include a Hom-PGD pseudocode block somewhere, say in the appendix, and also consider having Equation 35 in the main paper. I think perhaps the minor questions I would have are:

1. The theory assumes that the homeomorphism is twice-differentiable, while the gauge function is only twice-differentiable almost everywhere. While the iterates will almost surely never land on a point at which $\psi$ is not smooth, does this pose a problem in the analysis when specialized to the gauge homeomorphism?

2. The theoretical analysis assumes access to exact gradients and evaluations of the homeomorphism, while in the algorithm, the gradients and gauge map are approximated. It may be reasonable to provide bounds on how accurately these need to be approximated in order to achieve a target approximation error. However, computing such a bound seems like it would be fairly tedious but not necessarily enlightening.

---

> ### Author Rebuttal · Authors · 2025-07-30
>
> ### Dear Reviewer 4NvD,
>
> We appreciate your time and effort in reviewing our paper, and for your encouraging and insightful review. We greatly appreciate your recognition of our paper's clarity, theoretical completeness, and significant experimental results. We have addressed your questions one by one as follows.
>
> ### Comments/Questions in *Weakness* part
>
> ---
>
> > **C1: The theory assumes that the homeomorphism is twice-differentiable, while the gauge function is only twice-differentiable almost everywhere. While the iterates will almost surely never land on a point at which $\gamma$ is not smooth, does this pose a problem in the analysis when specialized to the gauge homeomorphism?**
>
> ---
>
> Thanks for the insightful question. We underscore that while the gauge function is twice differentiable almost everywhere (and iterates almost surely avoid the non-smooth points in practice), the assumption that the homeomorphism is twice differentiable is indeed required in theory.
>
> To bridge the gap, we can employ a classical smoothing technique to satisfy this assumption [1][2].
> - The non-smoothness arises from the maximum operator $\max \{\cdot\}$ when handling multiple constraints (see Table 2 in Appendix B.4).
> - Specifically, the gauge mapping $\psi$ can be defined by the inverse distance function $\kappa$, i.e, $\psi(z)=\kappa(x^\circ,z/|z|)\cdot z +x^\circ$ (see Appendx B.4).
> - For multiple constraints (Table 2), $\kappa(\cdot)$ can be written in the form $\kappa(x^\circ, \cdot)=\max_i{\kappa_i }$ where $\kappa_i$ is the individual inverse distance function for each constraint.
>
> We smooth this using the log-sum-exp approximation with parameter $\eta > 0$:
>
> $$
> \kappa_\eta(x) = \eta \log \left(\sum_{i=0}^n \exp\left(\frac{\kappa_i(x)}{\eta}\right)\right),
> $$
>
> with the corresponding gradient
>
> $$
> \nabla \kappa_\eta(x)=\sum_{i} \lambda_i \nabla \kappa_i(x) \quad \text{where} \quad \lambda_i=\frac{\exp\left(\kappa_i(x)/\eta\right)}{\sum_j \exp\left(\kappa_j(x)/\eta\right)}.
> $$
>
> By replacing the objective $h=f\circ\psi$ with $h=f\circ\psi_\eta$ where $\psi_{\eta}(\cdot)=\kappa_\eta(x^\circ,\cdot)\cdot z+x^\circ$, we obtain a twice-differentiable homeomorphism. The smoothing introduces an optimality gap of $\mathcal{O}(\eta)$ (here the hidden constant is dependent on the Lipschitz constant of $f$), which can be made arbitrarily small by choosing $\eta$ sufficiently small.
>
> In practice, this smoothing has a minor impact. Our new experiments show that Hom-PGD with the original gauge map and its smoothed version ($\eta=10^{-5}$) produce a similar convergence curve with respect to the optimality gap, but the smooth version costs a little bit more per-iteration computation time than the original gauge map.
>
> |            | Mean (s) | Std (s)         |
> |------------|----------|-----------------|
> | Original   | 0.0061   | 0.0033        |
> | Smoothed   | 0.0080   | 0.0086       |
>
> *Table: Comparison between Hom-PGD with gauge map and its smoothed version ($\eta=10^{-5}$) in 100-dim SOCP.*
>
> We will include this discussion and supporting experimental results in the final version. Thank you for highlighting this important theoretical consideration.
>
> Ref:
> - [1] Beck A, Teboulle M. Smoothing and first order methods: A unified framework[J]. SIAM Journal on Optimization, 2012, 22(2): 557-580.
> - [2] Grimmer B. Radial duality part II: applications and algorithms[J]. Mathematical Programming, 2024, 205(1): 69-105.
>
> ---
>
> > **C2: The theoretical analysis assumes access to exact gradients and evaluations of the homeomorphism, while in the algorithm, the gradients and gauge map are approximated. It may be reasonable to provide bounds on how accurately these need to be approximated in order to achieve a target approximation error. However, computing such a bound seems like it would be fairly tedious but not necessarily enlightening.**
>
> ---
>
> We appreciate the reviewer's insightful observation regarding the gap between theoretical assumptions and practical implementation.
>
> - Our theoretical analysis follows standard conventions in the optimization literature (e.g., [3]), assuming access to exact gradients and gauge mappings. This provides foundational insights, even though practical algorithms may rely on approximations.
> In more general cases where approximations are required, the incurred error is typically logarithmic (via bisection methods) and negligible in practice.
>
> - Nevertheless, we acknowledge that deriving explicit bounds on the accuracy of the gradient and gauge map approximations with respect to the final solution error is indeed an interesting direction, which has been studied in, e.g., [4].
> Typically, the optimality gap is augmented by an additional term of order \(\mathcal{O}(\delta)\), where \(\delta\) represents the error of the (gradient or gauge mapping) oracle at each iteration.
>
> We will add these discussions to the final version of our paper.
>
> Ref:
> - [3] Lu Z, Brukhim N, Gradu P, et al. Projection-free adaptive regret with membership oracles[C]//International Conference on Algorithmic Learning Theory. PMLR, 2023: 1055-1073.
> - [4] Devolder O, Glineur F, Nesterov Y. First-order methods of smooth convex optimization with inexact oracle[J]. Mathematical Programming, 2014, 146(1): 37-75.
>
> ---
>
> > **C3: Perhaps it would be nice to include a Hom-PGD pseudocode block somewhere, say in the appendix, and also consider having Equation 35 in the main paper.**
>
> ---
>
> Thank you for the suggestion. We will add it to the main body in the final version.
>
> ### Questions
>
> ---
>
> > **Q1: If we only care about the convergence rate in terms of iterations of projected gradient descent, do you think there would be any advantage in recomputing the gauge map centered at a point $x^\circ$ close to the iterates? Or, is it better to just choose $x^\circ$ to be some sort of centroid of $\mathcal{K}$?**
>
> ---
>
>
>
> Thanks for the insightful questions. We would like to clarify that it is not necessary—nor beneficial from a convergence standpoint—to continuously update $x^\circ$ during optimization.
>
> - From our theoretical analysis (Theorems 1, 2, and 3), the convergence rate depends primarily on the bi-Lipschitz constant of the gauge mapping, which is determined by the choice of the interior point $x^\circ$. Importantly, this dependency is independent of the current iterations.
>
> - Moreover, adaptively changing $x^\circ$ during the iterations could alter the gauge mapping at each step, effectively modifying the landscape of the optimization problem. This could destabilize the algorithm and even prevent convergence, as the optimization landscape would no longer be consistent across iterations.
>
> - In practice, recomputing the gauge map centered at a different point in each iteration is computationally expensive and may introduce additional errors.
>
> Therefore, choosing $x^\circ$ as a fixed central point (such as the centroid of $\mathcal{K}$) is preferable both theoretically and practically.
>
> ---
>
> > **Q2: You mentioned this approach builds upon [LMY23], which focuses on optimization on the probability simplex. In the case the objective is the simplex, another approach to deal with constraints is to run mirror descent (where the dual space is now unconstrained). The cost here is computing the mirror map versus the gauge map. Do you have a sense for which approach would be favorable?**
>
> ---
>
> Thanks for the interesting questions.
>
> - For optimization on the probability simplex specifically, both our gauge-based approach and mirror descent achieve comparable convergence rates. Mirror descent with entropic regularization is indeed well-established and efficient for this domain.
> - From a practical standpoint, our method may require tuning additional hyperparameters (e.g., the interior point $x^\circ$), while mirror descent for the simplex is more straightforward to implement with the standard entropic mirror map.
>
> However, the key advantage of our approach is its generality: it provides a unified framework that seamlessly extends to arbitrary compact convex sets without requiring problem-specific design. In contrast, mirror descent requires carefully crafted mirror maps for each domain, which may not always be available or computationally tractable. For instance, finding appropriate mirror maps for complex constraint sets (e.g., PSD cones with additional linear constraints) can be challenging, whereas our gauge-based approach handles these naturally.
>
> Thus, while mirror descent may be preferable for the simplex alone, our method offers broader applicability as a general-purpose constrained optimization framework.
>
> ---
>
> > **Q3: How amenable do you think this homeomorphism approach is to second-order methods?**
>
> ---
>
> Thanks for the insightful questions.
>
> - **Computational Complexity:** Second-order methods would require computing the Hessian of the composite objective $f\circ\psi$. Since the Hessian of the gauge map $\psi$ is a third-order tensor, this becomes both computationally intensive and memory-prohibitive. In contrast, our first-order approach only requires gradients, making it significantly more efficient.
>
> - **Convergence Behavior:**  We also remark that the composite objective $f\circ\psi$ is generally non-convex. Despite the well-established faster convergence behavior of second-order methods for convex problems, analyzing the convergence over a non-convex landscape to the global optimum is more challenging and would likely require additional smooth assumptions beyond the standard analysis used for second-order methods.
>
> While extending our framework to second-order methods is theoretically possible, these computational and analytical challenges suggest that first-order methods are more practical for this setting. We view second-order extensions as an intriguing direction for future research, particularly for special cases where the gauge mapping structure might simplify the Hessian computation.

---

### Note · Authors · 2025-08-15

Dear Reviewers, ACs, and SACs,

We would like to express our sincere gratitude to the reviewers and the area chair for their time, effort, and constructive feedback.

---
We appreciate the recognition of the **novelty**, **strength**, and **significance** of our contributions in proposing a fast, projection-free approach for optimization over general compact convex sets. In particular, we highlight the following contributions:

- We introduce **Hom-PGD**, a novel projection-free method that reparameterizes the original constrained optimization problem into an equivalent ball-constrained problem. By solving this reparameterized problem using gradient descent with a closed-form projection, and then mapping the solution back, we efficiently solve the original problem.

- We provide rigorous **convergence and complexity analyses** for Hom-PGD, demonstrating that it achieves the **optimal** convergence rate in the unaccelerated setting for both convex and non-convex objectives.

- Through extensive numerical experiments on both convex and non-convex problems, we show that Hom-PGD achieves a **4–6 orders of magnitude reduction in per-iteration runtime** compared to PGD/FW methods, verifying the practical efficiency of our approach.

---

During the rebuttal, we have addressed the concerns from reviewers, and will revise the paper to incorporate the reviewers' feedback and discussions in the final version:
- **Clarifications on assumptions and theoretical settings** (4NvD and w43j):
- **Conducting additional experiments**:
     - Applying the smoothing technique to meet the assumptions of our theoretical results (Reviewer 4NvD)
     - Including overall computational cost with both first- and second-order methods (Reviewer w43j),
     - Including initialization and final-step runtimes, addressing cases without closed-form gauge mappings, and retuning hyperparameters of BM methods for the SDP baselines (Reviewer HSQR),
     - Applications to deep learning tasks such as adversarial attacks (Reviewer ERsh).
- **Refining the presentation**:  we will also review and standardize notations, clarify definitions (e.g., “projection-free”), correct typos, and improve the overall structure and flow in the final version.

---

Once again, we thank the reviewers and chairs for their valuable feedback and the opportunity to further improve our work.

---

### Decision · Program_Chairs · 2025-09-17

**Decision:**

Accept (spotlight)

**Comment:**

This paper introduces Hom-PGD, a projection-free first-order method for optimizing over general compact convex sets without expensive linear minimization oracles. It leverages a homeomorphism between the constraint set and a unit ball to transform the original optimization problem into a ball-constrained formulation. The algorithm achieves optimal convergence rates matching gradient descent in (strongly) convex or non-convex settings. The paper received four detailed reviews, in which it is unanimously agreed that the proposed framework is novel and theoretically sound, and the demonstrated empirical speed-ups over baselines are impressive. Reviewers also raised  minor concerns regarding assumptions, terminology, and experimental transparency, which were effectively addressed in rebuttals.

Overall, based on the reviews, authors responses and discussions, it was assessed that this submission makes a sufficiently novel and significant contribution to the field of constrained optimization. The meta reviewer thus would be happy to recommend the paper for acceptance. Congratulations for the nice piece of work!